# Seasonal characteristics of trace gas transport into the extratropical upper troposphere/lower stratosphere

Yoichi Inai[1], Ryo Fujita[2,1], Toshinobu Machida[3], Hidekazu Matsueda[4], Yousuke Sawa[4], Kazuhiro Tsuboi[4], Keiichi Katsumata[3,5], Shinji Morimoto[1], Shuji Aoki[1], Takakiyo Nakazawa[1]

[1] Center for Atmospheric and Oceanic Studies, Graduate School of Science, Tohoku University, Sendai, 980-8578, Japan
[2] Department of Physics, Imperial College London, South Kensington Campus, London SW7 2AZ, United Kingdom
[3] National Institute for Environmental Studies, Tsukuba, 305-8506, Japan
[4] Meteorological Research Institute, Tsukuba, 305-0052, Japan
[5] Now at Takachiho Chemical Industrial Co., Ltd., Tokyo, 194-0004, Japan

*Correspondence to*: Yoichi Inai (yoichi_inai@tohoku.ac.jp)

**Abstract.** To investigate the seasonal characteristics of trace gas distributions in the extratropical upper troposphere and lower stratosphere (ExUTLS) as well as stratosphere–troposphere exchange processes, origin fractions of air masses originating in the stratosphere, tropical troposphere, mid-latitude lower troposphere (LT), and high-latitude LT in the ExUTLS are estimated using 10-year backward trajectories calculated with European Centre For Medium-Range Weather Forecasts (ECMWF) ERA-Interim data as the meteorological input. Time-series of trace gases obtained from ground-based and airborne observations are incorporated into the trajectories, thus reconstructing spatiotemporal distributions of trace gases in the ExUTLS. The reconstructed tracer distributions are analysed with the origin fractions and the stratospheric age of air (AoA) estimated using the backward trajectories. The reconstructed distributions of $SF_6$ and $CO_2$ in the ExUTLS are linearly correlated with that of AoA because of their chemically passive behavior and quasi-stable increasing trends in the troposphere. Distributions of $CH_4$, $N_2O$, and $CO$ are controlled primarily by chemical decay along the transport path from the source region via the stratosphere and subsequent mixing such stratospheric air masses with tropospheric air masses in the ExUTLS.

## 1 Introduction

The extra-tropical upper troposphere and lower stratosphere (ExUTLS; e.g., Gettelman et al., 2011) accounts for about 40 % of the total stratospheric air mass (Appenzeller et al., 1996) and about 20 % of stratospheric aerosols (Andersson et al., 2015). Trace gases and aerosols in the ExUTLS play an important role in atmospheric radiative processes. These species are transported to the ExUTLS from the deep stratosphere via stratospheric circulation (Brewer–Dobson circulation, BDC; Brewer, 1949; Dobson, 1956) and from the lower troposphere or the tropical troposphere via local convection, frontal cyclones, Rossby wave breaking at/along the subtropical jet, monsoon activity, and other systems (e.g., Holton et al., 1995;

Wernli and Bourqui, 2002; Manney et al., 2011; Pan et al., 2016; Vogel et al., 2016; Boothe and Homeyer, 2017; Ploeger et al., 2017).

Air-mass transport processes into the ExUTLS are strongly dependent on the season. This leads to stratospheric and tropospheric mixing fractions that show clear seasonality. For example, Appenzeller et al. (1996) estimated the mass flux across the 380 K isentrope due to global-scale meridional circulation and found that the downwelling mass flux from the stratosphere varies from $8 \times 10^9$ kg s$^{-1}$ in summer to $15 \times 10^9$ kg s$^{-1}$ in winter, whereas the Asian summer monsoon and local convection, which supply tropospheric air to the ExUTLS, are active only during the summer and early autumn (e.g., Randel and Park, 2006; Randel et al., 2010). The composition of air masses transported from the deep stratosphere, lower troposphere, and tropical troposphere also shows seasonal variations (e.g., Boenisch et al., 2009). The seasonal variability in air-mass composition and mass-flux strength makes it difficult to essentially understand the distributions of trace gases in the ExUTLS and to describe their transport into the layer.

This study focuses on mixing fractions of air masses originating in the stratosphere, tropical troposphere, mid-latitude lower troposphere (LT), and high-latitude LT (hereafter, referred to as "origin fractions") in the ExUTLS, based on the trajectory analysis of Inai (2018). Using estimated origin fractions, the transport of chemical species into the ExUTLS and the spatiotemporal distributions of methane ($CH_4$), nitrous oxide ($N_2O$), carbon monoxide (CO), sulphur hexafluoride ($SF_6$), and carbon dioxide ($CO_2$) in the layer are reconstructed with the aid of atmospheric trace gas observations including aircraft measurements, such as those of the Comprehensive Observation Network for TRace gases by AIrLiner (CONTRAIL; Nakazawa et al., 1993; Matsueda and Inoue, 1996; Ishijima et al., 2001; Matsueda et al., 2002; Machida et al., 2008; Umezawa et al., 2014; Sawa et al., 2015). Reconstructed distributions for the five species are discussed in terms of dynamical transport as well as chemical loss, using the stratospheric age of air (AoA) as an indicator of air mass transport via the deep and shallow branches of the BDC.

## 2 Methods

### 2.1 Estimating the origin fraction and age of air

The CONTRAIL data were obtained by collecting air samples once a month from April 2012 to December 2016 at longitudinal intervals of 10° or 15° along individual flight tracks at around 11 km altitude between France/Russia and Japan. The period and longitudinal locations of this analysis were selected based on the CONTRAIL measurements, for which air sampling in the ExUTLS was usually made over Siberia. To identify the origins of ExUTLS air masses, kinematic backward trajectories are calculated for 10 years following the method of Inai (2018). Trajectories are initialized at uniformly distributed grid points (5.0° longitude × 2.5° latitude) within 45° N–80° N and 0° E–140° E at geopotential heights of 5, 6, 7, 8, 9, 10, 11, 12, 13, 14, 15, and 16 km (Fig. 1). Initializations are made at 00:00 UTC on the 5th, 15th, and 25th of every month from January 2012 to December 2016, and use meteorological conditions prescribed by the European Centre for Medium-Range Weather Forecasts (ECMWF) ERA-Interim dataset (1.5° × 1.5° horizontal resolution, 6 hourly temporal

resolution, and 37 pressure levels; Dee et al., 2011). Although trajectories could be released at the exact CONTRAIL measurement locations and times, the grating initialization is employed because this study attempt to obtain uniform spatiotemporal tracer distributions as well as their transports by capitalizing on the CONTRIAL measurements. An example of the results is provided in Fig. 2, which shows where particles located as shown in Fig. 1 at 00Z on 15 January 2015 were

5 located 361 days prior (i.e., 00Z on 19 January 2014). Many particles ending up at altitudes greater than 13 km (orange dots) travelled from the stratosphere, above 18 km. However, many particles ending up at altitudes below 10 km (purple to blue-green dots) distribute below 15 km, typically in the troposphere. Although the accuracy of individual trajectories is limited by the long-term nature of the calculations, statistical features of air-mass transport can be investigated using a large number of trajectories.

Trajectories obtained from each run are categorized into several groups ($trj_k$; $k = 1$ to $kmax$) with criteria (hereafter denoted $cri_k$) of potential temperature, latitude, potential vorticity, and geopotential height along each trajectory. In this analysis, $kmax$ is set to 4, with $k = 1$ for the stratosphere, $k = 2$ for the tropical troposphere, $k = 3$ for the mid-latitude LT, and $k = 4$ for the high-latitude LT. Criteria for each $k$ are summarized in Table 1. The trajectories are also used to determine whether trajectories categorized as $trj_{k=1}$ passed through the deep or shallow branch of the BDC ($k = 1d$ or $1s$). These trajectories are

classified as shallow-branch if they cross 400 K but do not reach 30 hPa within 4 years, and as deep-branch if they exceed 30 hPa within 4 years, following the method of Lin et al. (2015). Trajectories were categorized as $trj_k$, according to the first set of 3 continuous days along the trajectory that satisfied the $cri_k$. This resulted in all trajectories being categorized as $k = 1, 2,$ 3, or 4 within 10 years. Trajectories $trj_k$ are assumed to travel along unique paths from origin $k$ to the initial position of the backward trajectory. Origin fractions of air parcels with origin $k$ (hereafter denoted $f_k$) are calculated as a function of

equivalent latitude ($\emptyset_{eq}$), potential temperature ($\theta$), and month ($\mathcal{M}$) of their release. Denoting as $N_k$ the number of trajectories, which are classified into $trj_k$ groups with distinct $\emptyset_{eq}$, $\theta$, and $\mathcal{M}$, the origin fraction for origin $k$ is given by

$$f_k = \frac{\sum_{i=1}^{N_k} \rho_{trj_k ini(i)} * \cos \emptyset_{trj_k ini(i)}}{\sum_{k=1}^{kmax} (\sum_{i=1}^{N_k} \rho_{trj_k ini(i)} * \cos \emptyset_{trj_k ini(i)})}, \tag{1}$$

where $\emptyset_{trj_k ini}$ and $\rho_{trj_k ini}$ indicate the initial latitude and density of the individual backward trajectories, respectively. Note that $\rho_{trj_k ini}$ is calculated from the equation of state. Results of a sensitivity analysis indicate that the estimated origin

fractions are independent of the resolution of the input meteorological data (see Appendix A).

Similar methods are used to estimate the AoA, which is calculated as the average elapsed time until a trajectory goes back to the troposphere where it satisfies whichever criteria $k = 2, 3,$ or 4. Thus, the AoA definition used here differs from that of Hall and Plumb (1994), who defined AoA as the elapsed time an air parcel spends in the stratosphere after across the tropopause. In our estimates of AoA, however, a small fraction of trajectories are still in the stratosphere at the end of the 10-

30 year calculation. Figure 3 shows the percentage of such remaining trajectories estimated as a function of $\emptyset_{eq}$, $\theta$, and $\mathcal{M}$ of their release. The percentages are almost zero in the region where potential vorticity is <4 PVU, whereas they are generally non-zero in the region where potential vorticity is >4 PVU. However, even in this region the values are <2.5 %. Here, we

define $\varepsilon$ as the percentage of trajectories that are still in the stratosphere after the 10-year backward calculation as a function of $\emptyset_{eq}$, $\theta$, and $\mathcal{M}$ (Fig. 3). Then, the AoA ($\Gamma_{Trj}$) is obtained using the elapsed time since each trajectory $trj_k(i)$ left its origin $k$ ($\equiv \tau_k(i)$) according to

$$\Gamma_{Trj} = \sum_{k=2}^{4} \frac{\sum_{i=1}^{N_k} \gamma TT * \tau_k(i) * \rho_{trj_k ini}(i) * \cos \emptyset_{trj_k ini}(i)}{\sum_{i=1}^{N_k} \rho_{trj_k ini}(i) * \cos \emptyset_{trj_k ini}(i)} * (1 - \varepsilon) + \overline{\Gamma_{tail}} * \varepsilon, \tag{2}$$

where $\overline{\Gamma_{tail}}$ is the average AoA for air parcels remaining in the stratosphere longer than the maximum length of the trajectory calculation $tf$ (= 10 years), calculated as follows:

$$\overline{\Gamma_{tail}} = \frac{\int_{\gamma TT * tf}^{\infty} \tau' * PDF(\tau') * \exp\left(-b * (\tau' - \gamma TT * tf)\right) d\tau'}{\int_{\gamma TT * tf}^{\infty} PDF(\tau') * \exp\left(-b * (\tau' - \gamma TT * tf)\right) d\tau'}, \tag{3}$$

$$= \left(\gamma TT * tf + \frac{1}{b}\right), \tag{3'}$$

where PDF is the age probability distribution function or "age spectrum," and $b$ is the exponential decay parameter of the PDF, with its value (b = 0.2038 yr$^{-1}$) from Diallo et al. (2012). The decay parameter in the present analysis may differ from that used by Diallo et al. (2012) because of differences in the vertical trajectory calculations (i.e., kinematic in the present study and diabatic in their work). However, this difference is expected to have little impact on the results because $\varepsilon$ is small, as shown in Figure 3. The term $\gamma TT$ in Eqs (3 and 3') is a correction factor for $\tau_k$ and is required because previous studies (e.g., Inai, 2018) have found that the AoA estimated by trajectory analysis using ERA-Interim data is underestimated, particularly when using a kinematic treatment. Inai (2018) found that this underestimation corresponds to 70 % of the observed value in the mid-latitude stratosphere. To address this underestimation, the AoA values calculated here are corrected by comparing with AoA derived from SF$_6$ mixing ratios ($\Gamma_{SF6}$) assuming a linear trend relative to the time series at Mauna Loa (https://www.esrl.noaa.gov/gmd/obop/mlo/) of 0.33 ppt yr$^{-1}$ (ppt: parts per trillion by mole, with similar definitions for ppm and ppb). When $\gamma TT$ is set at 1.5, $\Gamma_{Trj}$ agrees well with $\Gamma_{SF6}$ (Fig. 4). Thus, a value of 1.5 was used for $\gamma TT$ in this study. The AoA for air masses originating in the stratosphere and those that passed through the deep and shallow branches of the BDC were evaluated using $trj_{k=1}$, $trj_{k=1d}$, and $trj_{k=1s}$, respectively. Note that because this study performs a trajectory analysis using an objective reanalysis dataset, subgrid-scale processes, such as the sporadic injection of tropospheric air masses into the ExUTLS, cannot be explicitly reproduced. Thus, to remove the influence of such events, CONTRAIL data with CO mixing ratios higher than 80 ppb in the region above 340 K and north of 60° N equivalent latitude are not used in this comparison (the same criteria are applied to the comparison shown in Fig. 7 in Sect. 2.2.2).

## 2.2 Air-mass original composition and reconstruction

### 2.2.1 Reconstruction without chemical loss (step 1)

The relative abundance of chemical species in the ExUTLS is strongly affected by changes in the breakdown of transported air masses, reflecting the fact that air-mass chemical composition varies with origin. For example, low-latitude tropospheric air masses have relatively high N$_2$O mixing ratios, whereas high-latitude stratospheric air masses have low mixing ratios,

because $N_2O$ sources and sinks exist in the troposphere and the stratosphere, respectively. This study attempts to reconstruct the spatiotemporal distributions of the chemical species $CH_4$, $N_2O$, $CO$, $SF_6$, and $CO_2$, in the following two steps.

First, the chemically passive tracers (i.e., $SF_6$ and $CO_2$) are reconstructed. According to Inai (2018), if there is no chemical loss for $S$, the mixing ratio of chemical species $S$ in the ExUTLS ($X^S_{NoChem}$) can be reconstructed as a function of $\emptyset_{eq}$, $\theta$, and $\mathcal{M}$ in combination with $f_k$ and the chemical transport from origin $k$ ($X^S_k$):

$$X^S_{NoChem} = \sum_{k=1}^{kmax} f_k * X^S_k. \tag{4}$$

As the time series $X^S_k$ should be treated climatologically for each $k$ and $S$, as required for the origin fraction $f_k$, it is necessary to detrend their values. Therefore, the seasonality and trend of the mixing ratio of $S$ are separately treated in this study. By expressing the detrended mixing ratio of $S$ for an air mass originating in region $k$ ($\equiv X^S_{ORG_k}$) as a function of month ($\equiv \mathcal{M}_{ORG}(i)$) when trajectory $trj_k(i)$ goes back to origin $k$ after advection during $\gamma TT * \tau_k(i)$ and assuming the tropospheric linear trend ($\equiv \lambda^S$), $X^S_{NoChem}$ is calculated as a function of $\emptyset_{eq}$, $\theta$, and $\mathcal{M}$ as follows:

$$X^S_{NoChem} = \sum_{k=2}^4 \frac{\sum_{i=1}^{N_k} \left( X^S_{ORG_k}(\mathcal{M}_{ORG}(i)) - \lambda^S * \gamma TT * \tau_k(i) \right) * \rho_{trj_k ini}(i) * \cos \emptyset_{trj_k ini}(i)}{\sum_{i=1}^{N_k} \rho_{trj_k ini}(i) * \cos \emptyset_{trj_k ini}(i)} * (1 - \varepsilon) + \overline{X^S_{tail}} * \varepsilon, \tag{5}$$

where $\overline{X^S_{tail}}$ is average mixing ratio of $S$ for air parcels remaining in the stratosphere more than $tf$ and is calculated as follows:

$$\overline{X^S_{tail}} = \frac{\int_{\gamma TT * tf}^{\infty} (\chi_{2016} - \lambda^S * \tau\prime) * PDF(\tau') * \exp\left(-b * (\tau' - \gamma TT * tf)\right) d\tau\prime}{\int_{\gamma TT * tf}^{\infty} PDF(\tau\prime) * \exp\left(-b * (\tau' - \gamma TT * tf)\right) d\tau\prime}, \tag{6}$$

$$= \chi^S_{2016} - \lambda^S * \left(\gamma TT * tf + \frac{1}{b}\right). \tag{6'}$$

For $X^S_{ORG_k}$, detrending is applied to the observed values for $k = 2$, 3, and 4 in which a linear trend is determined for each dataset for the period 2012–2016 and all the observed values are normalized to those on January 2016. Monthly aircraft measurement data collected by Tohoku University (TU; Nakazawa et al., 1993; Ishijima et al., 2001; Umezawa et al., 2014) at around 2 km over the Pacific Ocean off the coast of Sendai, Japan are employed for $k = 3$ after taking a 3-month running average. For $k = 2$, an average of the data observed at ~11 km over 0° N–20° N using aircraft flying between Japan and Australia (Matsueda and Inoue, 1996; Matsueda et al., 2002) and the measurement data used for $k = 3$ are used. This averaging is required to account for underestimations of vertical transport from the LT in the trajectory analysis. This averaging procedure is discussed in more detail in Sect. 4.4 together with a caveat for the use of those aircraft measurement data which has somewhat different implication from the following ground-based data. For $k = 4$, ground-based monthly mean data measured by NOAA/ESRL (National Oceanic and Atmospheric Administration/Earth System Research Laboratory) at Summit, Greenland (SUM) and Barrow, Alaska (BRW) are used after averaging the data from the two stations. $N_2O$ and $SF_6$ data at both sites, and $CH_4$ and $CO_2$ data at BRW were continuously measured in situ, whereas other data were obtained using a flask sampling method (Dutton et al., 2017; Thoning et al., 2017; Dlugokencky et al., 2018a, 2018b, 2018c; Petron et al., 2018). These data are distributed by the World Meteorological Organization (WMO) World Data Centre for Greenhouse Gases (WDCGG; https://gaw.kishou.go.jp/). The $\chi^S_{2016}$ in Eqs (6 and 6') is assigned the mixing

ratio of $S$ for the mid-latitude LT ($k = 3$) after annual averaging for 2016. For the trend $\lambda^S$, 9.3 ppb yr$^{-1}$ for CH$_4$, 1.0 ppb yr$^{-1}$ for N$_2$O, 0.33 ppt yr$^{-1}$ for SF$_6$, 2.3 ppm yr$^{-1}$ for CO$_2$, and no trend for CO are assumed by reference to each time series from Mauna Loa (https://www.esrl.noaa.gov/gmd/obop/mlo/).

Figure 5 compares the reconstructions for CH$_4$, N$_2$O, CO, SF$_6$, and CO$_2$ and the CONTRAIL measurements after spatial interpolation to each measurement point for each month. The reconstructions for SF$_6$, and CO$_2$ generally agree with the measurements, with some outliers during the summer season. In particular, some observed CO$_2$ mixing ratios have much smaller values than the reconstructions during boreal summer (Fig. 5e). This might be caused by CO$_2$ absorption by the local Eurasian forest and enhanced subgrid-scale vertical transport (e.g., local convection) during summer. The reconstructions for other seasons, however, generally agree with the CONTRAIL measurements. In contrast to SF$_6$ and CO$_2$, reconstructions for chemically active species (i.e., CH$_4$, N$_2$O, and CO) overestimate the CONTRAIL measurements. Because this overestimation is likely due to chemical loss along their path from the origin region to the ExUTLS, in the next step we perform a reconstruction while taking chemical loss into account.

### 2.2.2 Reconstruction with chemical loss (step 2)

The mixing ratios of chemically active species (CH$_4$, N$_2$O, and CO) are reconstructed using a simple model wherein each chemical loss is simulated along the path from its source region to the ExUTLS. Although each trajectory $trj_k$ has a unique path and transit time from its origin $k$, an "average path" (AP; Schoeberl et al., 2000) can be defined by a cluster of such trajectories. In this study, APs are incorporated into the analysing framework using trajectories binned as a function of $\emptyset_{eq}$, $\theta$, and $\mathcal{M}$. Because both the AP and AoA are defined using the same cluster of trajectories, the two values are considered to be consistent with each other. The relationship between AoA and the chemical loss rate is determined from observation results of Volk et al. (1997), who presented correlations between CH$_4$ and N$_2$O mixing ratios and AoA as well as the gradient of the mixing ratios with respect to AoA (figure 6a of their paper). Using their results, a relationship between the chemical decay and AoA is assumed, as shown in Fig. 6a and b. Note that there are two caveats for this assumption. The first is that a large part of Volk's data was obtained in the Southern Hemisphere. Therefore, they may not be the best representation for chemical decay along the AP from the troposphere into the Northern Hemisphere ExUTLS. The second is that the relationship between AoA and the chemical loss rate is not only determined by the chemical decay along the AP in the stratosphere, but also by the tropospheric trend of tracers that propagate into the stratosphere. However, the trends of CH$_4$ and N$_2$O over the five years before the individual observations in Volk et al. (1997) and in the current study are similar. Therefore, this should not significantly affect the analysis presented here. The gradient of N$_2$O mixing ratio with respect to the AoA grows by $-3$ % yr$^{-1}$ a year, whereas that for CH$_4$ is constant at $-7$ % yr$^{-1}$ when the AoA is <2.5 years and becomes $-11$ % yr$^{-1}$ when the AoA is >3.4 years. Using the assumed chemical decay, the relative abundances of CH$_4$ and N$_2$O are calculated (Fig. 6a and b) and are found to agree well with the observed mixing ratios shown in figure 6a of Volk et al. (1997). The correlation between CO mixing ratio and AoA is not shown in their paper, so here it is assumed as follows. According to Herman et al. (1999), the chemical loss rate of CO is estimated to be 20-times larger than that of CH$_4$ in the

tropical UTLS and it exponentially attenuates with increasing height. Furthermore, the remaining fraction of CO in the stratosphere reaches an equilibrium value because of production processes balancing the chemical loss, which corresponds to ~10 % of the tropospheric value (e.g., Krause et al., 2018). Thus, the chemical decay for CO is assumed to be an e-folding time with respect to AoA ($\tau_{AoA}^{CO}$) that $\tau_{AoA}^{CO} = 0.7 * 2.0^{\Gamma}$, where $\Gamma$ is AoA in years. The corresponding relative abundance of
CO and the gradient with respect to AoA are evaluated as shown in Fig. 6c.

To adapt the correlations between chemical decay and AoA (Fig. 6a–c) to an AP, the chemical decay with respect to AoA is converted to an average loss rate with respect to transit time along an AP ($TT_{AP}$). Figure 6d–f shows the converted loss rates along an AP for the three tracers as well as the corresponding e-folding time. The converted loss rates produce the same relationships between the chemical decay and AoA shown in Fig. 6a–c if each species is reduced during $TT_{AP}$ with the given
e-folding time as a function of $TT_{AP}$. Using these e-folding times ($\equiv \tau_{AP}^{S}$), the mixing ratio of chemically active species $S$ after travelling an AP ($\equiv X^{S}$) is calculated as follows:

$$X^{S} = X_{NoChem}^{S} * \exp(-\frac{\Gamma_{Trj}}{\gamma_{Loss}^{S} * \tau_{AP}^{S}}), \tag{7}$$

where $\gamma_{Loss}^{S}$ is a correction factor for $\tau_{AP}^{S}$ and is determined as follows. Because chemical loss rates might change with the season, we determine a correction factor for each month, such that the reconstruction $X^{S}$ agrees with CONTRAIL
measurements. Scatter plots of CONTRAIL measurements versus reconstructions (Fig. 7) are linear with a slope of 1.0 for each month when the correction factors for $CH_4$ and $N_2O$ are those shown in Fig. 8. Because the scatter plots have large dispersion for CO, instead of the slope, the difference between the CONTRAIL measurements and reconstructions is used for the determination of $\gamma_{Loss}^{S}$ to minimize the difference. Thus, $CH_4$, $N_2O$, CO, $SF_6$, and $CO_2$ in the ExUTLS are reconstructed for a whole year ($X^{S} = X_{NoChem}^{S}$ for $SF_6$ and $CO_2$) and are summarised in Appendix B together with the origin
fractions and AoA. Detailed descriptions of these species are presented in the next section. As in the estimation of $\Gamma_{Trj}$ for stratospheric air masses, the original mixing ratio $S$ of air masses originating in the stratosphere $X_{ORG_{k=1}}^{S}$ is evaluated using only $trj_{k=1}$. The seasonal dependence of $\gamma_{Loss}^{S}$ (i.e., the relative rate of chemical loss) estimated here is discussed in Sects 4.2 and 4.3.

## 3 Results

### 3.1 Origin fraction

Distributions of origin fractions in a $\emptyset_{eq}$–$\theta$ cross-section are shown in Fig. 9 for January together with the climatology of monthly average potential vorticity for the period 2012–2016 obtained from ERA-Interim. In winter, origin fractions of the stratosphere dominate regions north of 40° N and higher than 340 K in altitude. In particular, regions where the altitude and equivalent latitude are greater than 360 K and 50° N, respectively, are almost entirely occupied by stratospheric air masses.
Furthermore, that via the deep branch of the BDC occupies roughly 30 % of the regions where the potential vorticity exceeds ~10 PVU (Fig. 9c). However, origin fractions of the tropical troposphere dominate regions of lower latitude and altitudes

where the potential vorticity is less than ~4 PVU. These origin fractions are >50 %, except in regions lower than 320 K in altitude. Air masses in regions lower than 310 K generally originate in the mid-latitude LT with mixing fractions up to ~70 %, with few air masses originating in the high-latitude LT.

The origin fractions for April are shown in Fig. 10. In spring, origin fractions of the stratosphere are similar to their winter values, and dominate regions north of 40° N and higher than 340 K in altitude. Origin fractions of the stratosphere via the shallow branch of the BDC become slightly smaller than during winter, and those of the deep stratosphere via the deep branch of the BDC increase instead. Tropical tropospheric air masses continue to dominate regions where the potential vorticity is less than ~4 PVU at equivalent latitudes below 50° N, except for regions below 320 K where mid-latitude air masses are present. Origin fractions of the high-latitude LT remain small during spring.

Estimated origin fractions for July are shown in Fig. 11. In summer, origin fractions of the stratosphere become less dominant. In particular, those originating in the deep branch of the BDC (Fig. 11c) are small over the whole ExUTLS. Stratospheric air masses, almost all of which originate in the shallow branch of the BDC, are generally distributed in a small region where the altitude and equivalent latitude are greater than ~370 K and 40° N exceeding 50 % of the origin fraction. In contrast, there is expansion of the region in which the origin fractions of the tropical troposphere are dominant. In particular, nearly 80 % of the air masses in the region above 340 K and south of 40° N originate in the tropical troposphere. Only during this season do origin fractions of air masses originating in the high-latitude LT reach up to ~70 %, but these are limited to a region below ~320 K. Origin fractions of the mid-latitude LT become smaller than during spring, but the region where they are higher than 30 % expands up to 340 K at all equivalent latitudes.

Origin fractions for October are shown in Fig. 12. During autumn, high origin fractions of the stratosphere broaden again in the region above 360 K. However, those originating in the deep branch of the BDC are small. Origin fractions of the high-latitude LT are suppressed, and the region where origin fractions of the tropical troposphere are higher than 50 % becomes larger than during summer and extends up to 80° N along 330–340 K potential temperature surfaces. In the region below 325 K, mid-latitude LT air masses dominate. These seasonal results are compared with previous studies in Sect. 4.1. The robustness and limitations of our estimates are discussed in Sect. 4.4.

## 3.2 Original composition and AoA

As described in Sect. 2.2, detrended mixing ratios of $CH_4$, $N_2O$, CO, $SF_6$, and $CO_2$ observed in the tropical troposphere, mid-latitude LT, and high-latitude LT are assigned to their original mixing ratios for $k = 2$, 3, and 4, respectively. For $k = 1$, the original mixing ratios are estimated by Eqs (5 and 6) using trajectories $trj_{k=1}$ for passive tracers and APs, and Eq. (7) for chemically active species. Figure 13 shows the original mixing ratios of each species assigned to an individual trajectory according to Eq. (5). Note that these values for stratospheric air masses are estimated based on their final state, unlike the case for regions k = 2, 3, and 4, for which the values correspond to their original state. Whereas $CH_4$ and $SF_6$ show seasonal variations and latitudinal gradients in the troposphere, $N_2O$ does not. In contrast to the troposphere, $CH_4$ and $N_2O$ in stratospheric air masses show distinct seasonal variations but somewhat different phase, with a minimum in boreal summer

and maximum in winter for CH$_4$, and a minimum in boreal spring/summer and maximum in autumn/winter for N$_2$O. SF$_6$ mixing ratios are significantly smaller in stratospheric air masses than in the troposphere throughout the year, and show seasonal variations with a maximum in September and minimum in March. A potential reason why the seasonality in the stratosphere differs among CH$_4$, N$_2$O, and SF$_6$ is discussed in Sect. 4.3, together with seasonal variations of $\gamma_{Loss}^{S}$ (Fig. 8).

For CO, there are large seasonal variations in high- and mid-latitude tropospheric air masses, but tropical tropospheric values show smaller seasonal variations. The CO mixing ratios for the stratosphere show little seasonal variability, and are less than ~40 ppb throughout the year. For CO$_2$, seasonal variations are largest in the high-latitude troposphere; mixing ratios in the stratosphere show relatively small seasonal variations, but with a phase that differs from that in the troposphere.

The estimated AoA of stratospheric air masses is shown in Fig. 13f. Stratospheric air masses transported via the deep branch

of the BDC have AoA exceeding 6 years, whereas those transported via the shallow branch have AoA of 1–1.5 years. The average AoA among air masses originating in both branches shows a seasonal variation, with maximum values of ~2.7 years in March and minimum values of ~1.9 years in September, of almost opposite phase to that of SF$_6$ mixing ratios. The relationship between the original composition of stratospheric air masses and their AoA is discussed in Sect. 4.2.

### 3.3 Reconstructions

Chemical distributions reconstructed in the manner described in Sect. 2.2 are shown for January (Fig. 14) together with observation results obtained from CONTRAIL measurements over Siberia and monthly average potential vorticity obtained from the ERA-Interim dataset during the period from 2012 to 2016. Spatial distributions of all chemical species generally show higher mixing ratios with decreasing potential temperature, equivalent latitude, or potential vorticity. Conversely, the distribution of AoA generally shows a higher age with increasing potential temperature, equivalent latitude, or potential

vorticity. In particular, an AoA of greater than 3 years is estimated in the deep ExUTLS for regions higher than 380 K and north of 70° N.

The reconstructions and AoA for April (Fig. 15) show spatial distributions of all species that generally increase with decreasing potential temperature, equivalent latitude, or potential vorticity, as is the case for January. However, the gradients are larger, particularly for CH$_4$ and N$_2$O mixing ratios, such that in regions where the potential vorticity is >6 PVU the

25 mixing ratios are much smaller than those in January, but in regions where the potential vorticity is <4 PVU the mixing ratios are almost the same as in January. The AoA distribution has a structure similar to that shown for January; i.e., age that increases with potential temperature, equivalent latitude, or potential vorticity.

The spatial distributions of the chemical species and AoA change more during summer than during winter and spring (Fig. 16). In particular, all five chemical species show minima at ~350 K north of 60° N equivalent latitude. These minima might

be formed by remainder of the deep stratospheric air masses which were transported during spring. The tracer minima near ~350 K at high equivalent latitudes begin forming in June. This "sandwich" structure in the ExUTLS has been reported by Ploeger and Biner (2016) for summer and by Krause et al. (2018) for spring. In agreement with their studies, the sandwich structures can show evidence for strong poleward transport above ~400 K, leading to mixing ratio minima at lower altitudes.

For $CO_2$, some CONTRAIL measurements show significantly lower mixing ratios than the reconstructed values. The difference between the CONTRAIL measurements and the reconstructions are discussed in Sect. 4.4. The AoA becomes significantly smaller during this season compared with winter and spring. In particular, the AoA of nearly the entire region is <1.6 year with the exception of the region where the tracer minima are formed.

In autumn, the chemical gradients for $CH_4$, $N_2O$, $SF_6$, and $CO_2$ in the ExUTLS are reduced (Fig. 17), in large part because $CH_4$ and $N_2O$ mixing ratios in the deeper ExUTLS increase up to 1750 ppb for $CH_4$ and 315 ppb for $N_2O$. The reconstructed $CO_2$ mixing ratios show a nearly homogeneous distribution in the ExUTLS, leading to a distribution of higher $CO_2$ air masses along the 6–8 PVU potential vorticity surface. The spatial distribution of CO, however, retains a steep gradient, because its chemical lifetime is small (several months). The distribution of AoA during autumn is similar to that during
summer, with the AoA of nearly the entire region with potential vorticity of <8 PVU being less than 1 year.

## 4 Discussion

One goal of the current study is to visualize how seasonal variations in air masses as well as trace gas transport affect the spatiotemporal distributions of chemical species in the ExUTLS. This is accomplished by determining the seasonal characteristics of origin fractions of ExUTLS air masses originating in each region $k$ at fixed points with those of the
15 reconstructions for each species, and comparing the distribution of each species in the ExUTLS with the original mixing fraction in each origin region. We next discuss the results of this analysis and some implications revealed through the reconstructing procedures, together with the limitations of the current study.

### 4.1 Seasonal variations in origin fractions and reconstructions at fixed locations

To identify the characteristics of seasonal variations in origin fractions and reconstructions at fixed locations, four regions
are selected: mid-equivalent latitude upper (MU) ExUTLS (45° N, 370 K), high-equivalent latitude upper (HU) ExUTLS (75° N, 370 K), mid-equivalent latitude lower (ML) ExUTLS (45° N, 320 K), and high-equivalent latitude lower (HL) ExUTLS (75° N, 320 K). Figure 18 shows seasonal variations in the origin fractions of each origin evaluated at the four locations. In the MU ExUTLS, origin fractions of the tropical troposphere become high, exceeding 50 % during summer and autumn. Accompanying this increase, trajectories originating in the tropical troposphere over around Asia are strengthened.
In the other seasons, origin fractions of the stratosphere dominate. In particular, those that travelled via the shallow branch of the BDC exceed 50 %. The origin fractions of the mid- and high-latitude LT are nearly zero throughout the year, with the exception of that for the mid-latitude LT in autumn. In the HU ExUTLS, origin fractions of the stratosphere dominate and exceed 60 % throughout the year. Furthermore, origin fractions of air masses that travelled via the deep branch of the BDC exceed 20 % during the period from January to April, whereas tropical tropospheric air masses generally fail to reach this
region during this period. In the ML ExUTLS, tropospheric origin fractions are dominant. In particular, those of the mid-latitude troposphere exceed 50 % during summer and those of the high-latitude troposphere exceed 20 % during July and

August. During winter and spring, however, tropical tropospheric air masses dominate. In the HL ExUTLS, origin fractions of the mid- and high-latitude LT are enhanced during summer. Origin fractions of the high-latitude LT are comparable to those in the ML ExUTLS, but smaller than those of the mid-latitude LT in the HL ExUTLS. This can be explained by enhanced exchange at the bottom edge of the subtropical jet (i.e., along the 320–330 K surface for summer, e.g., Gettelman et al., 2011). As shown in Fig. 11d, enhanced origin fractions of the mid-latitude LT are distributed along such isentropes. In winter, origin fractions of the tropical troposphere and stratosphere are roughly 50 % and 40 %, respectively.

In addition to seasonal variations in origin fractions, seasonal variations in the tracer mixing ratios in origin regions (Fig. 13) also affect chemical distributions in the ExUTLS. Figure 19 reveals that seasonal variations in the reconstructions for each species and the trajectory-estimated AoA in each of the four locations have patterns that differ because they are based on a superposition of the origin fractions shown in Fig. 18 with the original time series for $k = 1$–$4$ of the individual tracers shown in Fig. 13. Note that the CONTRAIL data are plotted if the measurement was conducted within ±5° in equivalent latitude and within ±5 K in potential temperature of one of the four locations. This results in few plotted CONTRAIL observations in the ML and HL ExUTLS regions during summer, and no observations in MU and HU ExUTLS regions from June to January. This is caused by the seasonality of the thermal and dynamical structures of the ExUTLS and fixed flight altitudes. Despite the sparse and non-uniform observational field, the spatiotemporal distributions of chemical species, together with the origin fractions of the original air masses, can be resolved. This ability is one of the important advantages of the current analysis. The mixing ratios of $CH_4$ and $N_2O$ show modest seasonal variations in the lower ExUTLS, whereas they show large seasonal variations in the upper ExUTLS, with minima in spring and maxima in autumn. The minima in spring are due to the transport of stratospheric air masses via the deep branch of the BDC, which have low $CH_4$ and $N_2O$ mixing ratios and also low AoA. This seasonal variation in chemical abundance for stratospheric air masses is discussed further in the next section. In contrast to $CH_4$ and $N_2O$, CO mixing ratios show smaller seasonal variations in the upper ExUTLS than in the lower ExUTLS, with the exception of the high mixing ratio in the upper ExUTLS in August. This can be explained by the transport of mid-latitude LT air masses, which have higher CO mixing ratios than the other air masses, to the lower ExUTLS during summer. In addition, a large fraction of air masses reach the upper ExUTLS only during August. The seasonal characteristics of $SF_6$ mixing ratios are similar to those of $CH_4$ and $N_2O$. The phase of seasonal variations in the upper ExUTLS is nearly synchronized with, but slightly precedes, those of $CH_4$ and $N_2O$, and more closely resembles the upside-down pattern of AoA variations (Fig. 19f). The phase of seasonal variations of $CO_2$ mixing ratios in the lower ExUTLS is nearly synchronized between the ML and HL ExUTLS, with the largest amplitude being evident in the ML ExUTLS. The phase of $CO_2$ variations in the upper ExUTLS is quite different from that in the lower ExUTLS, with maxima during summer/autumn. This seasonal variation in the upper ExUTLS is consistent with observational estimates by Hoor et al. (2004) and Strahan et al. (2007).

Seasonal variations in AoA evaluated at the four locations are shown in Fig. 19f. The phase of seasonal variations for the four locations is roughly synchronized, whereas the absolute values are clearly different. For example, AoA in the HU ExUTLS has a maximum of >2.5 years during spring and a minimum of ~1.3 years during the end of summer, whereas in

the ML ExUTLS the maximum is only ~0.5 years and occurs during the period from winter to spring. The amplitude of AoA variations in the ExUTLS is likely related to air-mass mixing from the stratosphere, particularly when this involves air masses that have been transported via the deep branch of the BDC. This point is discussed further in the next section, in relation to seasonal variations in chemical composition.

## 4.2 Original compositions and mixing effects

As discussed in the previous section, the distributions of CO and $CO_2$ in the ExUTLS are strongly affected by tropospheric air masses because CO has a short chemical lifetime and $CO_2$ shows large seasonal variations in the high- and mid-latitude LT. For $CH_4$, $N_2O$, and $SF_6$, however, seasonal variations in origin fractions of the stratospheric air masses and in the compositions of the original air masses are considered to be essential factors in their spatiotemporal distributions in the ExUTLS. Here, we discuss seasonal variations in the composition of stratospheric air masses and how this affects chemical distributions via mixing with tropospheric air masses in the ExUTLS. Figure 20 shows the relationships between chemical abundances from CONTRAIL measurements and the AoA estimated from the trajectories and interpolated to each CONTRAIL measurement location, along with these relationships for each original air mass. The AoA for stratospheric air masses are the same as those shown in Fig. 13f, whereas the AoA for the tropical, mid- and high-latitude troposphere are set to zero. Thus, the denotations for the tropospheric air masses only move vertically in the cross-sections according to their seasonal variations. Overall, the CONTRAIL measurements are roughly distributed on lines connecting the tropospheric and stratospheric air masses for all seasons and chemical compositions. This linear distribution suggests that dynamical mixing of tropospheric with stratospheric air masses shapes the chemical distributions of the ExUTLS. Such linear "mixing lines" also suggest that the mixing took place rapidly (i.e., at a time-scale shorter than their chemical lifetimes) along an isentropic surface (Plumb, 2007 and references therein). A comparison of the distribution of CONTRAIL measurements with trends in the troposphere for $SF_6$ shows that the CONTRAIL measurements are distributed along the lines of the sign-reversed trend. According to Engel et al. (2002) and Bönisch et al. (2009), the mixing ratios of $CO_2$ and AoA do not correlate below a level of ~3 years AoA because the propagated signal of the tropospheric seasonal cycle into the stratosphere is still detectable. In agreement with their results, the CONTRAIL $CO_2$ measurements also converge to the sign-reversed trend with increasing AoA. However, for $CH_4$ and $N_2O$, measurements depart from the sign-reversed trends toward lower mixing ratios with increasing AoA. This deflection can be interpreted as being due to their stratospheric sinks; i.e., chemical destruction of $CH_4$ and $N_2O$ in the stratosphere, with no such destruction of $SF_6$ and $CO_2$.

Both the AoA and chemical abundance of the original air masses from the stratosphere show seasonal variations that might be caused by seasonal variations in mass fluxes from the deep and shallow branches of the BDC. Figure 20f shows seasonal variations in AoA and the value that is calculated by integration of "age spectrum" (PDF) from 0 to $tf$ for air masses originating in the stratosphere as well as those separately evaluated for air masses that have travelled via the deep and shallow branches of the BDC. As the PDF is calculated with a weighting factor according to area and density, as in Eq. (1), their integrations reveal relative masses. Air masses originating in both the shallow and deep branches have minima in

September and maxima in March. These in-phase seasonal variations enhance both the seasonal variations of the total origin fractions of the stratosphere and its average AoA.

Interesting cyclic structures appear in $CH_4$ and $N_2O$ mixing ratios and their AoAs in stratospheric air masses. For example, the $CH_4$ mixing ratio is ~1750 ppb (AoA of ~2.3 years) in winter, ~1700 ppb (AoA of ~2.6 years) in spring, ~1650 ppb (AoA of ~2.3 years) in summer, and again ~1700 ppb (AoA of ~2.0 years) in autumn. Thus, clockwise rotations are the result of this pattern. The same is true for stratospheric $N_2O$ and AoA. These rotations are formed by seasonal variations in AoA that are at a maximum in spring and a minimum in autumn, in combination with seasonal variations in the relative chemical loss rate along the AP (defined as $\gamma_{Loss}^{S}$ and discussed in Sect. 2.2.2) that is at a maximum in winter and a minimum in summer. These $\pi/2$ phase-lagged seasonal variations result in rolling variations in the relationship between $CH_4$ and $N_2O$ mixing ratios and AoA in stratospheric air masses. The seasonal variation in AoA is determined by the mixing of stratospheric air masses via the deep branch of the BDC (Fig. 20f). Although the detailed mechanism driving the seasonality of the chemical loss rate along AP is unknown, it likely involves the seasonal change of the relationship between AP and AoA as a possible mechanism from a dynamical viewpoint. Other candidate mechanisms from a chemical viewpoint are seasonal changes in the abundance of disrupting substance along the AP, or seasonal changes in the solar radiation intensity and sunlit time. Further discussion of this topic is included in the next section, together with the mechanism driving the $\pi/2$ phase-lagged, i.e., rolling relationship between $CH_4$ and $N_2O$ mixing ratios and AoA in stratospheric air masses.

The abundance of $N_2O$ and $CH_4$ in stratospheric air mass may be related to the fraction of air masses travelling via the deep and shallow branches. The relationship between the chemical abundance and mass fraction of the two branches is now considered. The current study estimates approximately 24 % and 14 % of air masses following the deep branch are of stratospheric origin in spring and autumn, respectively, and the AoA is estimated to be ~6.4 years (Fig. 20f). Andrews et al. (2001) estimated the $N_2O$ mixing ratio in the mid-latitude deep stratosphere to be ~80 ppb and <40 ppb where the AoA is estimated from $CO2$ mixing ratio to be 5.5 years and 6.0 years, respectively. As their estimates are normalized to 1997 tropospheric values, the quantitative difference in the baseline $N_2O$ mixing ratios may differ by ~20 ppb from the present values. If we assume the $N_2O$ mixing ratio of air masses originating in the deep branch of the BDC is 60 ppb, and that air masses are mixed at ratios of 24 % and 14 % with air masses whose $N_2O$ mixing ratio is 330 ppb, such mixing leads to ~265 ppb and ~290 ppb $N_2O$ mixing ratios, respectively. These values are up to ~20 ppb lower than the $N_2O$ mixing ratios of ~280 ppb in May and ~310 ppb in November estimated for the original stratospheric air masses shown in Figs 13b and 20b. The same arguments are valid for $CH_4$ with respect to the relationship between stratospheric $CH_4$ mixing ratios and AoA; i.e., $CH_4$ mixing ratios are <600 ppb in regions where the AoA is >5.5 years, as estimated by Röckmann et al. (2011). These overestimations of $N_2O$ and $CH_4$ mixing ratios for the original stratospheric air masses might be due to overestimation of the AoA. This possibility is discussed further in the next section.

## 4.3 Rolling relationship between CH4/N2O and AoA in stratospheric air masses

To examine the mechanism that drives the rolling relationship between $CH_4$ and $N_2O$ mixing ratios and AoA in stratospheric air masses (Fig. 20a and b), separately estimated PDFs for air masses transported from individual origins are considered. Figure 21 shows an example of PDFs estimated for January. Each PDF has a spectral peak corresponding to the most probable transit time (modal time). For January, the modal times for high-latitude LT, mid-latitude LT, and tropical tropospheric air masses are <0.2 year, whereas that for stratospheric air masses is 1.0 year. The modal times for stratospheric air masses for a whole year are summarised in Table 2. These demonstrate seasonality with a maximum in winter, but remain at ~0.6 year during other seasons.

The PDF for stratospheric air masses also has a long exponentially decaying tail, which leads to a longer mean age. For example, the mean age is calculated to be 2.5 years (i.e., 2.5 times larger than the modal time) for January (Fig. 21). If we consider chemically passive species with linear trends in the troposphere, the relative abundance depends only on the AoA and they should be linearly correlated with each other. Therefore, the mean age corresponds to the average mixing ratio. This can be confirmed by seasonal variations with linear correlations between $SF_6$ and $CO_2$ mixing ratios and AoA in stratospheric air masses (Fig. 20d and e). However, for chemically active species, in particular $CH_4$ and $N_2O$, their abundance in stratospheric air masses as well as the ExUTLS is controlled primarily by chemical loss processes, as seen in the comparison in Figs 5 and 7. The chemical loss rate changes with season. Thus, the relationship between the relative abundance and AoA changes seasonally. The most influential air masses on the seasonal variation of average mixing ratios should be those that traveled over the modal time rather than the mean AoA, because of the larger PDF. In other words, the seasonality of the average mixing ratio of chemically active species is most sensitive to seasonal variations in the air masses that have been transported along a "modal path (MP)" that corresponds to the modal time. This provides an explanation of the rolling relationship between $CH_4$ and $N_2O$ mixing ratios and AoA in stratospheric air masses, as described in the next paragraph.

If we assume the transit time to be 1.0 year in winter and 0.6 years during other seasons, according to the modal time (Table 2), and if the typical season when stratospheric air masses are affected by chemical losses is assumed to be the middle of the transit time, the chemical loss processes of $CH_4$ and $N_2O$ are primarily affected 0.5 year and 0.3 years prior to ending up in the ExUTLS during winter and other seasons, respectively. The higher rates of chemical loss during May–August estimated for $CH_4$ and $N_2O$ in Fig. 8 are caused primarily by chemical processes at the midpoint of a MP during January–April, when from the aspect of the seasonal variation of the BDC, the MP is expected to extend deeper stratosphere. The slight phase difference between $CH_4$ and $N_2O$ in stratospheric air masses might reflect differences in their chemical loss mechanisms. The chemical loss of $CH_4$ is controlled by reactions with OH, $O(^1D)$, and Cl, whereas that of $N_2O$ is controlled primary by photolysis and secondarily by reactions with $O(^1D)$. Therefore, the seasonality of $CH_4$ is affected not only by seasonal variations of solar radiation that is primary and direct factor for $N_2O$ loss, but also by OH abundance along a MP. Thus, the seasonal variations of $CH_4$ and $N_2O$ mixing ratios in stratospheric air masses (Fig. 13a and b) leading to the rolling

relationship with AoA (Fig. 20a and b) are interpreted as a combination of seasonally varying chemical loss rates on a transport time-scale near the modal time and a path close to that of the MP.

As discussed above, a better approach might be to first model the chemical loss for active species based on the modal time and MP, and evaluate the mixing ratio in stratospheric air masses. Then, the distribution can be reconstructed using the

origin fraction of stratospheric air masses. However, the PDF might change depending on the $\phi_{eq}$ and $\theta$ of the trajectory releasing point, and it is difficult to obtain adequate estimates of the PDF without a sufficient number of trajectories for all bins. Such an approach will be the focus of future work. The use of modal time can result in smaller values for the correction factors for e-folding times (Fig. 8) because of smaller values for $\Gamma_{Trj}$ in Eq. (7). Such an adjustment will affect the correction factors for e-folding times, however, it will not significantly affect results presented here, particularly those related to the

reconstructed distributions of the five trace gases.

## 4.4 Limitations of the current study

This study provides a detailed explanation of seasonal variations in chemical distributions and transport in the ExUTLS from a dynamical standpoint using trajectory analysis in combination with aircraft measurements. Results suggest that the spatiotemporal distributions of $CH_4$, $N_2O$, $SF_6$, and AoA in the ExUTLS are controlled primarily by air-mass transport via

the deep and shallow branches of the BDC and by their mixing with tropospheric air masses in the ExUTLS, whereas those of CO and $CO_2$ are controlled largely by tropospheric air masses, because CO has a short chemical lifetime and $CO_2$ shows large seasonal variations in the mid-latitude LT. However, some assumptions and limitations of the current study should be mentioned.

First, some uncertainty results from the use of ERA-Interim data in trajectory analyses. Trajectory results generally depend

on the resolution of the input data. We performed sensitivity analyses to clarify this dependency in our origin fraction estimates (Appendix A). Results confirm that our estimates are independent of the resolution of the ERA-Interim data, at least as they relate to statistical characteristics. Furthermore, it is known that AoA calculated from trajectory analyses using ERA-Interim data are somewhat young-biased. For example, these estimated AoA values are ~30 % younger than those estimated from balloon-borne observations in the middle stratosphere, as demonstrated by Inai (2018). To address this issue,

trajectory-based AoA values are uniformly corrected by a correction factor of 1.5 (determined with reference to the AoA obtained from $SF_6$ mixing ratios) in this study. There is, however, a possibility that the bias differs with the meteorological region, because different mechanisms drive the shallow and deep branches of the BDC (e.g., Birner and Bönisch, 2011). This is a possible cause of the inconsistent relationship between the abundance of $N_2O$ and $CH_4$ in stratospheric air mass and the mass fraction of the air masses travelling via the deep and shallow branches of the BDC. If the AoA of air masses travelling

via the deep branch is assumed to be ~5 years, the $N_2O$-AoA and $CH_4$-AoA relationships approach those of Andrews et al. (2001) and Röckmann et al. (2011), respectively. Trajectory results also generally depend on the vertical condition, i.e., kinematic (employed by the current study) or diabatic (employed by, for example, Diallo et al., 2017). Previous studies suggest that using kinematic trajectories leads to a stronger dispersion and somewhat young bias in AoA estimates compared

with using diabatic trajectories (e.g., Schoeberl et al., 2003; Diallo et al., 2012). Therefore, using diabatic trajectories in this analysis might result in a correction factor ($\gamma TT$) of <1.5.

The second limitation is related to the criteria for the determination of air mass origin. These criteria may strongly affect origin fraction estimates and are thus expected to contribute to the uncertainty of this analysis, to some degree. A comprehensive sensitivity test to address this issue, focusing on in-mixing in the TTL, has been reported by Inai (2018), who found that the mixing fraction can vary by 40 % to 180 %, depending on the choice of criteria. Though the same test could be applied to the current study, the estimated origin fraction distributions are comparable to those estimated based on trace gas observations by the In-service Aircraft for a Global Observing System-Civil Aircraft for the Regular Investigation of the atmosphere Based on an Instrument Container (IAGOS-CARIBIC; Umezawa et al., 2015). Moreover, these estimates are indirectly validated by the CONTRAIL observations, through the reconstruction of the chemical distributions (as evident in Figs 5d and e, and 7). This agreement supports our criteria selection and suggests that our estimated origin fractions are not, at least, grossly wrong. However, the breakdown of stratospheric air masses is subject to the limitations described in the last part of Sect. 4.2. If the relative fraction of air masses travelling via the deep branch is 7 % smaller than the estimated values (i.e., if they were 17 % and 7 % in spring and autumn), the relationship between the abundance of $N_2O$ and $CH_4$ in stratospheric air mass and the mass fraction of the air masses travelling via the deep and shallow branches also approaches that of Andrews et al. (2001) and Röckmann et al. (2011).

Another limitation may arise from the analysis methodology. The observed mixing ratios of $CH_4$, $N_2O$, CO, $SF_6$, and $CO_2$ are used after removing linear trends for each time-series, and are considered to be a function of month and treated separately from the long-term trend. This treatment decreases the precision of observations if the observed values have non-linear interannual variations, which is mainly concerned for $CO_2$. Furthermore, the CONTRAIL measurements were conducted once a month. Thus, one observed value represents atmospheric conditions at a specific spatiotemporal point, whereas the analysis field has a coarser spatiotemporal resolution, corresponding to, at minimum, that of the grid scale of the ERA-Interim dataset. Such a mismatch in spatiotemporal resolution may contribute to the lack of agreement between the reconstructions and CONTRAIL measurements during summer, particularly for $CO_2$ (Fig. 5e). However, uncertainties arising from these issues are minimized by the use of equivalent latitude and potential temperature, which are dynamically conserved quantities in the stratosphere. In the troposphere, which is more unstable, potential temperature and potential vorticity are not conserved, or are conserved only on much short timescales, because of diabatic motion. It should be noted that tracer uplift from the LT into the UT during summer (particularly for $CO_2$, as discussed above) cannot be reduced with the coordinate system employed here. Though the current study covers only the ExUTLS over a longitudinal range from 0° E to 140° E for comparison with the CONTRAIL measurements, the origin fractions and reconstructions are trial-evaluated over North America (Appendix A). Results confirm that the origin fractions are consistent between the two regions, and thus support the robustness of the current study. In this study, linear trends for $CH_4$, $N_2O$, $SF_6$, and $CO_2$ are assumed for the reconstruction. Although this is a simplified treatment, given the length of the analysis period, these trends are roughly constant over this time period with the exception of $CH_4$, and the $CH_4$ reconstructions are more strongly affected by

chemical loss, as is evident in a comparison of Figs 5a and 8a. In the reconstruction procedure described in Sect. 2.2.1, it was necessary to assign the average of tropical aircraft and mid-latitude LT measurements to the original values for $k = 2$ to prevent underestimations. As discussed in Sect. 2.2.1, one cause of this underestimation might be subgrid-scale tropospheric upward transport that, although common, cannot be accounted for in the trajectory analysis.

For the aircraft measurements data used as original mixing ratios for air masses originating in the tropical troposphere and mid-latitude LT, particularly, those collected by TU over sea close to Japan may contain a mixture of polluted and unpolluted air masses in some degree. On this point, they have different implication from measurement data obtained by background monitoring sites which is employed as that for the high-latitude LT. For example, the background CO and $SF_6$ mixing ratios in mid-latitude are comparable to those in high-latitude troposphere (as confirmed, e.g., in NOAA/ESRL web
sites;                                                 https://www.esrl.noaa.gov/gmd/ccgg/globalview/co/co_intro.html, https://www.esrl.noaa.gov/gmd/hats/combined/SF6.html), whereas those used for the mid-latitude LT are significantly larger than those for high-latitude LT, except during winter. In real atmosphere, the tracer distribution in the ExUTLS is determined not only by influx of background air masses, but also by that of polluted air masses. Therefore, it must take such polluted air masses into account to reconstruct plausible distribution of trace gases, i.e., CO and $SF_6$, in the ExUTLS. Since
their artificial sources are mostly distributed in mid-latitude LT, therefore we might have been able to reconstruct the tracer distribution which agrees well with the CONTRAIL measurements. A more proper approach would be to assign background values with the addition of incremental values due to pollution assumed in each latitude region, such an approach will be the focus of our future work.

## 5 Summary

To identify the origin of air masses in the ExUTLS, kinematic backward trajectories were calculated for 10 years following the method of Inai (2018) using ECMWF ERA-Interim data as input. The analysis period extends from January 2012 to December 2016, and trajectories were categorized by origins in the stratosphere, tropical troposphere, mid-latitude LT, and high-latitude LT based on meteorological parameters along each individual trajectory. The origin fractions of air masses originating in each region were estimated as a function of equivalent latitude, potential temperature, and month. Furthermore,
using the same trajectory, the mixing fractions of air masses originating via the shallow and deep branches of the BDC were separately estimated along with the AoA.

The origin fractions show obvious seasonal variations. In the mid-equivalent latitude upper ExUTLS, origin fractions of the tropical troposphere exceed 50 % during boreal summer and autumn, whereas origin fractions of the stratosphere via the shallow branch of the BDC are dominant during winter and spring. In the high-equivalent latitude upper ExUTLS, origin
fractions of the stratosphere exceed 60 % throughout the year. In the mid- and high-equivalent latitude lower ExUTLS, origin fractions of the mid- and high-latitude troposphere are large during summer, whereas during winter, origin fractions of the tropical troposphere are dominant.

By incorporating the time-series of mixing ratios for several chemical species obtained from ground-based and air-borne observations into the estimated trajectories, the spatiotemporal distributions of the chemical species $CH_4$, $N_2O$, CO, $SF_6$, and $CO_2$ in the ExUTLS were reconstructed, along with estimations of the chemical decay during advection for $CH_4$, $N_2O$, and CO. The reconstructions are calculated to agree with CONTRAIL measurements in the ExUTLS. Furthermore, uniform spatiotemporal species distributions are obtained for the ExUTLS from non-uniform observations. The origin fractions and AoA of each reconstruction are discussed. Distributions of $SF_6$ and $CO_2$ in the ExUTLS are linearly correlated with that of AoA because of their chemically passive behavior and quasi-stable increasing trends in the troposphere. Distributions of $CH_4$, $N_2O$, and CO are controlled primarily by chemical decay along the transport path from the source region via the stratosphere and subsequent mixing such stratospheric air masses with tropospheric air masses in the ExUTLS. This interpretation is consistent with estimated transport time-scale and the aspect of the seasonal variation of the BDC.

This study developed and demonstrated a unique and effective method to exploit the advantages of observational data in combination with trajectory analysis. This method provides a means to understand both air-mass transport and chemical decay from a new perspective. Furthermore, this technique can be applied to other data (e.g., species isotope ratios) or analyses of regions where trajectory calculations are effective.

**Appendix A. Sensitivity analyses**

It is well known that results from trajectory analyses are affected by the resolution of input meteorological data. For example, Inai (2018) suggests that the origin fraction of stratospheric air masses in the upper TTL can vary by ~50 % in magnitude. Here, the sensitivity of our results to data resolution is tested. Figure A1 shows the dependence of origin fractions on the resolution of meteorological data for trajectories launched each month. Note that here, the trajectory calculation length is limited to 90 days due to limited computing resources. Origin fractions calculated from ERA-Interim data and used in this study ($1.5° \times 1.5°$ horizontal resolution, 37 vertical levels) are compared with those using a finer resolution ($0.75° \times 0.75°$ horizontal resolution, 60 vertical levels). Origin fractions were evaluated for each bin set in an equivalent latitude–potential temperature cross-section (crosses). Results confirm that these points are distributed in a linear fashion with slopes of around 1.0 regardless of season. This suggests that the origin fractions are not quantitatively or qualitatively dependent on the resolution of the input data. This independence differs from the findings of Inai (2018), possibly because transport mechanisms in the ExUTLS are related to synoptic-scale mechanisms rather than convective activity, which dominates the tropical region.

In the current study, origin fractions were estimated only for the longitudinal region between 0° E and 140° E, selected for comparison with CONTRAIL measurements over Siberia. Previous studies have investigated mixing processes between tropospheric and stratospheric air masses over different longitudinal regions; e.g., over North America (Pan et al., 2010). To compare our results with these studies, the dependence of origin fraction on longitudinal region was tested. Figure A2 compares origin fractions evaluated over Siberia and North America. Results confirm that the data points in Fig. A2

(crosses) are distributed in a linear fashion with slopes of around 1.0 regardless of season. This suggests that the origin fractions are not quantitatively and qualitatively dependent on longitudinal region. This independence may arise from the employment of equivalent latitude and potential temperature, which are dynamically conserved parameters, in this analysis.

**Appendix B. Large-scale perspective of origin fractions and reconstructions**

Detailed analyses of origin fractions and reconstructions for specific months are informative, but taking a larger perspective might provide insight into seasonal transport processes and tracer distributions in the ExUTLS. Here, we present a larger-scale perspective of origin fractions and reconstructions for the ExUTLS. Figure B1 shows monthly origin fractions as a function of time of year. The axes of each panel are as in Figs 9–12. Thus, the seasonal behavior of air-mass transport into the ExUTLS from surrounding areas is visualized. Origin fractions of the stratosphere via both branches of the BDC increase
from winter to spring. Subsequently, those via the deep branch become small during summer and autumn. In contrast, origin fractions of the tropical troposphere are prominent during summer and autumn, with the exception of regions of lower potential temperature. The lower ExUTLS is dominated by air masses originating in the mid-latitude LT throughout the year, but those originating in the high-latitude LT contribute to this lowermost region during summer. The seasonal behavior of reconstructed chemical species is shown in Fig. B2. The patterns of chemically passive tracers, particularly $SF_6$, follow that
of AoA. However, $CO_2$ in the lower ExUTLS undergoes different seasonal variations. (Note that the $CO_2$ mixing ratio is likely not well-reconstructed during summer in the lower ExUTLS.) The patterns of $CH_4$ and $N_2O$ are similar in that the mixing ratios in the deep ExUTLS become small during spring and summer. However, their seasonal transitions differ slightly from each other, with that of $CH_4$ varying more gradually than that of $N_2O$. The mixing ratios of CO in the deep ExUTLS are small throughout the year, but increase slightly during autumn.

**Author contribution**

Yoichi Inai designed and carried out the study. Toshinobu Machida, Hidekazu Matsueda, Yousuke Sawa, Kazuhiro Tsuboi, and Keiichi Katsumata obtained the measurement data, and Shinji Morimoto, Shuji Aoki, and Takakiyo Nakazawa developed the measurement system. Yoichi Inai and Ryo Fujita prepared the manuscript with contributions from all co-authors.

**Acknowledgements**

The authors would like to acknowledge the support of many engineers from Japan Airlines and JAMCO Tokyo. All trace gas mixing ratio data at ground-based sites were provided by NOAA/ESRL (National Oceanic and Atmospheric Administration/Earth System Research Laboratory) and were downloaded from the WMO/WDCGG website
(https://gaw.kishou.go.jp/). This work was supported by Grants-in-Aid for Scientific Research (18K03738 and 26220101)

from the Japan Society for the Promotion of Science and the Arctic Challenge for Sustainability (ArCS) Project by the Ministry of Education, Culture, Sports, Science and Technology, Japan. We thank Masashi Kohma for valuable discussions. We also thank ECMWF for providing the ERA-Interim data. All figures were produced using the GFD-DENNOU Library. The authors sincerely appreciate the constructive comments of two reviewers that greatly improved the manuscript.

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

**Table 1: Criteria for determining air mass origin. Each trajectory is categorized once it continuously satisfies one set of criteria $k$ = 1, 2, 3, or 4 during three continuous days along its path.**

| Category # | Origin | Criteria |
|---|---|---|
| $k = 1d$ | Deep stratosphere | Pot. temperature >400 K; P <30 hPa within 4 years |
| $k = 1s$ | Shallow stratosphere | Pot. temperature >400 K; not satisfied $k = 1d$ |
| $k = 2$ | Tropical troposphere | Pot. temperature <350 K; lat. <20° N; pot. vorticity <1 PVU |
| $k = 3$ | Mid-latitude LT | Z <4 km; 20° N < lat. <45° N |
| $k = 4$ | High-latitude LT | Z <4 km; lat. >45° N |

**Table 2: Modal time of air masses originating in the stratosphere. Note that there are two peaks in the PDF (0.5 and 1.0 year) for February, the average is listed below.**

| Month | Jan | Feb | Mar | Apr | May | Jun | Jul | Aug | Sep | Oct | Nov | Dec |
|---|---|---|---|---|---|---|---|---|---|---|---|---|
| Modal time [year] | 1.0 | 0.8 | 0.6 | 0.6 | 0.6 | 0.7 | 0.6 | 0.6 | 0.6 | 0.6 | 0.8 | 0.9 |

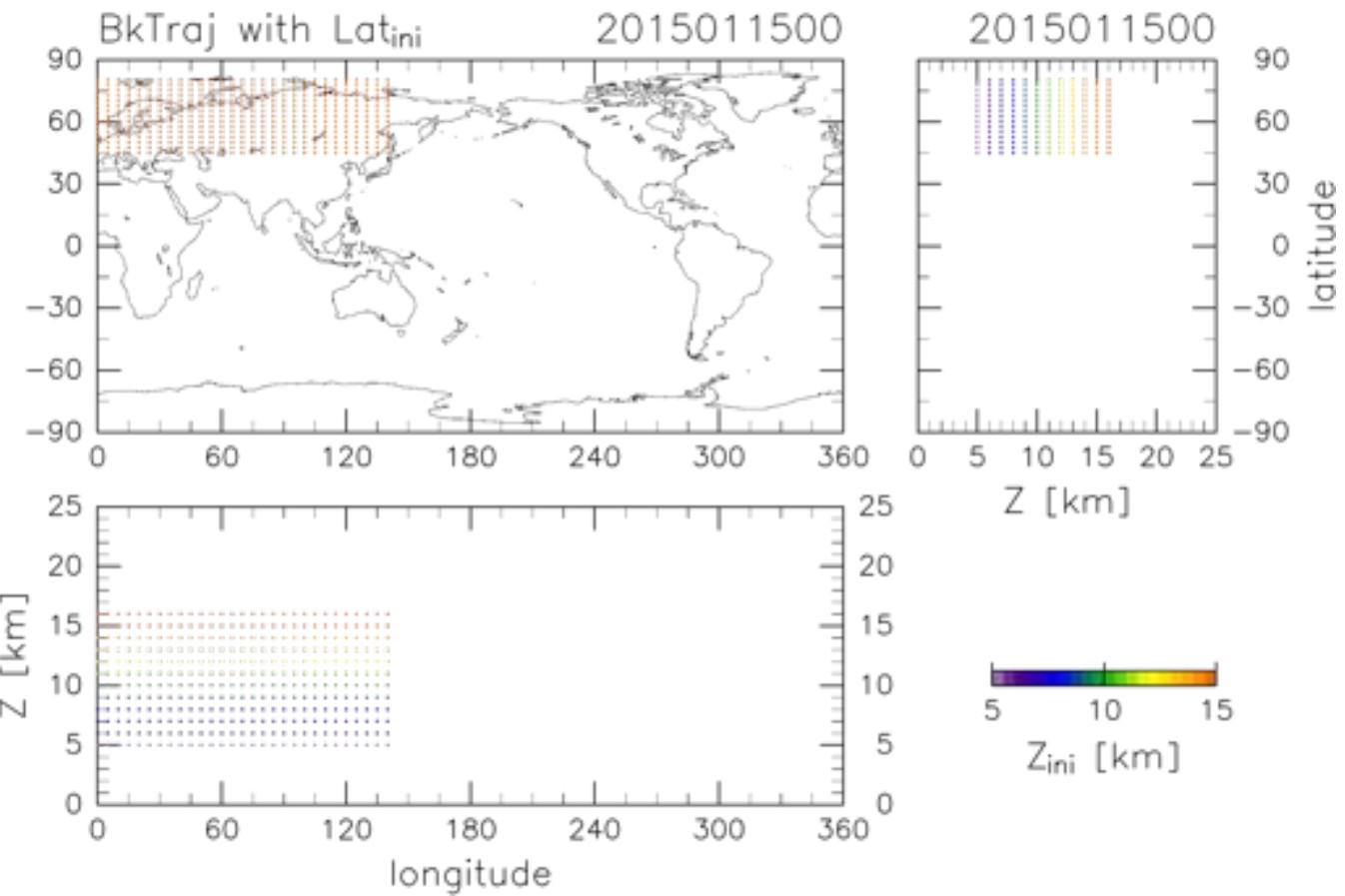

**Figure 1: Initial trajectory positions projected in (top-left) longitude–latitude, (top-right) height–latitude, and (bottom) longitude–height sections. Colours indicate the initial height for each position.**

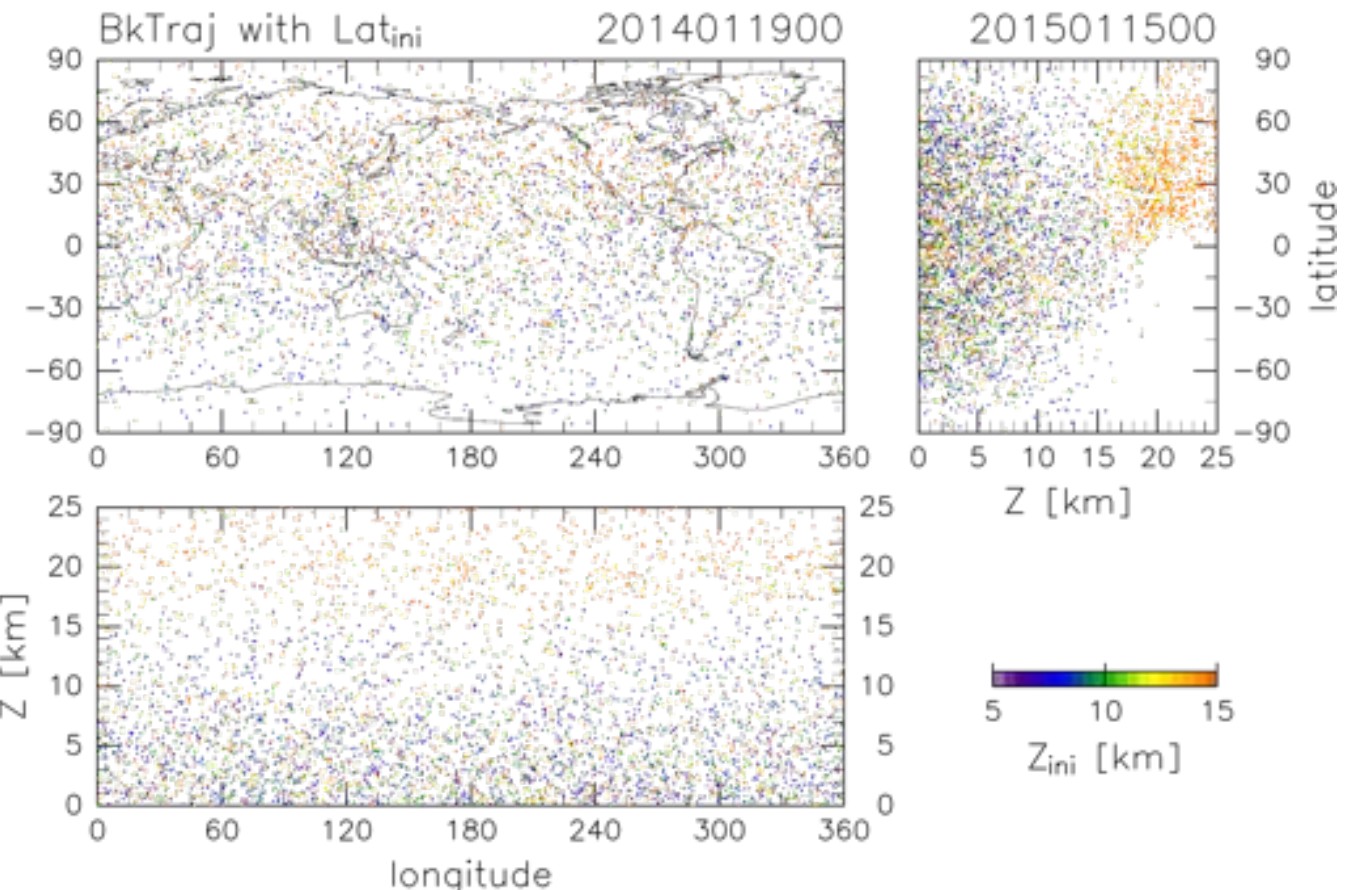

**Figure 2: As in Fig. 1, but for the terminal positions of trajectories after calculating backward for 361 days.**

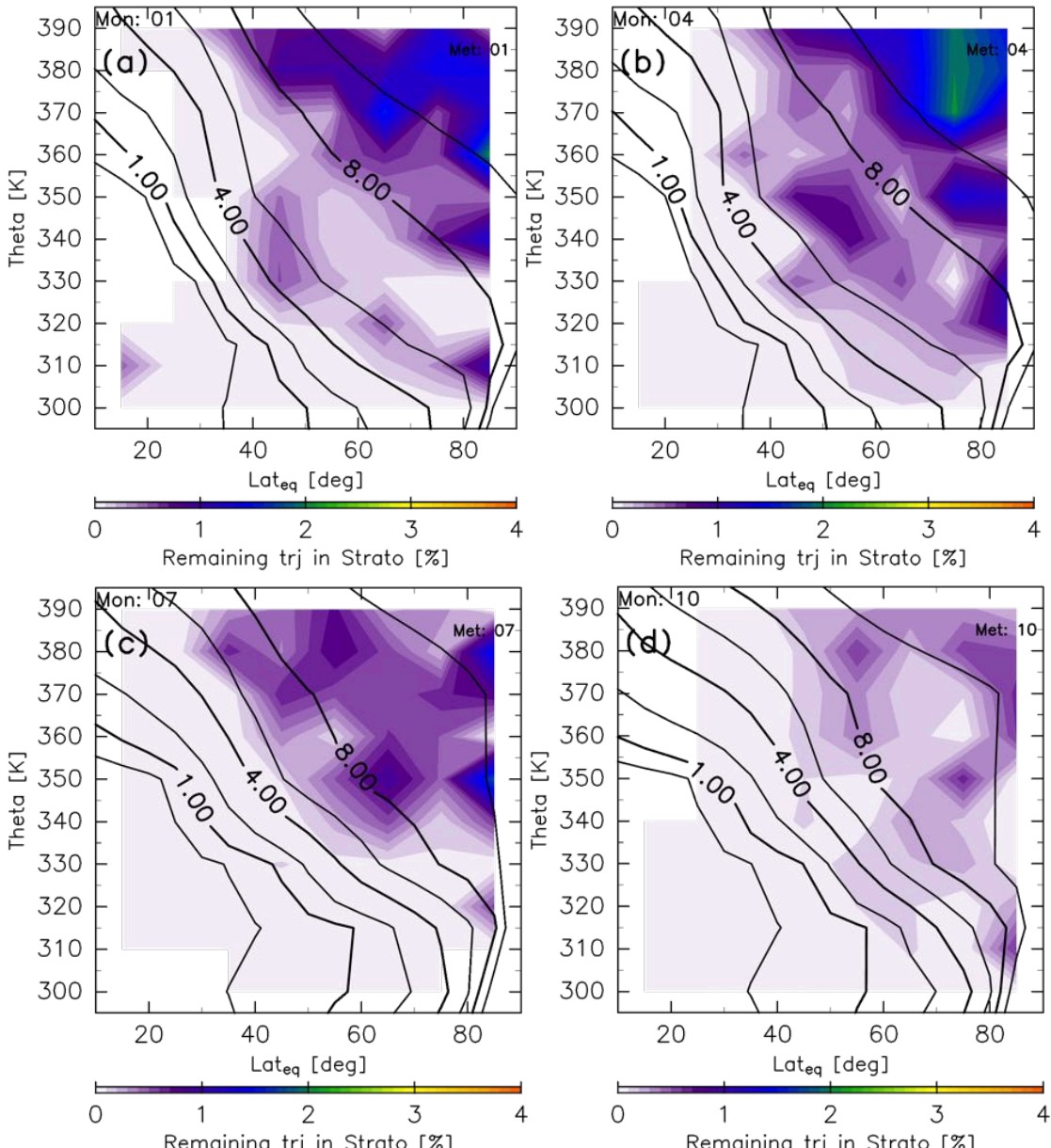

**Figure 3: Meridional distributions of percentage of trajectories that remain in the stratosphere after 10-year backward calculations for (a) January, (b) April, (c) July, and (d) October. Black contours indicate monthly average potential vorticity during 2012–2016.**

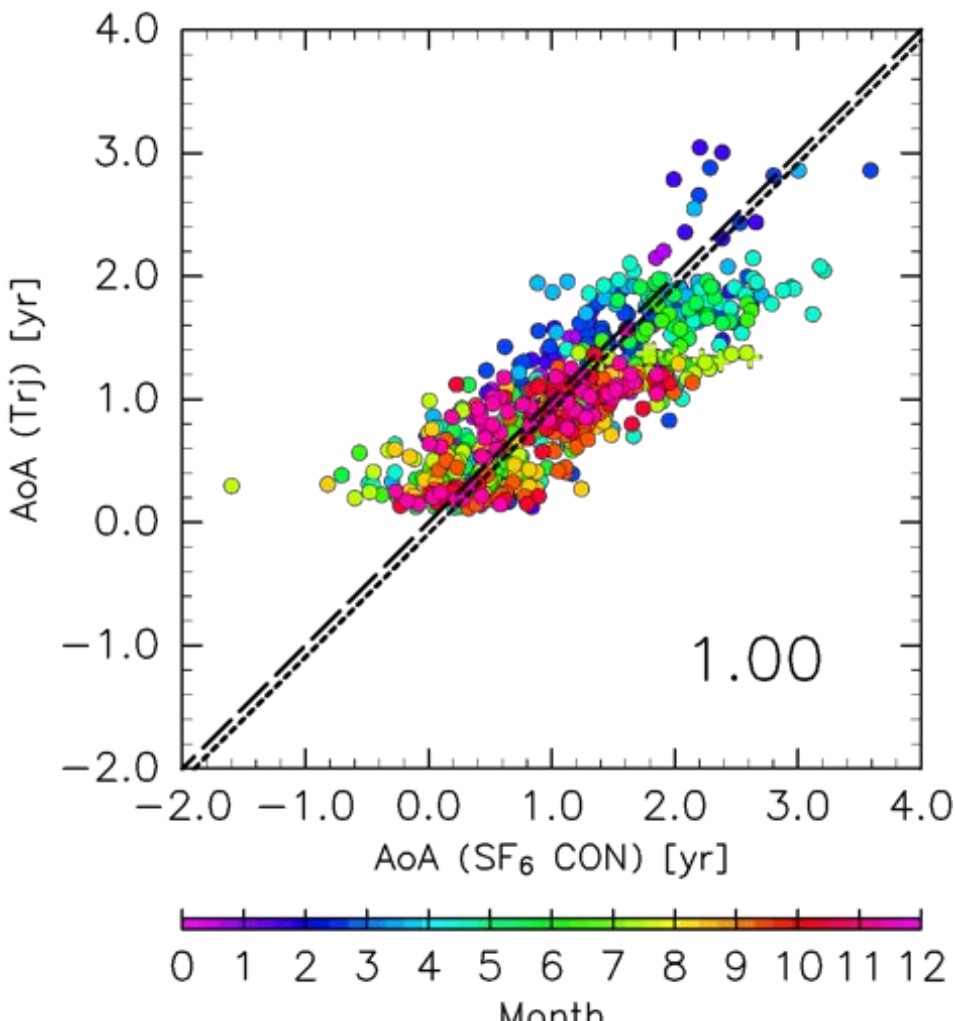

**Figure 4: Scatter plot of the age of air (AoA) estimated from $SF_6$ mixing ratios obtained from CONTRAIL measurements versus those from trajectories with a correction factor of 1.5 (see text for details). Colours indicate the month, and the dashed and dotted lines indicate the 1:1 line and the regression line, respectively. The number in the lower-right of the panel indicates the slope of the regression line. CONTRAIL data with CO mixing ratios higher than 80 ppb in the region above 340 K and north of 60° N equivalent latitude are plotted in crosses.**

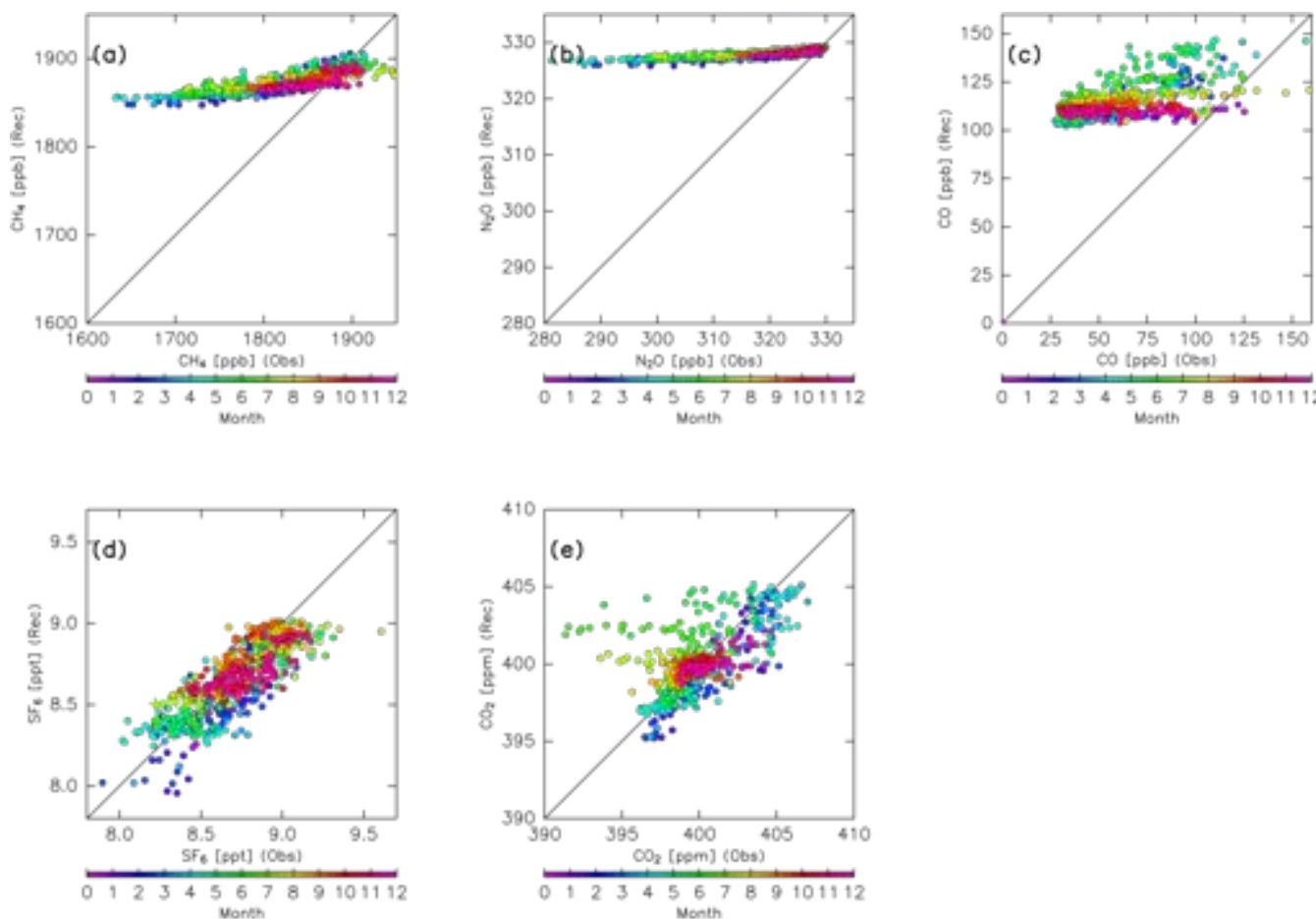

**Figure 5: Scatter plots of CONTRAIL measurements versus reconstructions for (a) CH₄, (b) N₂O, (c) CO, (d) SF₆, and (e) CO₂ without chemical loss. Colours indicate the month.**

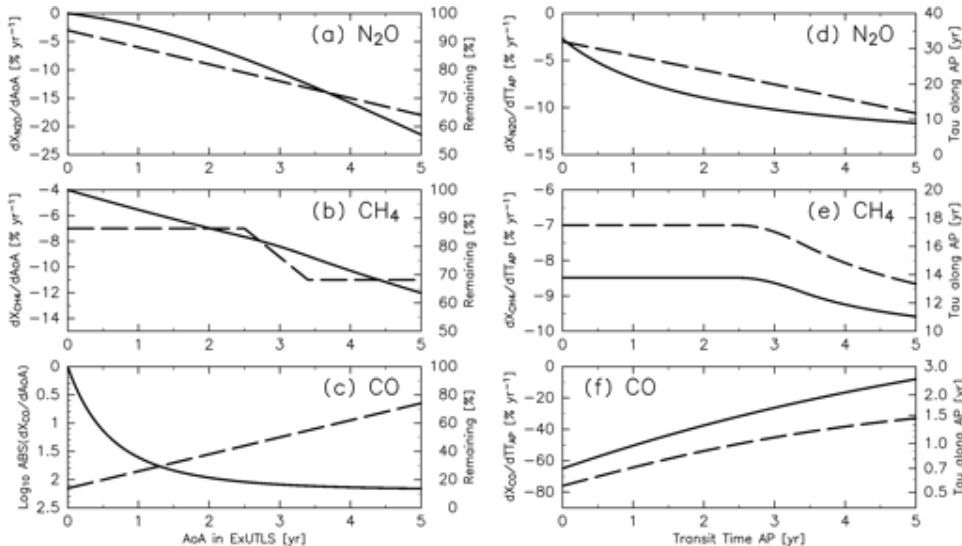

**Figure 6:** Relationships between (a–c) the age of air (AoA) and the gradient of chemical loss rates (dashed lines; left axis) and relative abundance (solid lines; right axis), determined according to figure 6a of Volk et al. (1997) for (a) N$_2$O and (b) CH$_4$, and to Herman et al. (1999) and Krause et al. (2018) for (c) CO (see text for details). Panels (d), (e), and (f) indicate relationships between transit time along the "average path" (AP) and the average chemical loss rate along an AP that produces the same relationship between AoA and the gradient of chemical loss rate shown in panels a, b, and c, respectively (dashed lines; left axis; see text for details), and e-folding times corresponding to chemical loss rates along an AP (solid lines; right axis).

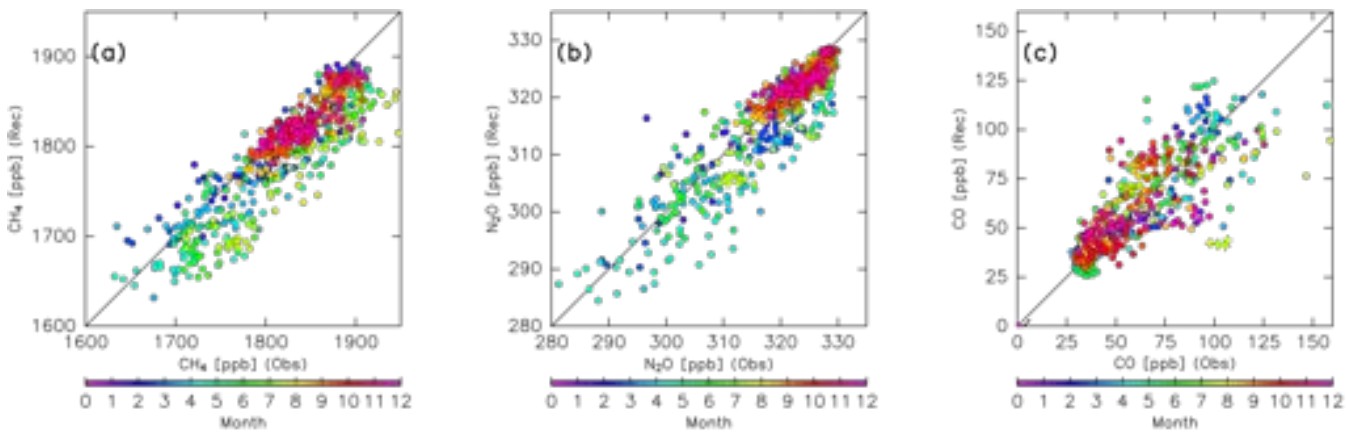

**Figure 7:** As in Fig. 5, but for reconstructions with chemical loss.

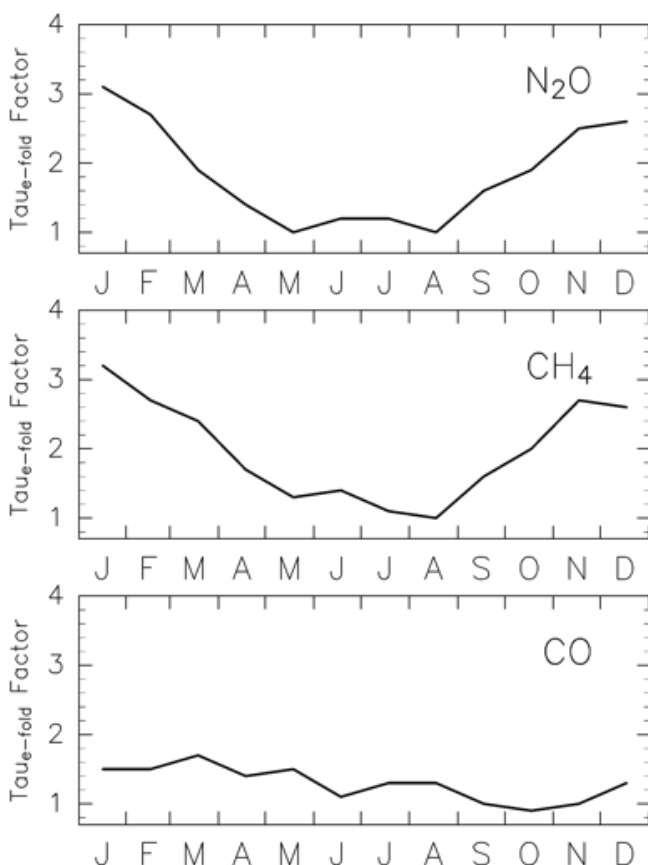

**Figure 8: Estimated correction factor for e-folding times for three chemically active species.**

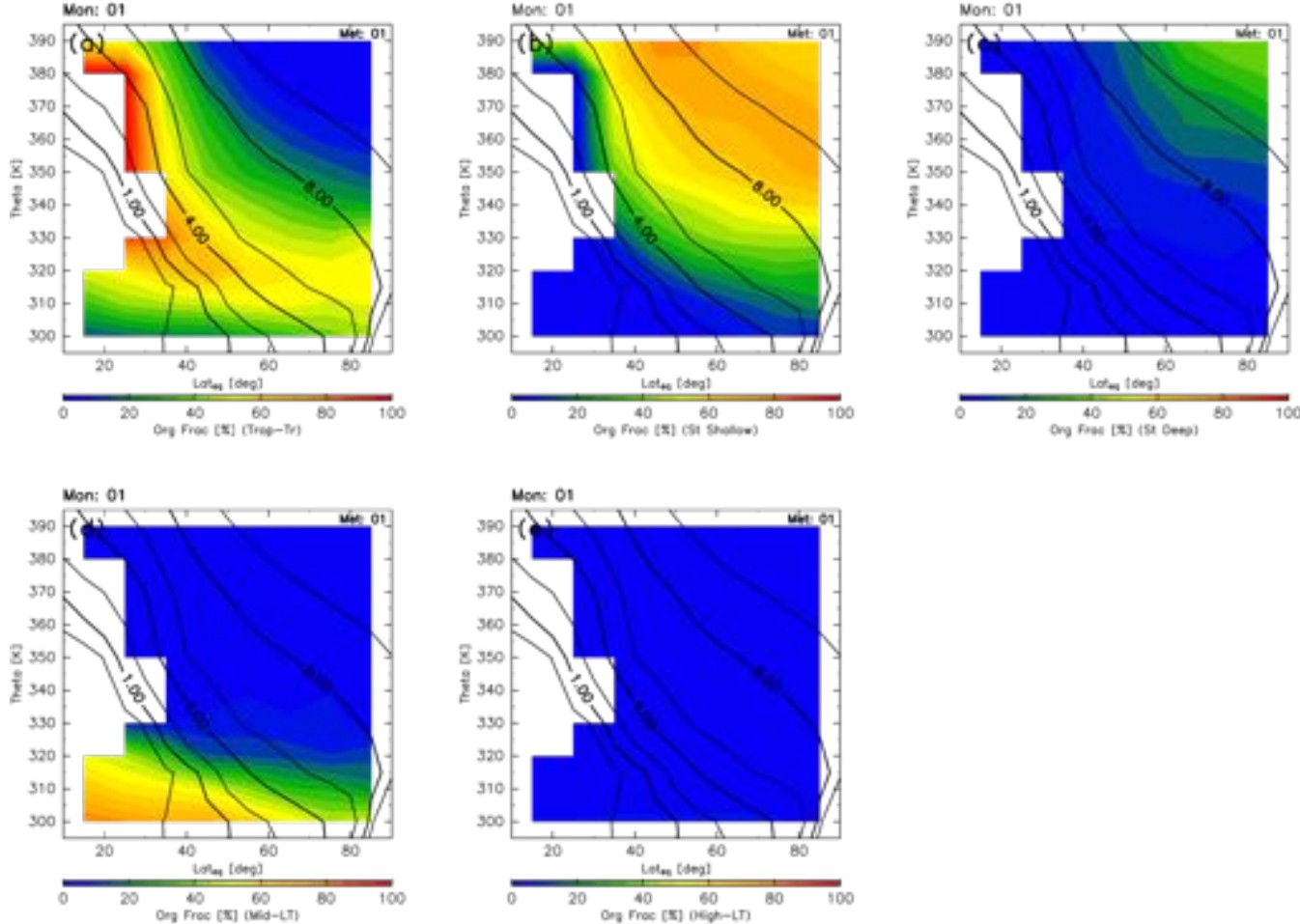

**Figure 9: Meridional distributions of origin fractions for (a) tropical tropospheric, (b) stratospheric (through the shallow branch of the BDC), (c) stratospheric (through the deep branch of the BDC), (d) mid-latitude LT, and (e) high-latitude LT air masses estimated for January. Black contours indicate monthly averaged potential vorticity during the period from 2012 to 2016.**

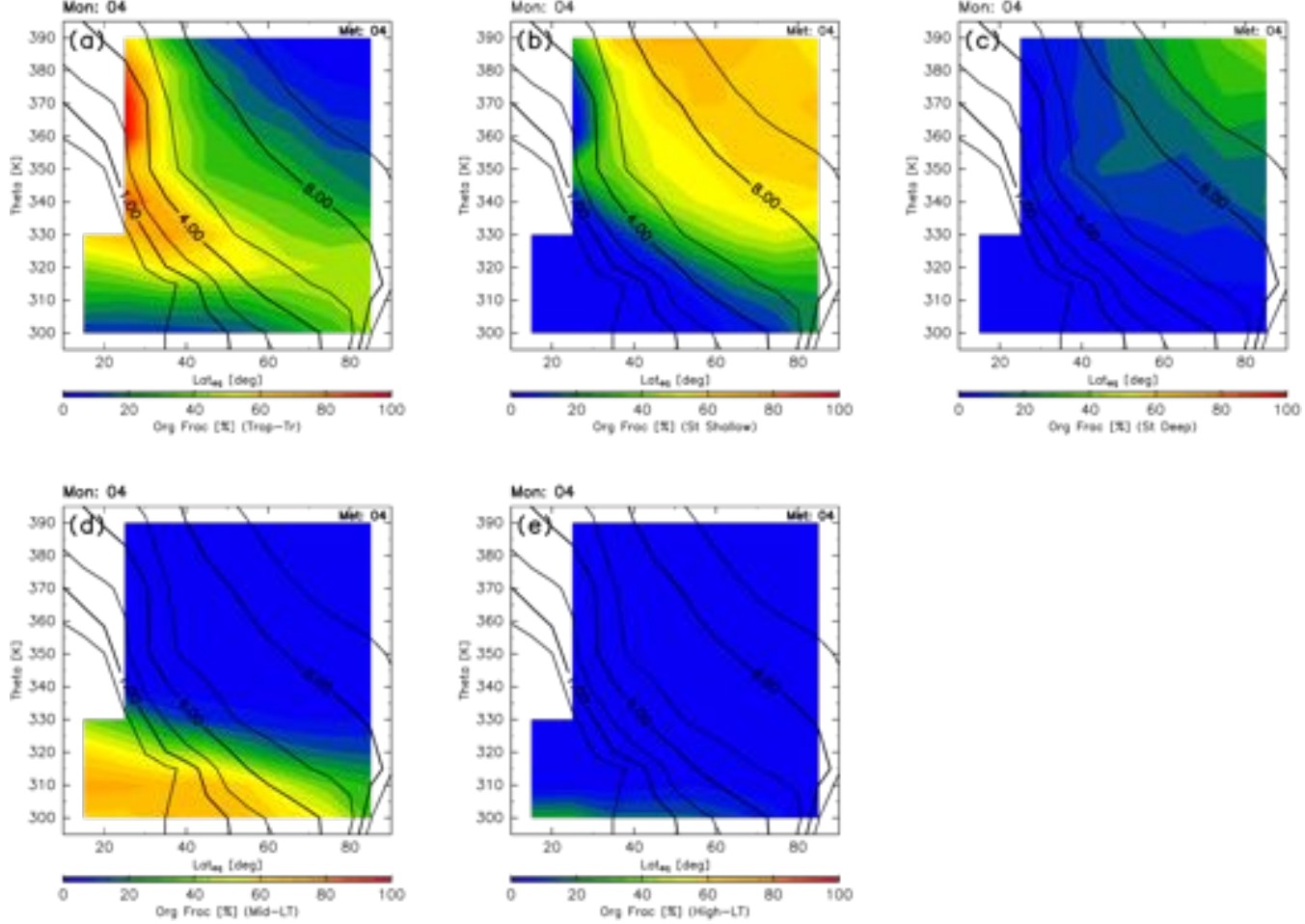

**Figure 10: As in Fig. 9, but for April.**

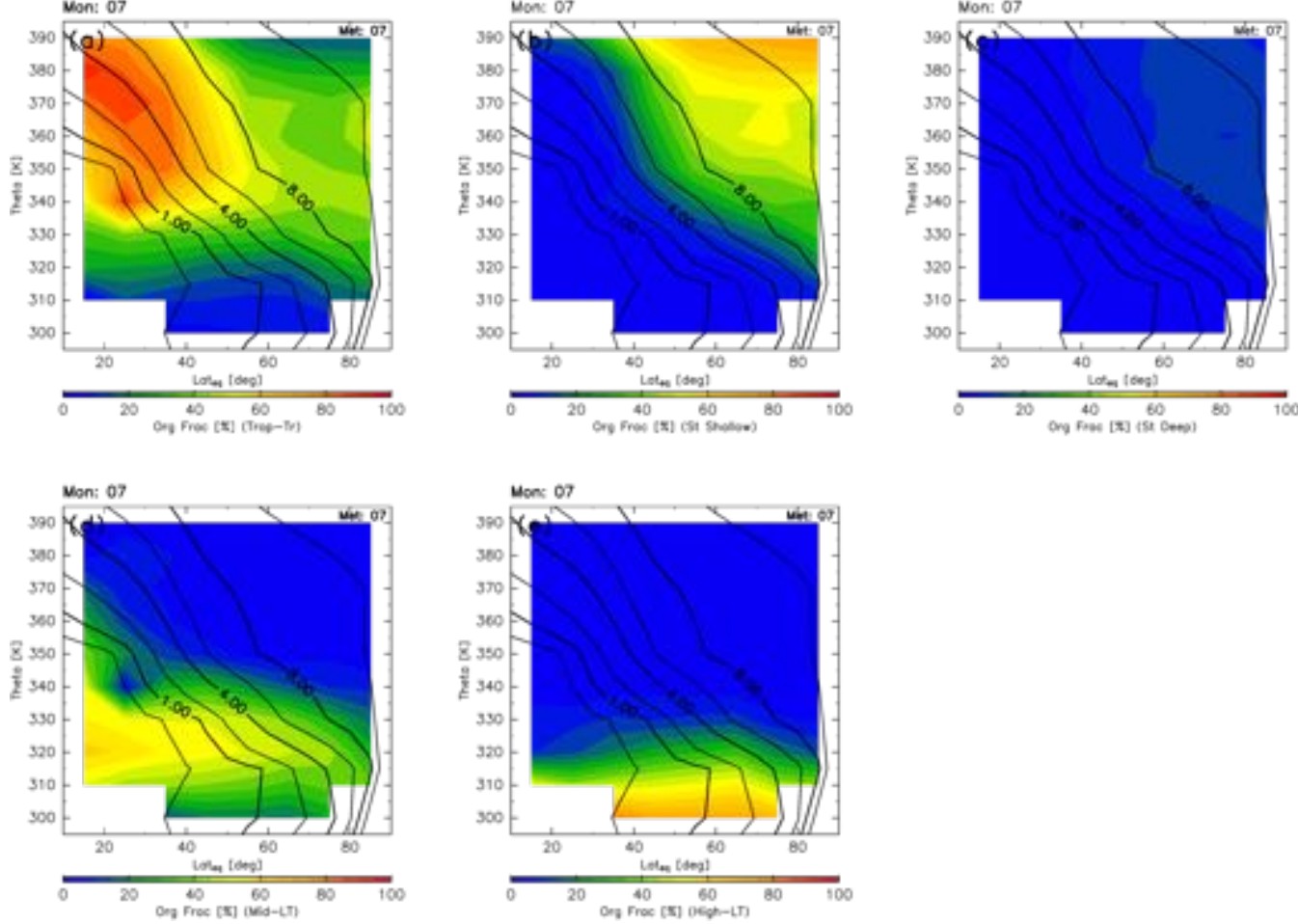

**Figure 11: As in Fig. 9, but for July.**

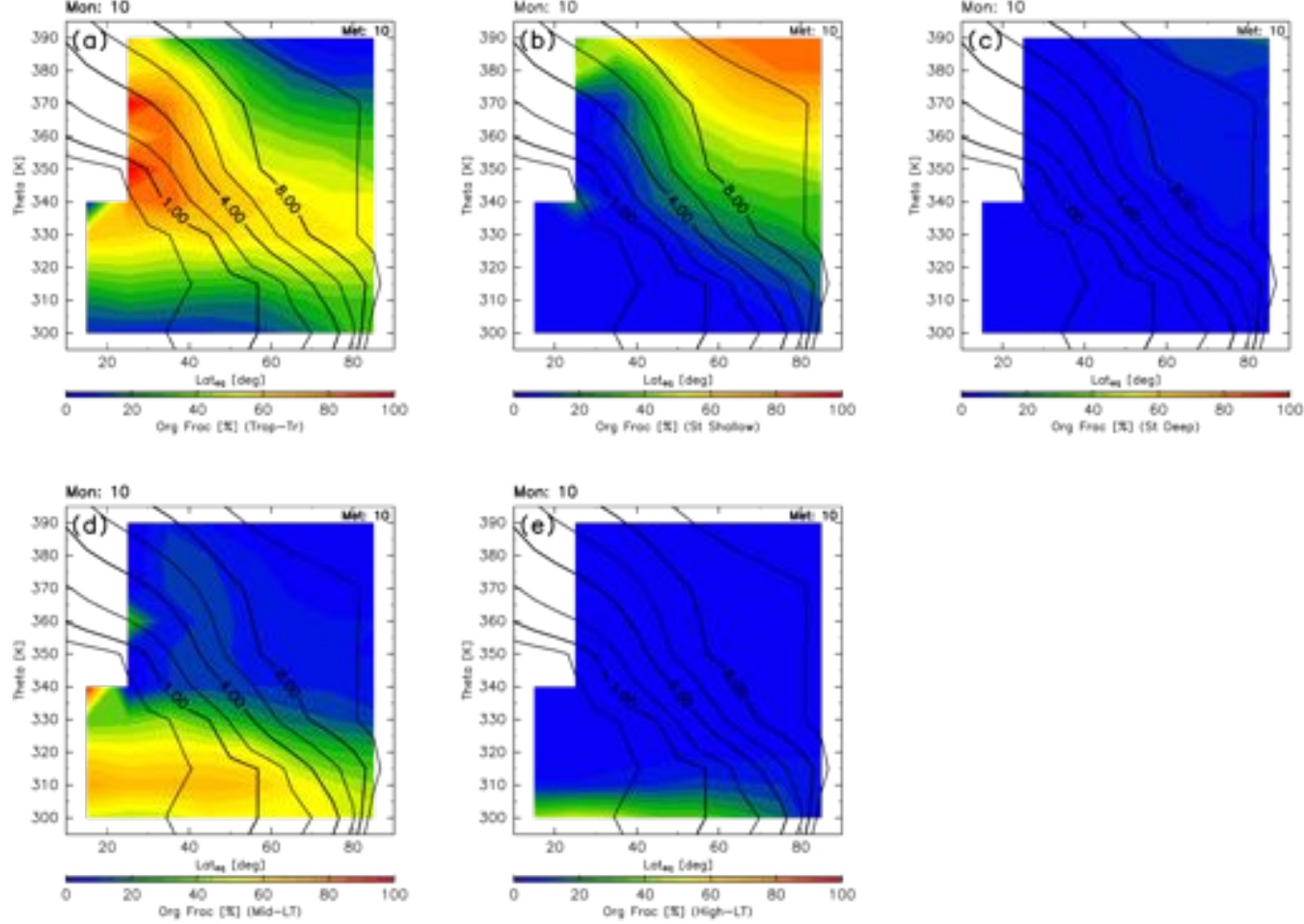

**Figure 12: As in Fig. 9, but for October.**

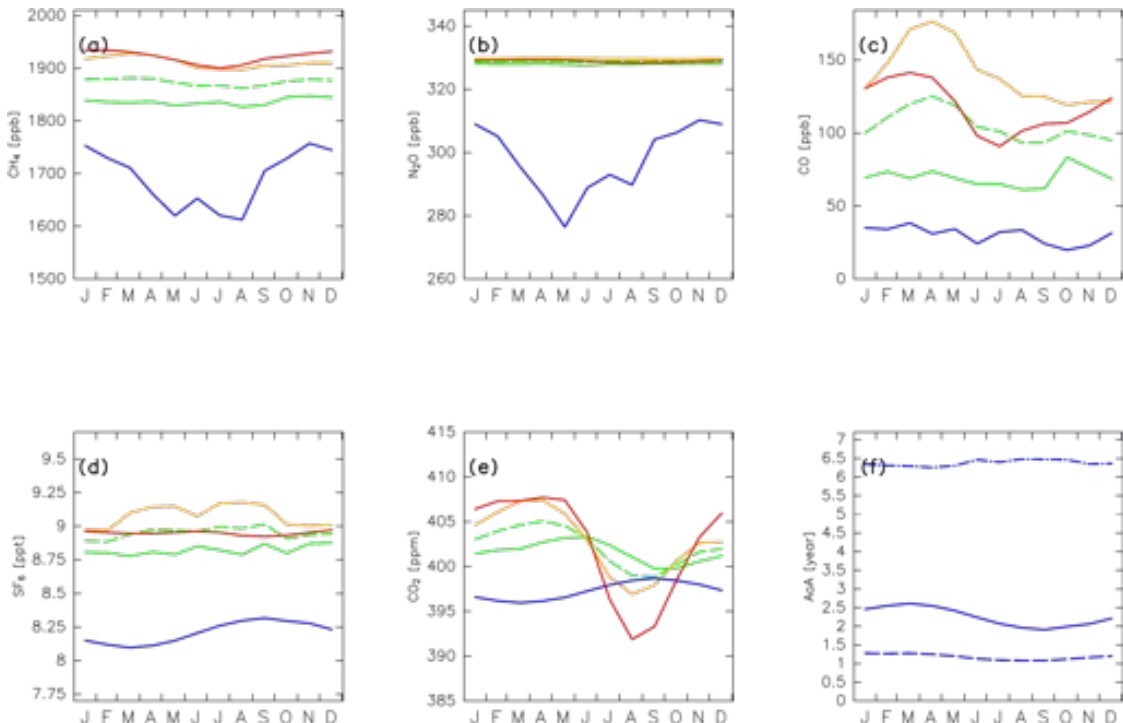

**Figure 13: Seasonal variations in (a) CH₄, (b) N₂O, (c) CO, (d) SF₆, and (e) CO₂ mixing ratios assumed to (solid green) tropical tropospheric, (orange) mid-latitude LT, and (red) high-latitude LT air masses. Note that green dashed lines in (a–e) show the average mixing ratios of the tropical tropospheric and mid-latitude LT, and they are practically assigned to tropical tropospheric air masses to account for underestimations of vertical transport from the LT in the trajectory analysis. Blue lines in (a–e) show the mixing ratios of each species estimated for stratospheric air masses (see text for details). Seasonal variations in the age of air (AoA) estimated for (blue solid lines) stratospheric air masses are shown in (f). Dashed–dotted and dashed lines in (f) indicate the AoA separately estimated for stratospheric air masses that travelled via the deep and shallow branches of the BDC, respectively.**

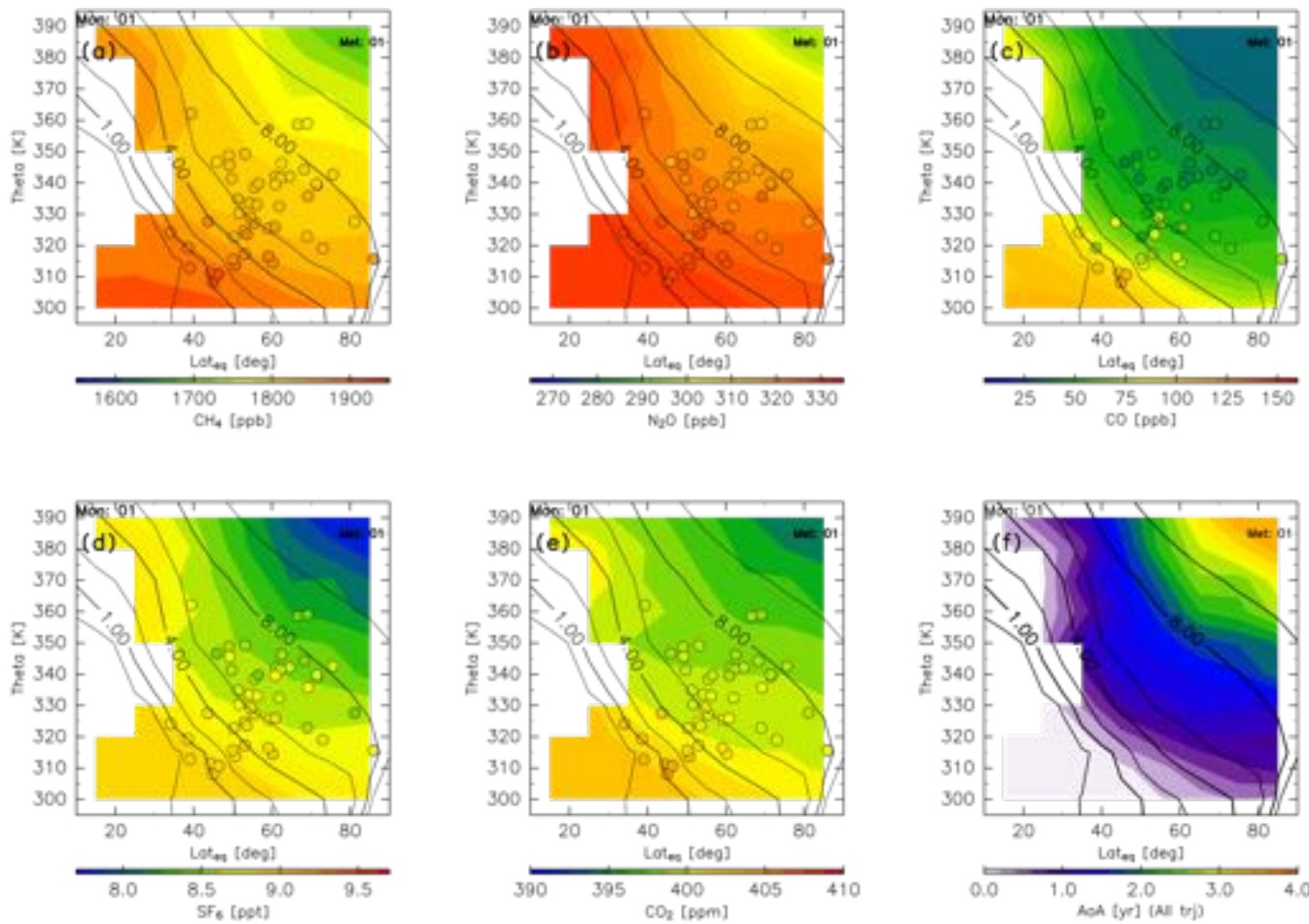

**Figure 14: Meridional distributions of reconstructions for (a) CH₄, (b) N₂O, (c) CO, (d) SF₆, and (e) CO₂ for January. Detrended CONTRAIL measurements in January are plotted as circles using the same colour scale. The distribution of the age of air (AoA) estimated for January is shown in (f). Black contours indicate monthly average potential vorticity during the period from 2012 to 2016.**

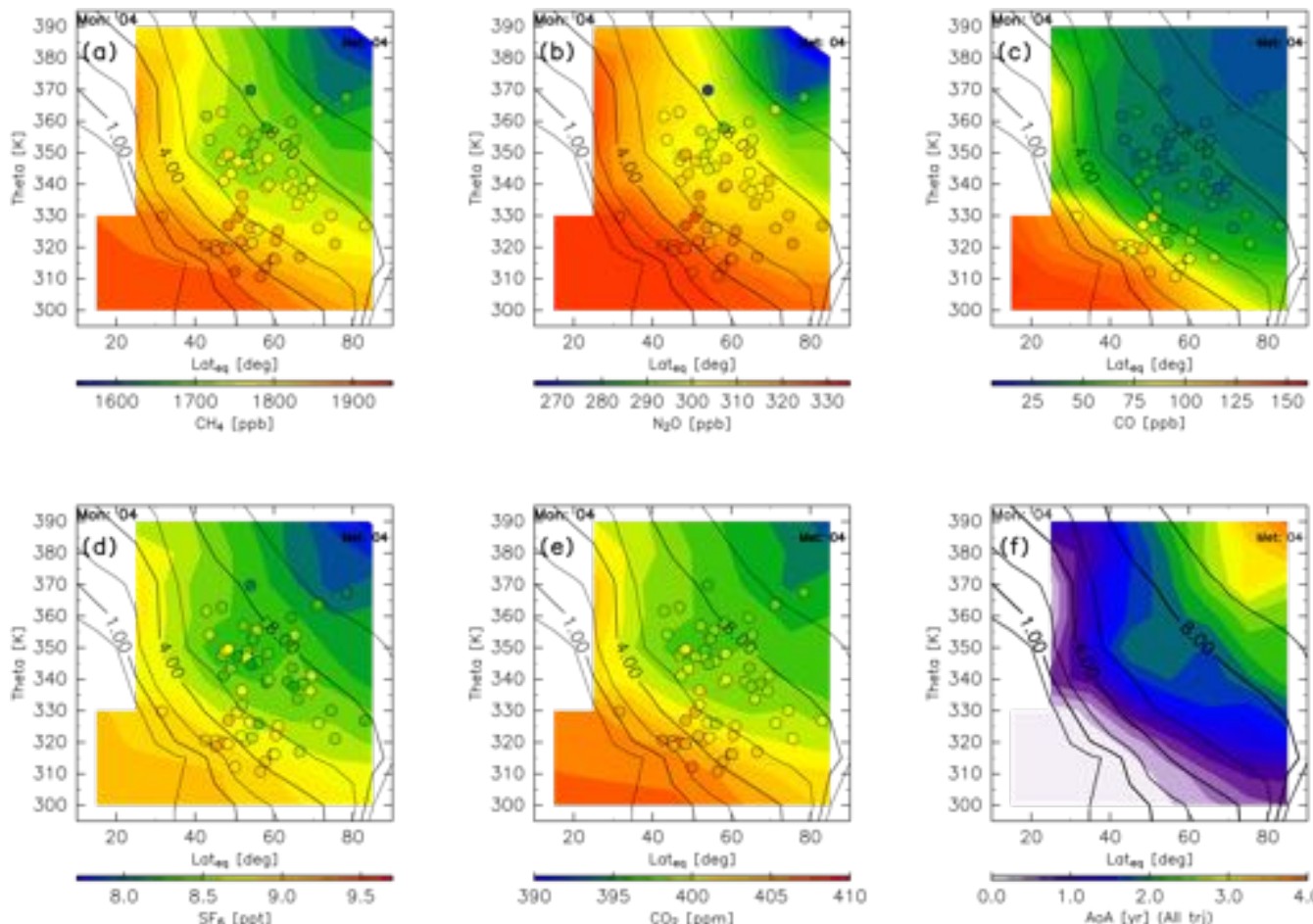

**Figure 15: As in Fig. 8, but for April.**

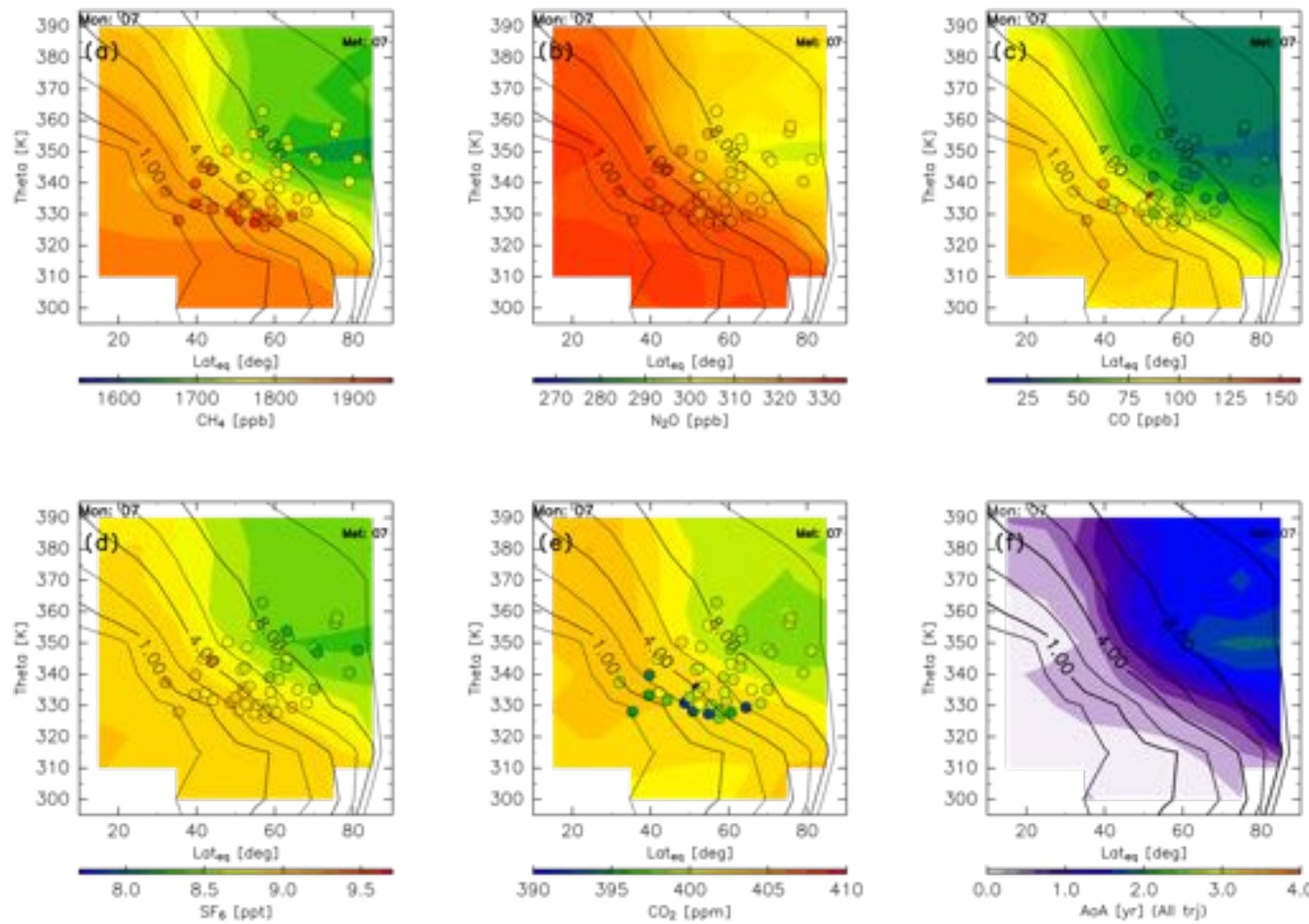

**Figure 16: As in Fig. 8, but for July.**

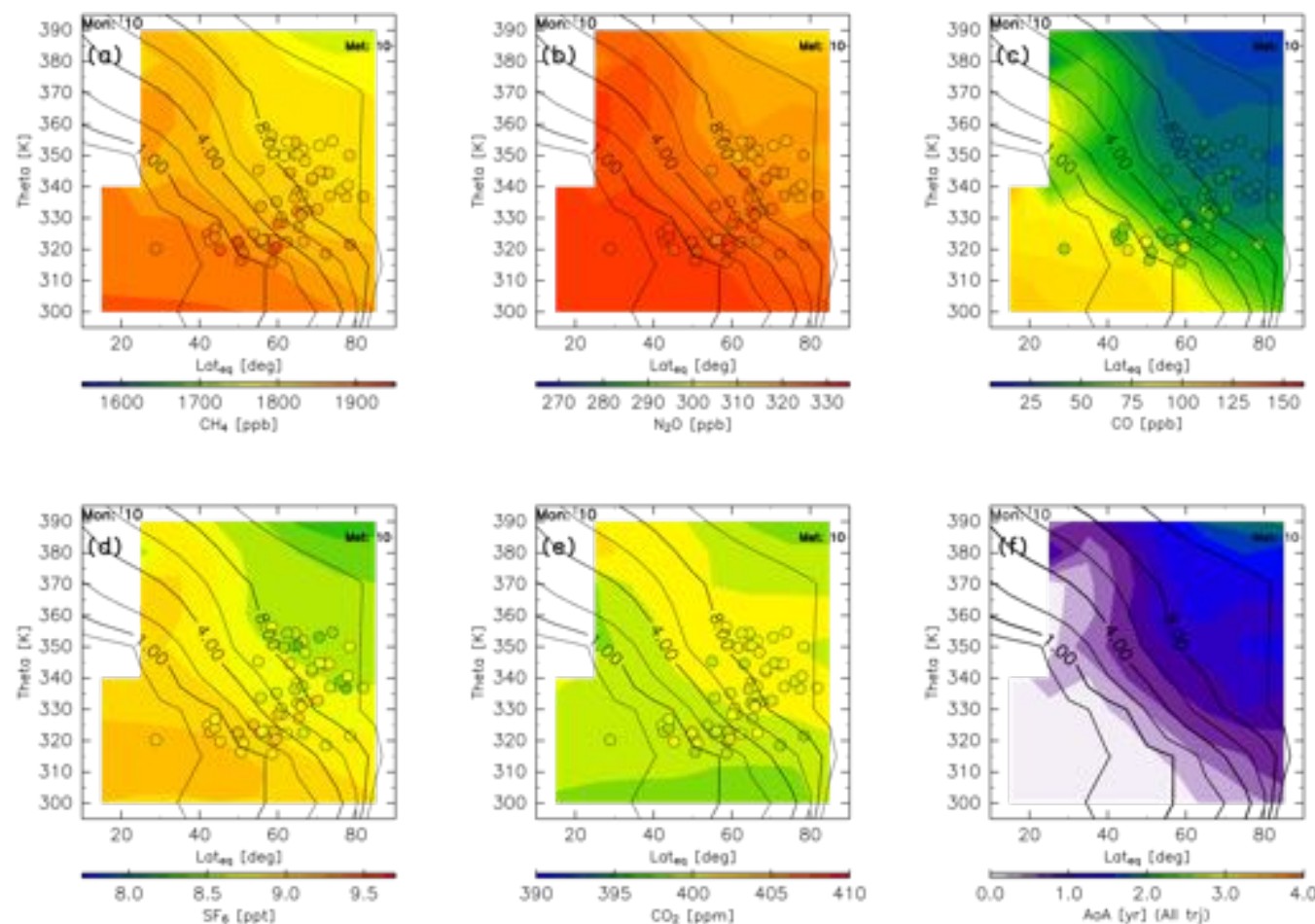

**Figure 17: As in Fig. 8, but for October.**

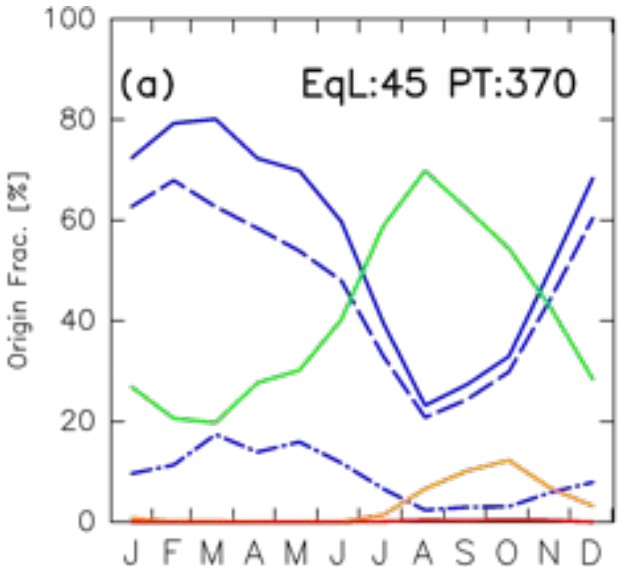
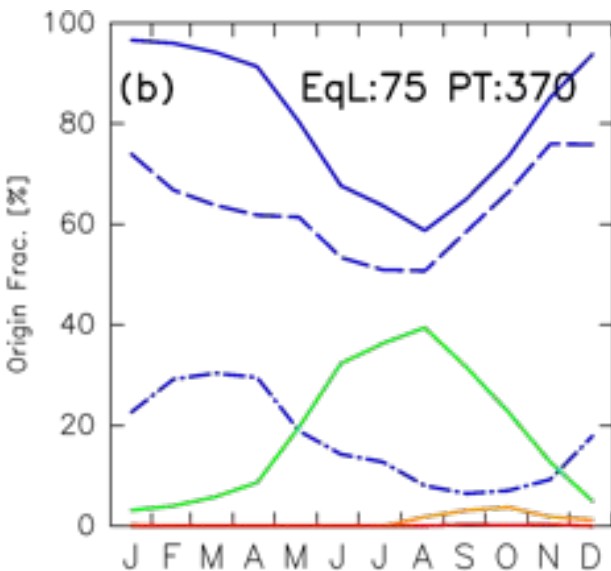
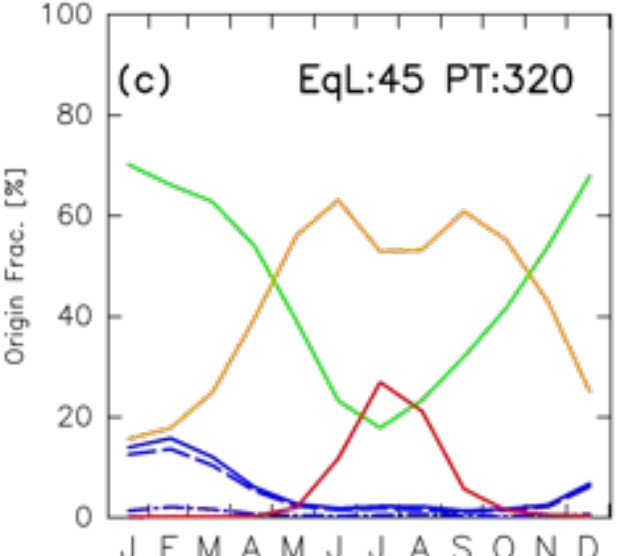
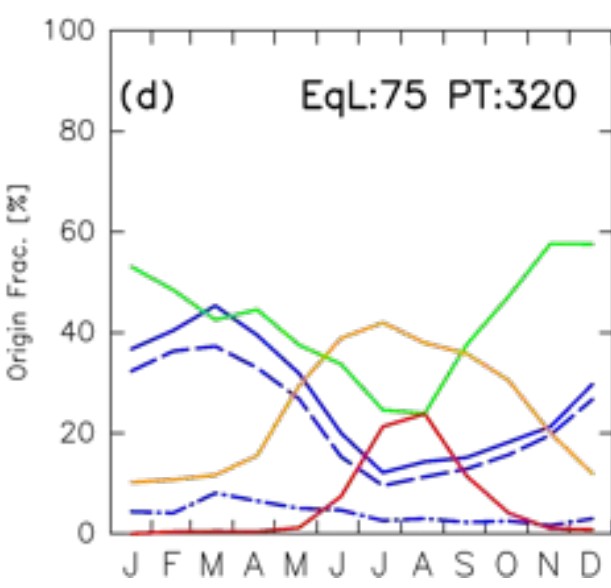

**Figure 18: Seasonal variations in (green) tropical tropospheric, (blue) stratospheric, (orange) mid-latitude LT, and (red) high-latitude LT origin fractions estimated for the (a) mid-latitude upper ($\emptyset_{eq} = 45°N; \theta = 370$ K), (b) high-latitude upper ($\emptyset_{eq} = 75°N; \theta = 370$ K), (c) mid-latitude lower ($\emptyset_{eq} = 45°N; \theta = 320$ K), and (d) high-latitude lower ($\emptyset_{eq} = 75°N; \theta = 320$ K) ExUTLS. The blue dashed–dotted and dashed lines show the origin fractions of stratospheric air masses that travelled through the deep and shallow branches of the BDC, respectively.**

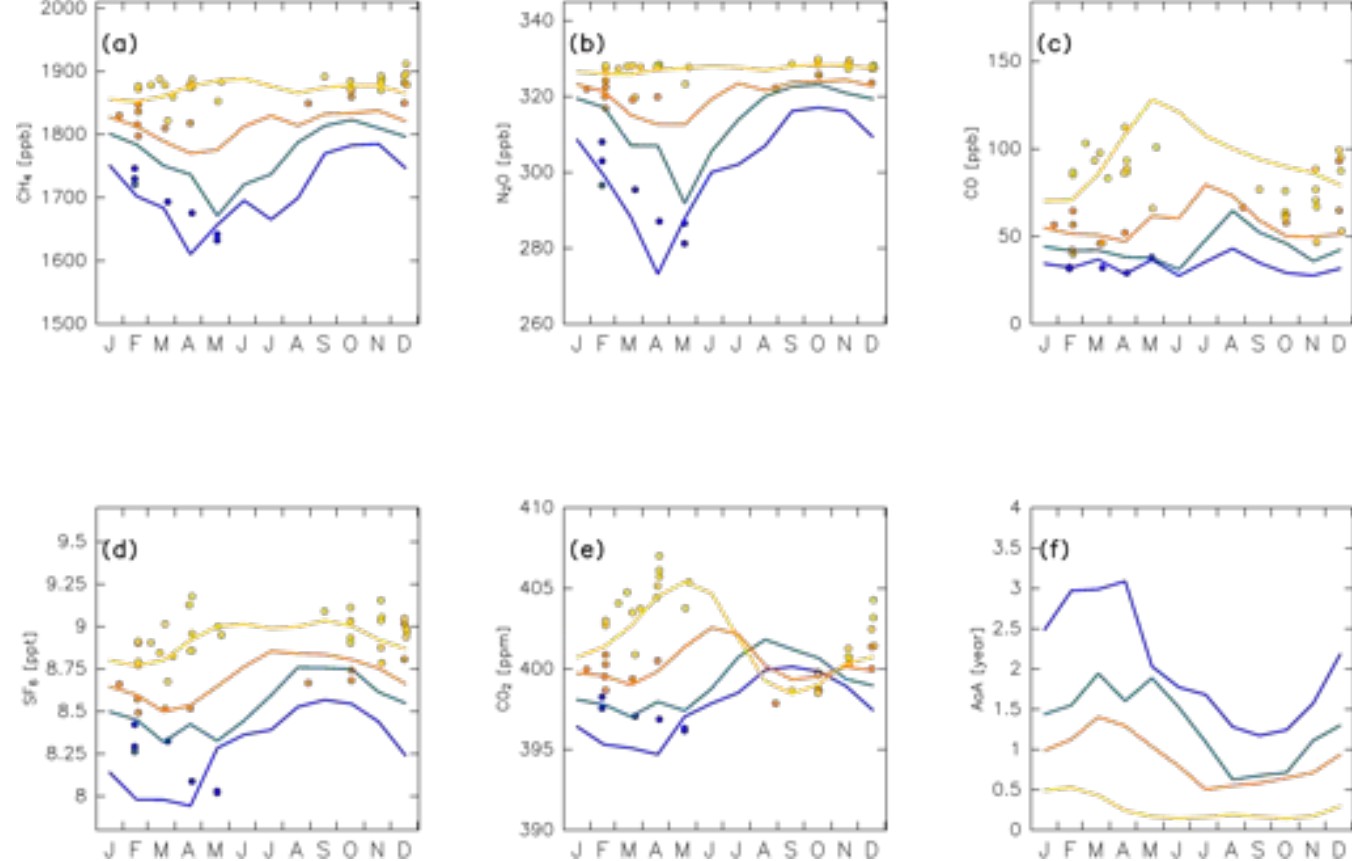

**Figure 19: Seasonal variations in (a) CH₄, (b) N₂O, (c) CO, (d) SF₆, and (e) CO₂ mixing ratios estimated for the (green) mid-latitude upper ($\emptyset_{eq} = 45°N; \theta = 370$ K), (blue) high-latitude upper ($\emptyset_{eq} = 75°N; \theta = 370$ K), (yellow) mid-latitude lower ($\emptyset_{eq} = 45°N; \theta = 320$ K), and (orange) high-latitude lower ($\emptyset_{eq} = 75°N; \theta = 320$ K) ExUTLS superimposed on detrended CONTRAIL measurements, which are colour-coded according to measurements within ±5° in equivalent latitude and ±5 K in potential temperature of the reconstruction regions. Seasonal variations of the age of air (AoA) estimated for the same locations are shown in (f).**

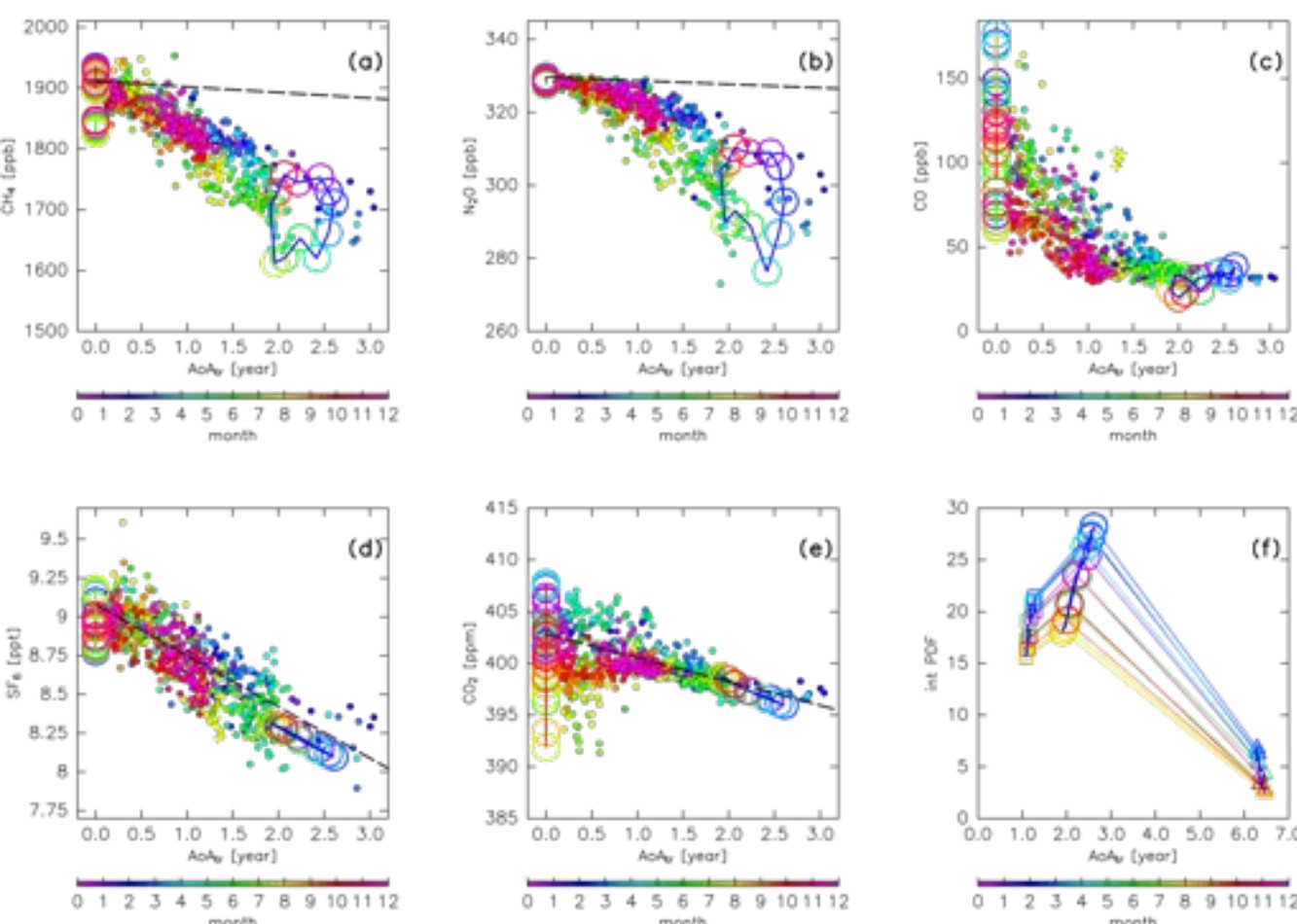

**Figure 20: Scatter plots of the mean age of air (AoA) versus (a) $CH_4$, (b) $N_2O$, (c) CO, (d) $SF_6$, and (e) $CO_2$ mixing ratios measured by CONTRAIL (filled circles; colours indicate the month). Lines with open circles, coloured according to month, show the original compositions for (green) tropical tropospheric, (blue) stratospheric, (orange) mid-latitude LT, and (red) high-latitude LT air masses. Dashed lines in (a), (b), (d), and (e) show the sign-reversed trends of tropospheric $CH_4$ (−9.3 ppb/year), $N_2O$ (−1.0 ppb/year), $SF_6$ (−0.33 ppt/year), and $CO_2$ (−2.3 ppm/year) with intercepts of the annual averaged mixing ratios at mid-latitudes for 2016 (1911 ppb, 330 ppb, 9.08 ppt, and 403 ppm), respectively. Mixing ratios estimated for stratospheric air masses in (a–e) are plotted after taking 3-month running averages to reduce fluctuations. Panel (f) shows the AoA estimated for air masses originating in the stratosphere (open circles), along with those estimated only for air masses passing through the deep (triangles) and shallow branches (squares) of the BDC. The ordinate is the integral of PDF of the "age spectrum" for each subset.**

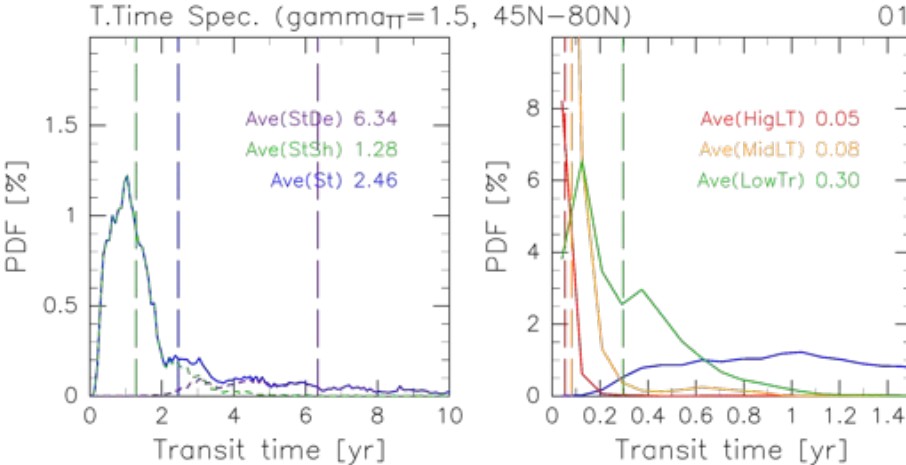

**Figure 21:** "Age spectrum" (probability distribution function; PDF) for (left) air masses originating in the (blue) stratosphere as well as those separately evaluated for air masses that have travelled via the deep (dashed purple) and shallow branches of the BDC (dashed green), and (right; green) tropical troposphere, (orange) mid-latitude LT, and (red) high-latitude LT estimated for January. Note that the transit time is corrected with $\gamma TT = 1.5$, as described in Sect. 2.1. Vertical dashed lines indicate average AoA calculated for air masses from each origin.

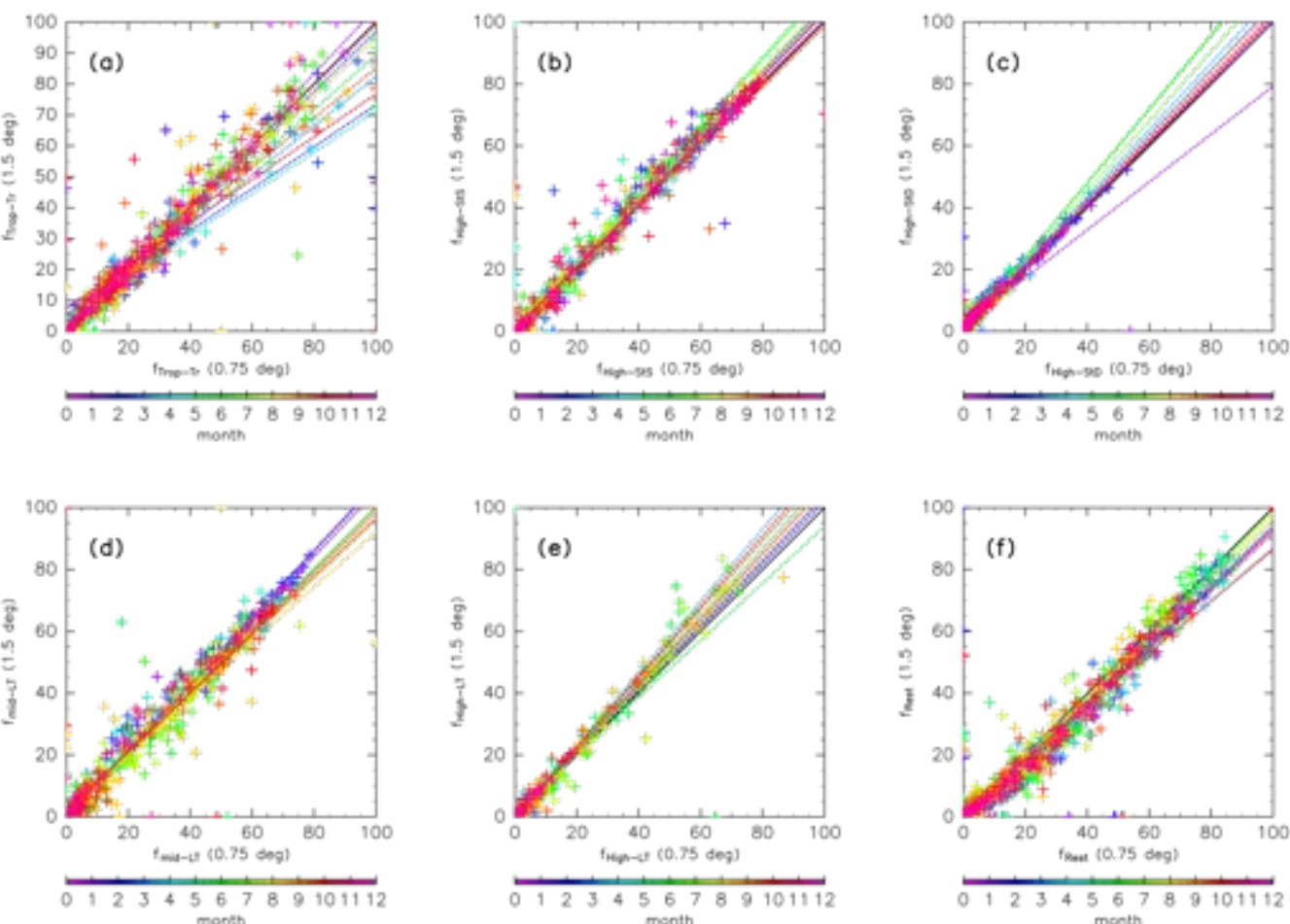

**Figure A1: Scatter plots of origin fractions calculated using ERA-Interim data with a horizontal resolution of 0.75° and 60 model levels versus those with 1.5° horizontal resolution and 37 pressure levels. Both are estimated by 90-day trajectory calculations because of computing limitations. Crosses indicate mixing fractions evaluated for all bins in the $\emptyset_{eq}$–$\theta$ cross-sections shown in Figs 10–13 for (a) tropical tropospheric, (b) stratospheric (through the shallow branch of the BDC), (c) stratospheric (through the deep branch of the BDC), (d) mid-latitude LT, (e) high-latitude LT, and (f) unclassified air masses from 90-day trajectories. Colours indicate the month and dotted lines indicate the regression line for each month.**

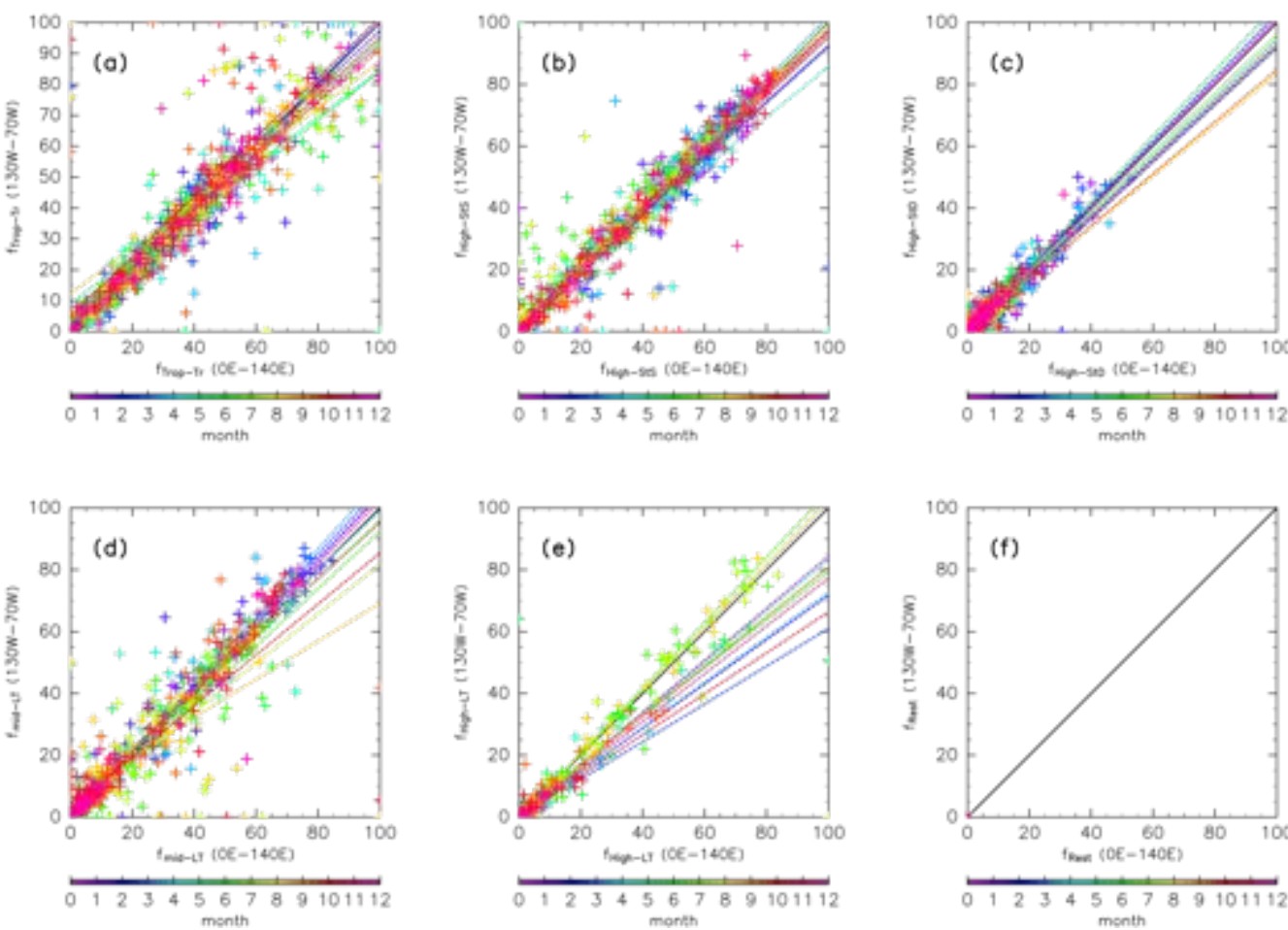

**Figure A2: As in Fig. A1, but for origin fractions using 10-year trajectories calculated for the longitudinal region within 0° E–140° E (default) versus those for the region within 130° W–70° W.**

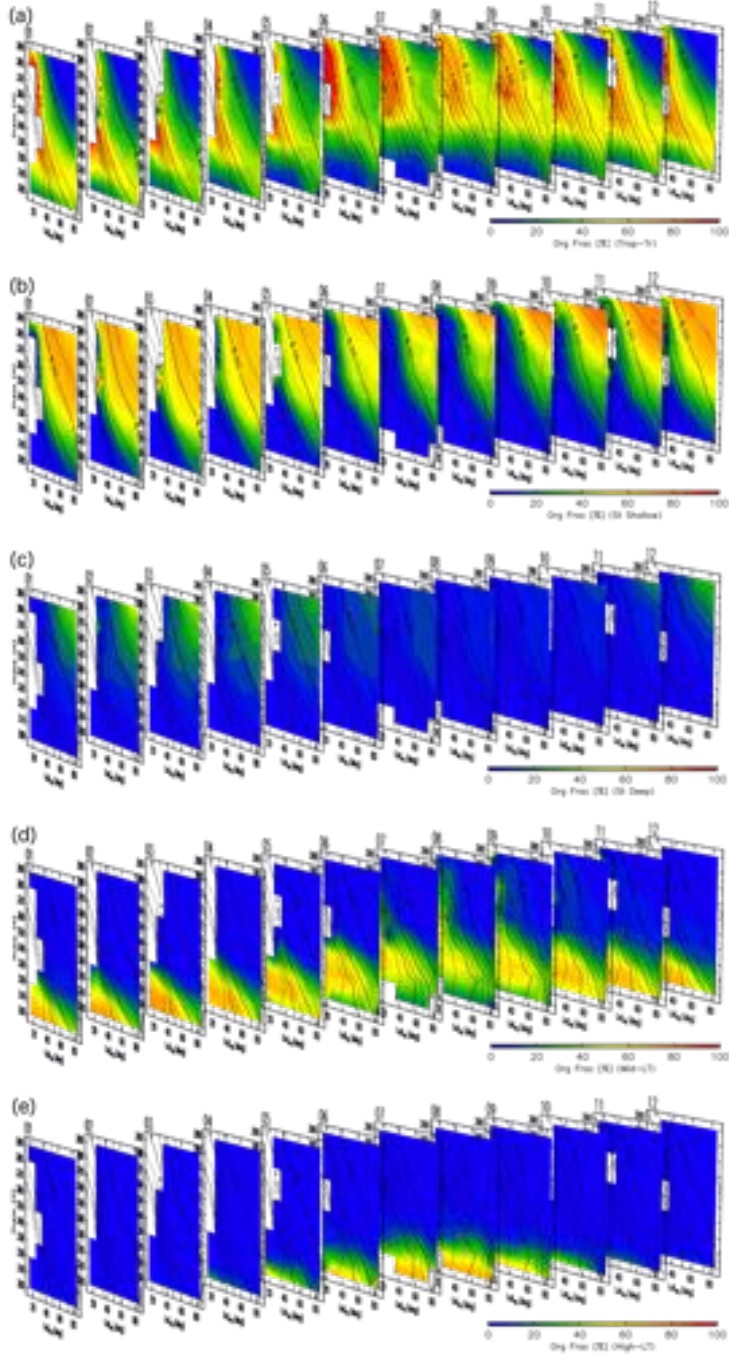

**Figure B1: Origin fractions for (a) tropical tropospheric, (b) stratospheric (via the shallow branch of the BDC), (c) stratospheric (via the deep branch of the BDC), (d) mid-latitude LT, and (e) high-latitude LT air masses estimated for each month of the year with axes as in Figs 9–12. Black contours indicate monthly average potential vorticity during the period 2012–2016.**

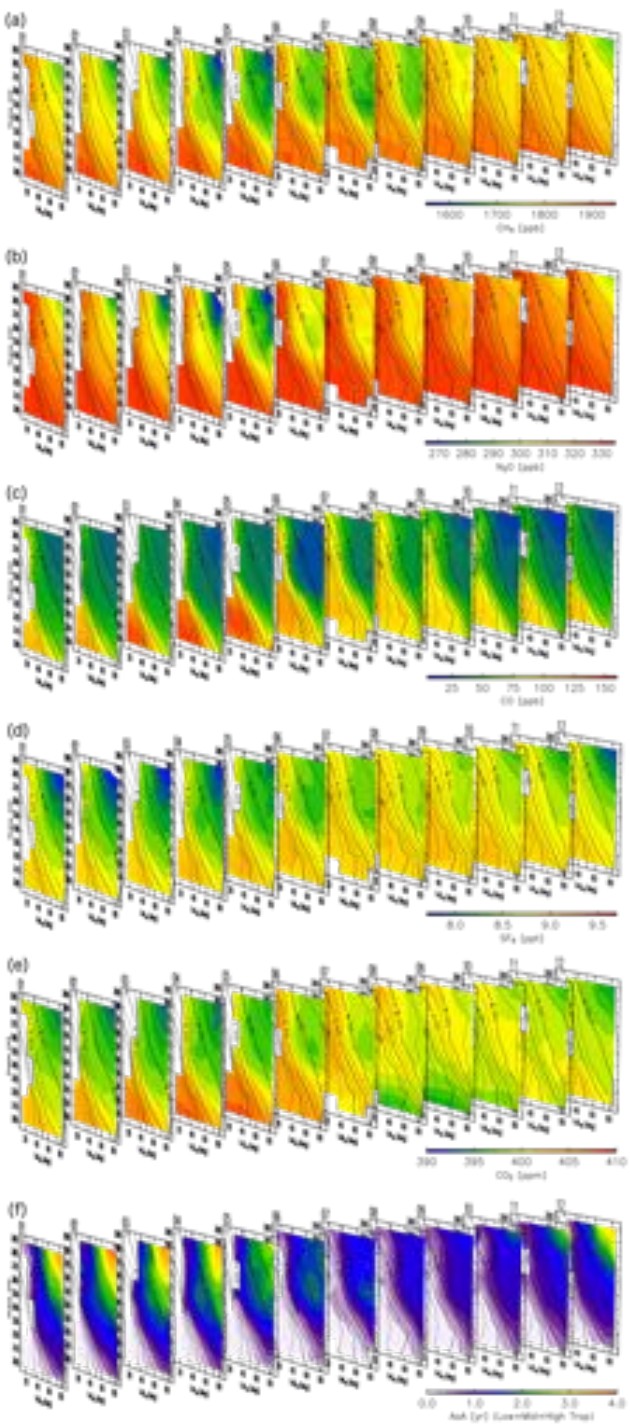

**Figure B2: As in Fig. B1, but for reconstructions for (a) CH₄, (b) N₂O, (c) CO, (d) SF₆, and (e) CO₂, along with (f) the age of air (AoA) for each month.**