# Peer review of "Seasonal characteristics of trace gas transport into the extratropical upper troposphere/lower stratosphere"

_Atmospheric Chemistry and Physics, 2018_

## Referee Comment (RC1) · Anonymous Referee #1 · 27 Dec 2018

Reviewer (Comments):
**Review of "Seasonal characteristics of chemical and dynamical transports into the extratropical upper troposphere/lower stratosphere" (ExUTLS) by Yoichi Inai et al.**

**Recommendation: Publication after major revision**

The paper is very well organised and written. The topic discussed in this paper, transport into the ExUTLS, is in general of high relevance. Our actual limitations in simulating water vapour transport in this complex region introduce large uncertainties in the Earth radiation budget (see Riese et al., 2012). Trajectory analysis in combination with observations could be and have been used in many cases to improve our knowledge on tracer transport and distribution in the UTLS, e.g. for H2O: Fueglistaler et al. (2004, 2005a, 2005b), e.g. for CO and $H_2O$: Hoor et al. (2010), e.g. for STE (stratosphere-troposphere exchange) and $O_3$: Skerlag et al. (2014), e.g. for $CO_2$ and AoA: Diallo et al. (2012, 2017). This manuscript here falls a bit short of explaining what the novel aspect of the presented method really is and how the presented results augment our actual knowledge on the seasonal characteristics of transport in and into the ExUTLS, e.g. in comparison to the early studies by Appenzeller et al. (1996) and Ray et al. (1999) or to the many studies summarised in the ExUTLS review paper by Gettlemann et al. (2011).

The paper should be submitted after addressing the comments below.

**General comments:**

First of all, I don't fully understand the title (and/or the scope) of this paper. What is the meaning of chemical and dynamical transport? Is chemical transport a synonym for transport of chemical active tracers ($N_2O$, $CH_4$ and CO)? Should dynamical transport describe the transport of passive tracers ($SF_6$, $CO_2$ and AoA)? Pure Lagrangian transport (here backward trajectories) differs from (both) tracer transports: There is no mixing and no chemistry included along the individual transport pathways.
The latter is definitively a problem for CO, because the chemical decay along the 90-days backward trajectories cannot be neglected. For $N_2O$ and $CH_4$ the chemical decay is not that significant, because only a few of the initialised trajectories will travel through the sink regions of both tracers higher up in the stratosphere during the 90-days. However, the unknown (non-observed) time series of the high-latitude stratospheric background (k=1) and the tropical and extratropical UTLS background (k=5) conditions for all tracers (not only for the chemical active tracers $N_2O$ and $CH_4$) still remain the major problem for the reconstruction of the observed tracer distributions by CONTRAIL using only 90-days backward trajectories. The reconstruction in the chosen setup could not be used for quantitative studies of neither ERA-Interim (or other reanalysis data sets) nor the transport processes in the ExUTLS, because the trajectories itself are needed to define the boundary conditions for the original time series $X^S_{ORG\_k}$ in the high-latitude stratosphere (k=1) and the UTLS (k=5) which are again the prerequisite to reconstruct the observed mixing ratios in the ExUTLS derived from CONTRAIL. This is a circular reference between the trajectory analysis and the CONTRAIL observations in the ExUTLS, whereby the non-observed (inverse reconstructed) original time series for k=1 and 5 could be seen as a kind of free parameters to tune the system or in other words to close the budget for the individual tracers.

The main problem, why this paper could, to my point of view, not add to the actual state of knowledge, although it has the potential, is the limitation of the backward trajectories to 90 days. The consequence of this limitation is the circular reference explained above (the authors call this an inversion technique) that has to be introduced to reconstruct the original time series of the tracer mixing ratios in the stratospheric overworld (k=1) and in the UTLS (k=5).

The authors claim that the mixing fractions derived from the coarser resolution ERA-Interim data (1.5x1.5, 37 levels) are the same as for the finer resolution ERA-Interim data (0.75x0.75, 60 levels). If this is the case, why not using the 10-year instead of the 90-days backward trajectories? This would at least solve the problem to reconstruct the non-observed boundary conditions for the trajectories residing in the ExUTLS (k=5) during the 90 days and also partially for the trajectories originating from the stratospheric overworld (k=5). The latter is unfortunately only true for the passive tracers, $SF_6$ and $CO_2$ respectively. Both tracers could be reconstructed by combining their original tropospheric time series (k=2,3,4) and the 10-years trajectories all starting in the target region of this study – the ExUTLS – and ending in the troposphere (e.g. Diallo et al., 2017), beside a small residuum of trajectories that remains in the stratosphere and that has to be characterised (see e.g. Ploeger et al, 2016).

The strong point of this study here is to my opinion the combination of backward trajectories driven by a state-of-the-art reanalysis data set (ERA-Interim) with simultaneous measurements in the ExUTLS of five tracers with different characteristics in their lifetimes and tropospheric time series. My recommendation would be to separate the analysis on transport and chemical processes. In the first step, one could use the 10-years backward trajectories (if manageable, it would be better using the high resolution ERA-Interim data) together with the passive tracers $CO_2$ and $SF_6$ (the latter is literally the same as AoA in the UTLS) to evaluate the mixing fractions and transport timescales derived from the ERA-Interim driven backward trajectories by reconstructing the CONTRAIL observations of both passive tracers. This is already a valuable extension to the method shown by Diallo et al. (2017), because the additional simoultaneous $SF_6$ observations are a second independent constraint for the evaluation due to the different (linearly independent) tropospheric time series compared to $CO_2$.

In the next step, one could exploit the additional information from the simultaneously measured chemical active tracers. Given the transport properties (mixing fraction, air mass origin and timescale) that has been quantified and evaluated in the first step, the chemical decay along the transport pathways from the tropospheric origin into the ExUTLS could be analysed with the simultaneous measurements of the chemical active tracers CO, $CH_4$ and $N_2O$. The difference between the reconstructed passive CO, $CH_4$ and $N_2O$ tracers without chemistry and observed values including chemical decay should allow to assign a photochemical loss for air parcels along an "average pathway" with the same AoA (see Schoeberl et al., 2000). This "average pathway" could be defined by a bulk of trajectories, e.g. by the trajectories in a given equivalent latitude-potential temperature bin. There might be many more and better approaches to derive quantitative information on the chemical decay along the transport pathways, but the huge advantage in general of using backward trajectories together with simoultaneous measured passive and chemical active tracers would be that one could disentangle dynamical and chemical effects on the observed tracer distribution in a unique way. An urgent question that has to be answered to understand the processes driving and driven by climate change in this complex and important region of the atmosphere.

If the authors decide to stay with 90-days backward trajectory setup then the limitations of the 90-days backward trajectories and the sensitivity of the results due to these limitations have to be discussed in much more detail – see also the specific comments below.

**Specific comments:**

P.1, L.26: "..., *especially in the Arctic climate.*"
Please cite references for this statement or delete it.

P.1, L.27: "*... via stratospheric residual circulation (Brewer-Dobson circulation, BDC; Brewer, 1949; Dobson, 1956).*"
The stratospheric residual circulation describes the mean mass transport. The BDC includes mean mass transport and two-way mixing. The latter, by definition, does not lead to net mass exchange but may lead to net tracer exchange. Therefore, I would suggest to use stratospheric circulation instead of stratospheric residual circulation as a synonym for the BDC (see e.g. Shepard, 2002; Birner & Boenisch, 2011).

P.2, L.28: In the text is written that the trajectories have been initialised between 0°E and 140°E longitude. In Fig. 1, the initialisation is all around the globe (0°E-360°E). What is actually the correct initialisation: figure or text?

P.3, L.3-5: "*The distribution of some of the particles ...*"
I can hardly see the described feature in Fig. 2. The data should be presented in a different way to illustrate this more clearly.

P.3, L.6-9 & Table 1: The criteria for the classification of the air mass origins (k=1-5) seems to me somehow uncomplete or ambiguous:

1.) How trajectories are classified, if they satisfy the criteria < 350 K, < 4 km and 20°N < lat. < 30°N? Are they counted as tropical troposphere (k=2) or as mid-latitude LT (k=3)?

2.) How trajectories are classified, if they satisfy the criteria > 380 K, lat. < 45° N and pot. vorticity > 6 PVU? Are the counted as UTLS (k=5)? This would mean that backward trajectories initialised e.g. at 15-16 km geopotential height and lat. = 45°N which has travelled up- and equatorward would be counted as UTLS (see Fig. 2 right panel all points south of 45°N and above 15 km). To my feeling, some trajectories that should be assigned to the shallow branch (k=1b) of the BDC are classified here as UTLS (k=5).
Another problem of this UTLS criteria (k=5) in combination with the 90-days backward trajectory limitation is that the mixing ratios of the tracers assigned to this category or region spanning from > 350 K in the tropics or > 4 km in the extratropics up to 25 km for lat. > 45°N are very different. For example CO, mixing ratios ranging from > 100 ppb (extratropical UT) to < 20 ppb (stratospheric values for lat. < 45°N) are condensed into the original UTLS time series needed to reconstruct the observations. The consequence is that this original time series is mainly defined by the fact where in the UTLS the trajectory stemmed from.

P.3, L.22: The AoA definition in this paper is different to that of Hall&Plumb (1994). Here also for purely tropospheric transport AoA values are calculated, i.e. from the UT to the lower extratropical troposphere (< 4 km) or the tropical troposphere < 350 K potential temperature. Hall and Plumb defined an only stratospheric AoA using the tropopause as the reference surface. However, the AoA defined here is closer to the AoA derived from tracer measurements, e.g. $SF_6$, for which the reference surface is in most cases and for practical reasons the tropical lower troposphere. This should be mentioned and clarified somehow, because tropospheric AoA are not really common.

P.3, L.26-30: It is not evident, if the underestimation of AoA found by Inai (2018) in the mid-latitude stratosphere holds for the UTLS. This could be evaluated with AoA derived from the $SF_6$ CONTRAIL observation in the ExUTLS. This issue is briefly discussed in section 4.3 and it is implicitly shown in Fig. 12f, but it would be much clearer, if the authors would show a figure with $SF_6$-derived AoA vs. 10-years backward trajectories derived AoA. This issue is of high interest (too short transport timescales into the stratosphere for ERA-Interim driven trajectories) and it also would have implications for the interpretation of chemical active tracers, for which the exposure time to stratospheric photochemistry is of interest.

P.4, L.10: No chemical decay during transport from the origin to the initial position during the 90 days of transport is included – this is definitely not true for CO (see also the general comments above).

P.4, L.19-27: Would it not be more consistent to use higher temporal resolved reference data from the NOAA/ESRL atmospheric baseline observatories for the definitions of the tropospheric time series? You already use the Barrow site (BRW) together with the Summit site (SUM, downgraded to a sampling site) to define high-latitude (lat. $> 45°N$) lower troposphere (k=4) time series. The airborne measurements at 11 km between $10°N$ and $30°N$ could be used to evaluate the differences between the remote tropical LT and the flight level.

P.5, L.3: It is hard to believe that this equation system is not under-determined. At least there should be some auxiliary constraints, e.g. mixing ratio X for a tracer with stratospheric sink should be lower for high-latitude stratosphere (k=1) than for the UTLS (k=5), i.e. X(k=1) < X(k=5).
How the minimisation of the equation 4 has been technically performed? With a simple but robust parameter sweep or with a more sophisticated (but maybe numerical more instable) algorithm? This is to my opinion quite essential for the outcome of this paper. Therefore, this (inverse) procedure and the sensitivity of the results to the choice of parameters should be explained and shown in more details (see also general comments).

P.5, L.13-15: This means that you exclude most or at least a large part of the upper tropospheric CONTRAIL data, because CO > 80 ppb is not a spurious event in the extratropical UT of the northern hemisphere (e.g. Engel et al., 2006 and references within), especially during winter and spring. Sometimes, it would be better to use tropopause related coordinates (or filter) instead of exclusively using equivalent latitude-potential temperature coordinates.

P.6, L.1-2: This finding is different from the results of Hoor et al. (2005) and Boenisch et al. (2009). Are there any explanation for these differences? I would expect that there is a certain time lag between the time of maximal downwelling (winter) and the maximal stratospheric characteristic of the LMS (spring).

P.7, L.1-2: The seasonality and the mixing ratios for CO in the high-latitude stratosphere (k=1) and UTLS (k=5) are unrealistic, especially for spring (see e.g. Tilmes et al., 2010 and references within). This is to my view the consequence that errors, e.g. missing chemistry, in the reconstruction of ExUTLS observations will be compensated by the reconstructed original time series of CO for the regions k=1 and 5.

P.7, L.5-10 & Figure 7: Why does the seasonality of AoA and $SF_6$ differ, especially in the UTLS (see Fig. 7d vs. 7f)? During August, the phase of the oldest AoA in the UTLS, one would expect the lowest (detrended) $SF_6$ mixing ratios. This seems here not to be the case,

August corresponds to the season with the highest (detrended) $SF_6$ mixing ratios, equivalent to the youngest AoA. What is the explanation for this contradiction?

P.7, L.19-21, & Figure 8-11: For me, it looks like April is simply the month with the most stratospheric characteristic of the LMS – highest AoA and lowest mixing ratios of the chemical active tracers above the 4pvu-contour.

P.7, L.29 & Figure 10: Why does $SF_6$, as a proxy for AoA, not show, in contrast to $N_2O$ and $CH_4$, the minima at 370 K in the ExUTLS region (see Fig 10 a+b+d)?

P.8, L.6-7 & Figure 10-11: *"The distribution of AoA during this season (autumn, comment by the reviewer) is similar to that during summer, with the AoA of nearly the entire region with potential vorticity of < 8 PVU being less than 1 year."*
The seasonality of AoA here is different to that found by Boenisch et al. (2009). They found the minimum in AoA in October with AoA below 0.5 years for most of the LMS (< 8pvu). What is the explanation for the difference in the seasonality found in the study here?

P.8, L.16-17: A direct comparison of AoA derived from $SF_6$ and backward trajectories would be better (see comment: P.3, L.26-30). How does the contradiction of different seasonality of $SF_6$ and AoA (derived from trajectories) in the UTLS fit to this result (see comment: P.7, L.5-10 & Figure 7)?

P.8, L.29-30: How you confirm the impact of the Asian Monsoon (ASM) with your study? Do you use an algorithm marking and detecting ASM air, e.g. like Vogel et al. (2016)?

P.9, L.6 & Figure 13c: *"During winter, however, tropical tropospheric air masses dominate."*
This is true not only during winter, but also during spring (until the beginning of May).

P.9, L.10:*"In the high-latitude lower ExUTLS, mixing fractions of the mid- and high-latitude LT are enhanced but their fractions are lower than those of the mid-latitude lower ExUTLS."*
What is the reason? More of the trajectories started in the stratosphere in high- compared to mid-**equivalent** latitude lower ExUTLS (PV>2pvu at the initial starting point)?
A tropopause related analysis would help here to understand how much of the effect here is related to the starting location (UT or LMS) and how much is related to weaker uplift into the UT in mid- compared to high-**equivalent** latitudes.
Equivalent latitude might be not the optimal coordinate in the troposphere. In contrast to the stratosphere, where PV is dominated by the strong stratification (dTheta/dz), in the troposphere PV is dominated by relative vorticity. A consequence is that e.g. WCBs in the UT assigned with high tropospheric PV values (e.g. Madonna et al., 2014) would be classified as high-**equivalent** latitude air mass.

P.9, L.11-13 & Fig. 13+14: The seasonal pattern in Fig 13 and 14 has not the same pattern for species with strongly varying original time series in the different compartments (k=1-5), because the reconstructed time series shown in Fig 14 are a superposition of the mixing fractions shown in Fig. 13 with the original time series for k=1-5 of the individual tracers. The difference in the seasonal pattern between Fig 13 and Fig 14 is most obvious for $CO_2$ which has a strong tropospheric seasonal cycle that is superimposed on the tropospheric mixing fractions.

P.9, L.33: Please add here the reference to Hoor et al. (2004)

P.10, L.20 & Fig. 15: The tracer-tracer relationships or "mixing lines" (AoA is a tracer like e.g. $SF_6$) for sufficient long-lived tracers (chemical lifetime must be greater than at least the horizontal transport timescale) is in theory the consequence of sufficiently rapid mixing along isentropic surfaces. Please cite here the review by Plumb (2007) which includes many of the references to the pioneering works on this topic in the 80s and 90s.

P.10, L.21-23 & Fig. 15: $CO_2$ mixing ratios and AoA does not correlate below a level of about 3 years AoA, because the propagated signal of the tropospheric seasonal cycle into the stratosphere is still detectable (not smeared out over a broad enough age spectra covering several seasonal cycles). This is the simple reason why $CO_2$ mixing ratios in the LMS cannot be used to calculate AoA (see e.g. Engel et al., 2002; Boenisch et al., 2009).

P.10, L.28-30 & Fig. 15f: "*Figure 15f shows seasonal variations in AoA and integrated PDF from 0 to 10 years for air masses originating in the high-latitude stratosphere*"
Fig. 15f only shows integrated PDFs for AoA from 0 to 6 years – please correct this.

P.12, L.3-4: The young-bias of AoA derived from backward trajectories in the LS should be verified (see my comment P.3, L.26-30 above)

P.12, L.8-9: Please show this (see comment above).

P.12, L.15-17: "*Moreover, these estimates are indirectly validated by the CONTRAIL observations, through the reconstruction of the chemical distributions (as evident in Figs 8–11).*"
This is only partially true, because you have a kind of free parameters, these are the original mixing ratio from the deep stratosphere (k=1) and the mixing ratios resided in the ExUTLS (k=5) during the 90 days of the backward trajectory simulation. Herewith, the interaction between CONTRAIL observed and trajectory-based mixing ratios can be adjusted. This is to my opinion most obvious for CO. The estimated CO for k=1 and 5 compensates other errors, e.g. chemical decay of CO during the transport from source region to the ExUTLS (see also other specific and general comments above).

P.12, L.18-24: Both problems discussed here briefly, non-linear tropospheric trend and the lack of agreement in reconstruction of CONTRAIL observations during summer (Fig. 10e), concern mainly $CO_2$, so please clarify and mention this here.

P.12, L.24-26: It is true that the equivalent latitude-potential temperature (EqLat-Theta) coordinate system accounts for dynamical features in the stratosphere, because adiabatic motion is dominant in this strongly stably stratified region of the atmosphere. This is not true for the troposphere which is much more unstable (low static stability). Potential temperature (and PV) are not conserved or only for a much shorter timescale, because diabatic motion is much more prominent there. Therefore, potential temperature and equivalent latitude are not the coordinate system of choice in the troposphere. Also the problem of tracer uplift from the PBL into the UT during summer (most prominent for $CO_2$, see above) is not minimised in an EqLat-Theta coordinate system.

P.13, L.7: The mentioned role of Asian monsoon (ASM) is very likely, but it is speculation here, because it is not shown in this study, how much of the trajectories originated from the ASM (see also comment above).

P.13, L.15: "*The reconstructions agree well with CONTRAIL measurements in the ExUTLS.*"

If this is one key messages of the summary then the limitations of the 90-days backward trajectories and the sensitivity of the results due to this limitation have to be discussed in much more detail (see general and specific comments above).

P.13, L.23-24:"*This method provides a means to understand both dynamical transport and chemical distribution from a new perspective.*"
There has been done a lot to understand dynamical, tracer transport and chemical processes in the UTLS. Some of these studies has been mentioned in this review and should be discussed in relation to the results in this manuscript. As outlined in my general comment, I am not convinced that the actual manuscript could contribute to the actual state of knowledge, but the results should be at least discussed in this framework. The uniqueness of this approach here, combination of different tracers and backward trajectories, could to my opinion be exploited much better, if one would use 10-years instead of 90-days backward trajectories.

**References:**

Appenzeller, C., J. R. Holton, and K. H. Rosenlof (1996), Seasonal variation of mass transport across the tropopause, J. Geophys. Res., 101(D10), 15071-15078.

Birner, T., and H. Bönisch (2011), Residual circulation trajectories and transit times into the extratropical lowermost stratosphere, *Atmos. Chem. Phys.*, *11*(2), 817-827, doi:10.5194/acp-11-817-2011.

Bönisch, H., A. Engel, J. Curtius, T. Birner, and P. Hoor (2009), Quantifying transport into the lowermost stratosphere using simultaneous in-situ measurements of $SF_6$ and $CO_2$, Atmos. Chem. Phys., 9(16), 5905-5919, doi:10.5194/acp-9-5905-2009.

Diallo, M., Legras, B., and Chédin, A. (2012): Age of stratospheric air in the ERA-Interim, Atmos. Chem. Phys., 12, 12133-12154, https://doi.org/10.5194/acp-12-12133-2012, 2012.

Diallo, M., B. Legras, E. Ray, A. Engel, and J. A. Añel (2017), Global distribution of $CO_2$ in the upper troposphere and stratosphere, Atmos. Chem. Phys., 17(6), 3861-3878, doi:10.5194/acp-17-3861-2017.

Engel, A., M. Strunk, M. Muller, H. P. Haase, C. Poss, I. Levin, and U. Schmidt (2002), Temporal development of total chlorine in the high-latitude stratosphere based on reference distributions of mean age derived from $CO_2$ and $SF_6$, *J. Geophys. Res.*, *107*(D12), doi:Artn 4136

Doi 10.1029/2001jd000584.

Engel, A., et al. (2006), Highly resolved observations of trace gases in the lowermost stratosphere and upper troposphere from the Spurt project: an overview, Atmos. Chem. Phys., 6, 283-301, doi:DOI 10.5194/acp-6-283-2006.

Fueglistaler, S., H. Wernli, and T. Peter (2004), Tropical troposphere-to-stratosphere transport inferred from trajectory calculations, Journal of Geophysical Research: Atmospheres, 109(D3), doi:10.1029/2003JD004069.

Fueglistaler, S., M. Bonazzola, P. H. Haynes, and T. Peter (2005), Stratospheric water vapor predicted from the Lagrangian temperature history of air entering the stratosphere in the tropics, Journal of Geophysical Research: Atmospheres, 110(D8), doi:10.1029/2004JD005516.

Fueglistaler, S., and P. H. Haynes (2005), Control of interannual and longer-term variability of stratospheric water vapor, Journal of Geophysical Research: Atmospheres, 110(D24), doi:10.1029/2005JD006019.

Gettelman, A., P. Hoor, L. L. Pan, W. J. Randel, M. I. Hegglin, and T. Birner (2011), The Extratropical Upper Troposphere and Lower Stratosphere, Rev. Geophys., 49, doi:Artn Rg3003, Doi 10.1029/2011rg000355.

Hall, T. M., and R. A. Plumb (1994), Age as a Diagnostic of Stratospheric Transport, *J. Geophys. Res.*, *99*(D1), 1059-1070.

Hoor, P., C. Gurk, D. Brunner, M. I. Hegglin, H. Wernli, and H. Fischer (2004), Seasonality and extent of extratropical TST derived from in-situ CO measurements during SPURT, Atmos. Chem. Phys., 4, 1427-1442.

Hoor, P., H. Fischer, and J. Lelieveld (2005), Tropical and extratropical tropospheric air in the lowermost stratosphere over Europe: A CO-based budget, Geophys. Res. Lett., 32(7), L07802, doi:10.1029/2004gl022018.

Hoor, P., H. Wernli, M. I. Hegglin, and H. Boenisch (2010), Transport timescales and tracer properties in the extratropical UTLS, Atmos. Chem. Phys., 10(16), 7929-7944, doi:10.5194/acp-10-7929-2010.

Madonna, E., H. Wernli, H. Joos, and O. Martius (2014), Warm Conveyor Belts in the ERA-Interim Dataset (1979–2010). Part I: Climatology and Potential Vorticity Evolution, *Journal of Climate*, 27(1), 3-26, doi:10.1175/jcli-d-12-00720.1.

Ploeger, F., and T. Birner (2016), Seasonal and inter-annual variability of lower stratospheric age of air spectra, Atmos. Chem. Phys., 16(15), 10195-10213, doi:10.5194/acp-16-10195-2016.

Plumb, R. A. (2007), Tracer interrelationships in the stratosphere, *Rev. Geophys.*, 45(4), Artn Rg4005, doi:10.1029/2005rg000179.

Ray, E. A., F. L. Moore, J. W. Elkins, G. S. Dutton, D. W. Fahey, H. Vomel, S. J. Oltmans, and K. H. Rosenlof (1999), Transport into the Northern Hemisphere lowermost stratosphere revealed by in situ tracer measurements, J. Geophys. Res., 104(D21), 26565-26580.

Riese, M., F. Ploeger, A. Rap, B. Vogel, P. Konopka, M. Dameris, and P. Forster (2012), Impact of uncertainties in atmospheric mixing on simulated UTLS composition and related radiative effects, J. Geophys. Res., 117(D16), D16305, doi:10.1029/2012jd017751.

Schoeberl, M. R., L. C. Sparling, C. H. Jackman, and E. L. Fleming (2000), A Lagrangian view of stratospheric trace gas distributions, Journal of Geophysical Research: Atmospheres, 105(D1), 1537-1552, doi:10.1029/1999JD900787.

Shepherd, T. G. (2002), Issues in Stratosphere-troposphere Coupling, Journal of the Meteorological Society of Japan. Ser. II, 80(4B), 769-792, doi:10.2151/jmsj.80.769.

Škerlak, B., M. Sprenger, and H. Wernli (2014), A global climatology of stratosphere–troposphere exchange using the ERA-Interim data set from 1979 to 2011, Atmos. Chem. Phys., 14(2), 913-937, doi:10.5194/acp-14-913-2014.

Tilmes, S., et al. (2010), An aircraft-based upper troposphere lower stratosphere O3, CO, and H2O climatology for the Northern Hemisphere, *Journal of Geophysical Research: Atmospheres*, 115(D14), doi:10.1029/2009JD012731.

Vogel, B., et al. (2016), Long-range transport pathways of tropospheric source gases originating in Asia into the northern lower stratosphere during the Asian monsoon season 2012, *Atmos. Chem. Phys.*, 16(23), 15301-15325, doi:10.5194/acp-16-15301-2016.

---

## Referee Comment (RC2) · Anonymous Referee #2 · 2 Jan 2019

The paper by Inai et al. investigates the air mass composition of the extratropical upper troposphere and lower stratosphere (exUTLS), and relates to CONTRAIL in-situ observations of several trace gas species (e.g., CH4, N2O, SF6, CO, CO2). The focus of the study lies on seasonal variations in air mass fractions and mixing ratios. In particular, it is found that seasonality in CH4, N2O and SF6 mixing ratios is controlled by transport from the deep stratosphere, due to the locations of the main chemical sink regions, whereas CO and CO2 are mainly controlled by transport from the tropical troposphere.

The air mass and tracer composition of the exUTLS is of particular relevance for global

climate due to the radiative characteristics of this region. Hence, the present study fits well into the scope of ACP. The paper is well written and presented, and the current literature is appropriately discussed. I recommend publication after taking into account the several comments below, which I regard somewhere between major and minor.

Detailed comments:

1. Initialization: The trajectory initialization is somewhat unclear to me. In the respective text part it is said, that back trajectories are initialized between 0-140 deg E, but the corresponding Fig. 1 shows initialization locations for 0-360 deg E (P2/L27). How is the initialization done exactly?

2. Model-measurement comparison: The CONTRAIL measurements are mainly from Siberia. How is the model-measurement comparison done, exactly at the measurement locations/times, or just averaged over specific regions? I would suggest to explain this clearly directly after the description of the trajectory initialization (P2).

3. Reconstruction method: It would be good to mention (around P4/L10) that Eq. (2) holds only for species which are chemically inert along the trajectories. Can you give some quantitative information how well this assumption holds for the species and regions considered here? Perhaps some of the difference between reconstruction and measurements (e.g., Figs. 7-10) is related to neglecting chemistry effects?

4. Origin mixing ratios (P4/L28): Why not using higher altitude in-situ measurements (e.g., from balloons, Geophysica/Halo/ER2/... aircrafts) or global satellite observations for the reference mixing ratios? At least the "inversion method" outlined below could be validated with such data.

5. Minima in tracer distributions around 370K in spring/summer (P7/L21ff): I do not think these minima are just artifacts of the reconstruction. The fact that spring/summer transport of young tropical air strengthens first around 380-400K, leading to a "sandwich" structure with older air masses below is consistent with recent findings by Krause

et al. (2018) (see e.g. their Fig. 14) and Ploeger and Birner (2016) (e.g., their Fig. 7). In agreement with these papers, Fig. 9/10 show evidence for strongest polward transport above about 380K, causing the mixing ratio minima below. I would suggest to discuss these distributions more appropriately.

6. Trajectory method: Kinematic trajectories show stronger dispersion compared to diabatic trajectories (e.g., Schoeberl et al., 2003). Are the results presented here robust also for diabatic transport? At least include appropriate discussion in Sect. 4.3 ("Limitations of the current study").

Specific and technical comments:

P1/L29: maybe better "at/along the subtropical jet"?

P3/L23: "...where IT satisfies..."?

P3/L28: What is the "actual value" what is referred to here? Observations? Which?

P7/L29: ware –> were

P9/L10ff: The sentence "In addition ..." sounds unclear to me - I suggest rewording.

P9/L19: shown –> show

P10/L28ff: I don't understand the description of Fig. 15f. What PDF is integrated here (transit time pdf?). What is the unit of the y-axis? Please clarify and improve the description.

P12/L7: The Ploeger and Birner reference cited here is not in the reference list.

P12/L20: non-linear

References:

Krause et al. (2018), Atmos. Chem. Phys., 18, 6057-6073.

Schoeberl et al. (2003), J. Geophys. Res, 118, D3.

---

## Author Comment (AC1) · 27 Feb 2019

GENERAL REPLY

The authors thank the two reviewers for their constructive comments and helpful suggestions that have improved the manuscript.

As mentioned by the both reviewers, the reconstruction method had some shortcomings, but the reviewers, especially reviewer #1, gave concrete and great suggestions to resolve them. The manuscript has been largely revised and improved along the suggestions. The authors describe first the revisions in the reconstruction method as a General Reply.

The suggestions are as follows:
(1) use of 10-year trajectory instead of 90-day, which can eliminate the remaining trajectories in the UTLS (k=5);
(2) reconstruction of chemical passive tracers with evaluation of transport timescale (as the first step); and
(3) reconstruction of chemical active tracers including chemical decay (as the second step).
The authors have made these procedures and some relating revisions as follows.

For point (1), as noted by the reviewers, the remaining trajectories in the UTLS (k=5) could be eliminated, i.e., it was confirmed that the all trajectories are categorized into any origins of k = 1, 2, 3, or 4 within 10 years (Note that the criteria have also been revised to avoid some shortcomings). In addition, the inversion method to estimate tracer mixing ratio for k=1 and 5 in the original manuscript could be also eliminated, i.e., the all tracers in the ExUTLS have been reconstructed only from their mixing ratios assumed in the high-latitude LT, mid-latitude LT, and tropical troposphere.

For point (2), using the 10-year trajectory, Age of Air (AoA) as well as SF6 and CO2 distributions have been reconstructed from the trajectories including the "Tail correction" (e.g., Diallo et al., 2012). The CH4, N2O, CO have been also reconstructed without any chemical decay in this step. AoA has been also estimated using observed SF6 mixing ratios obtained by CONTRAIL, and then the two AoAs have been compared to correct transport timescale expressed in the trajectories.

For point (3), the chemical active tracers, CH4, N2O, and CO, are finally reconstructed with simulating their chemical loss along an "average path" (Schoeberl et al., 2000). The use of the concept of average path was also suggested by the reviewer #1. Based on this concept, the authors believe that the active

tracers have been successfully reconstructed together with estimation of seasonally depending their chemical loss rate.

In relation to a suite of these revisions, the latter half of section 2.1 (Estimating the origin fraction and AoA), large part of section 2.2 (Air mass original composition and reconstruction) were significantly revised, especially section 2.2 was reorganized and a new subsections 2.2.1 and 2.2.2 were created for reconstructions of chemical passive and active tracers, respectively. The analyzing results, figures, and discussions were thus also changed in association with the revision of the reconstruction method, but the main thesis was essentially not changed.

References:

Diallo, M., Legras, B., and Chédin, A.: Age of stratospheric air in the ERA-Interim, Atmos. Chem. Phys., 12, 12133-12154, https://doi.org/10.5194/acp-12-12133-2012, 2012.

Schoeberl, M. R., Sparling, L. C., Jackman, C. H., and Fleming, E. L.: A Lagrangian view of stratospheric trace gas distributions, Journal of Geophysical Research: Atmospheres, 105(D1), 1537-1552, doi:10.1029/1999JD900787, 2000.

The authors believe that the revised manuscript has been improved by incorporating the more appropriate reconstruction method.

Point-by-point responses to the comments of individual reviewer are provided below.

REPLY TO COMMENTS BY REVIEWER #1

The authors are grateful for the thorough review and constructive comments on the manuscript. All of the points raised by the reviewer have been addressed. Regarding the reviewer's major comments, please also refer to the "General Reply" section. Point-by-point responses are detailed below, in red text.

Reviewer (Comments):
Review of "Seasonal characteristics of chemical and dynamical transports into the extratropical upper troposphere/lower stratosphere" (ExUTLS) by Yoichi Inai et al.

Recommendation: Publication after major revision
The paper is very well organised and written. The topic discussed in this paper, transport into the ExUTLS, is in general of high relevance. Our actual limitations in simulating water vapour transport in this complex region introduce large uncertainties in the Earth radiation budget (see Riese et al., 2012). Trajectory analysis in combination with observations could be and have been used in many cases to improve our knowledge on tracer transport and distribution in the UTLS, e.g. for H2O: Fueglistaler et al. (2004, 2005a, 2005b), e.g. for CO and H2O: Hoor et al. (2010), e.g. for STE (stratosphere-troposphere exchange) and O3: Skerlag et al. (2014), e.g. for CO2 and AoA: Diallo et al. (2012, 2017). This manuscript here falls a bit short of explaining what the novel aspect of the presented method really is and how the presented results augment our actual knowledge on the seasonal characteristics of transport in and into the ExUTLS, e.g. in comparison to the early studies by Appenzeller et al. (1996) and Ray et al. (1999) or to the many studies summarised in the ExUTLS review paper by Gettlemann et al. (2011).

The paper should be submitted after addressing the comments below.

General comments:
First of all, I don't fully understand the title (and/or the scope) of this paper. What is the meaning of chemical and dynamical transport? Is chemical transport a synonym for transport of chemical active tracers (N2O, CH4 and CO)? Should dynamical transport describe the transport of passive tracers (SF6, CO2 and AoA)? Pure Lagrangian transport (here backward

trajectories) differs from (both) tracer transports: There is no mixing and no chemistry included along the individual transport pathways.

What the authors meant to write by the phrase "chemical and dynamical transports" is 1) transport of chemical species and 2) transport of air mass which is expressed by the mixing fractions of air mass originating in the stratosphere, tropical troposphere, mid-latitude LT, and high-latitude LT. In order to make it clearer, the title has been changed to "Seasonal characteristics of trace gas transport into the ExUTLS."

The latter is definitively a problem for CO, because the chemical decay along the 90-days backward trajectories cannot be neglected. For N2O and CH4 the chemical decay is not that significant, because only a few of the initialised trajectories will travel through the sink regions of both tracers higher up in the stratosphere during the 90-days. However, the unknown (non-observed) time series of the high-latitude stratospheric background (k=1) and the tropical and extratropical UTLS background (k=5) conditions for all tracers (not only for the chemical active tracers N2O and CH4) still remain the major problem for the reconstruction of the observed tracer distributions by CONTRAIL using only 90-days backward trajectories. The reconstruction in the chosen setup could not be used for quantitative studies of neither ERA-Interim (or other reanalysis data sets) nor the transport processes in the ExUTLS, because the trajectories itself are needed to define the boundary conditions for the original time series $XS\_ORG\_k$ in the high-latitude stratosphere (k=1) and the UTLS (k=5) which are again the prerequisite to reconstruct the observed mixing ratios in the ExUTLS derived from CONTRAIL. This is a circular reference between the trajectory analysis and the CONTRAIL observations in the ExUTLS, whereby the non-observed (inverse reconstructed) original time series for k=1 and 5 could be seen as a kind of free parameters to tune the system or in other words to close the budget for the individual tracers.

The main problem, why this paper could, to my point of view, not add to the actual state of knowledge, although it has the potential, is the limitation of the backward trajectories to 90 days. The consequence of this limitation is the circular reference explained above (the authors call this an inversion technique) that has to be introduced to reconstruct the original time series of the tracer mixing ratios in the stratospheric overworld (k=1) and in the UTLS (k=5).

The authors claim that the mixing fractions derived from the coarser resolution ERA-Interim data (1.5x1.5, 37 levels) are the same as for the finer resolution ERA-Interim data (0.75x0.75, 60 levels). If this is the case, why not using the 10-year instead of the 90-days backward

trajectories? This would at least solve the problem to reconstruct the non-observed boundary conditions for the trajectories residing in the ExUTLS (k=5) during the 90 days and also partially for the trajectories originating from the stratospheric overworld (k=5). The latter is unfortunately only true for the passive tracers, SF6 and CO2 respectively. Both tracers could be reconstructed by combining their original tropospheric time series (k=2,3,4) and the 10-years trajectories all starting in the target region of this study – the ExUTLS – and ending in the troposphere (e.g. Diallo et al., 2017), beside a small residuum of trajectories that remains in the stratosphere and that has to be characterised (see e.g. Ploeger et al, 2016).

The strong point of this study here is to my opinion the combination of backward trajectories driven by a state-of-the-art reanalysis data set (ERA-Interim) with simultaneous measurements in the ExUTLS of five tracers with different characteristics in their lifetimes and tropospheric time series. My recommendation would be to separate the analysis on transport and chemical processes. In the first step, one could use the 10-years backward trajectories (if manageable, it would be better using the high resolution ERA-Interim data) together with the passive tracers CO2 and SF6 (the latter is literally the same as AoA in the UTLS) to evaluate the mixing fractions and transport timescales derived from the ERA-Interim driven backward trajectories by reconstructing the CONTRAIL observations of both passive tracers. This is already a valuable extension to the method shown by Diallo et al. (2017), because the additional simoultaneous SF6 observations are a second independent constraint for the evaluation due to the different (linearly independent) tropospheric time series compared to CO2.

In the next step, one could exploit the additional information from the simultaneously measured chemical active tracers. Given the transport properties (mixing fraction, air mass origin and timescale) that has been quantified and evaluated in the first step, the chemical decay along the transport pathways from the tropospheric origin into the ExUTLS could be analysed with the simultaneous measurements of the chemical active tracers CO, CH4 and N2O. The difference between the reconstructed passive CO, CH4 and N2O tracers without chemistry and observed values including chemical decay should allow to assign a photochemical loss for air parcels along an "average pathway" with the same AoA (see Schoeberl et al., 2000). This "average pathway" could be defined by a bulk of trajectories, e.g. by the trajectories in a given equivalent latitude-potential temperature bin. There might be many more and better approaches to derive quantitative information on the chemical decay along the transport pathways, but the huge advantage in general of using backward trajectories together with simoultaneous measured passive and chemical active tracers would be that one could disentangle dynamical and chemical effects on the observed tracer

distribution in a unique way. An urgent question that has to be answered to understand the processes driving and driven by climate change in this complex and important region of the atmosphere.

If the authors decide to stay with 90-days backward trajectory setup then the limitations of the 90-days backward trajectories and the sensitivity of the results due to these limitations have to be discussed in much more detail – see also the specific comments below.

Thank you for this great constructive suggestion on how to reconstruct both the chemically passive and active compositions. Following above suggestions, the reconstruction method has been revised, and the authors believe that it has been largely improved. Please see the General Reply.

Specific comments:
P.1, L.26: "*…, especially in the Arctic climate.*"
Please cite references for this statement or delete it.

The statement has been deleted.

P.1, L.27: "*… via stratospheric residual circulation (Brewer-Dobson circulation, BDC; Brewer, 1949; Dobson, 1956).*"
The stratospheric residual circulation describes the mean mass transport. The BDC includes mean mass transport and two-way mixing. The latter, by definition, does not lead to net mass exchange but may lead to net tracer exchange. Therefore, I would suggest to use stratospheric circulation instead of stratospheric residual circulation as a synonym for the BDC (see e.g. Shepard, 2002; Birner & Boenisch, 2011).

Thank you very much for informative comments. Following this comment, the statement "stratospheric circulation" has been used, instead of "stratospheric residual circulation."

P.2, L.28: In the text is written that the trajectories have been initialised between 0° E and 140° E longitude. In Fig. 1, the initialisation is all around the globe (0° E-360° E). What is actually the correct initialisation: figure or text?

Fig. 1 in the original manuscript was not correct. It has been corrected.

P.3, L.3-5: "*The distribution of some of the particles …*"
I can hardly see the described feature in Fig. 2. The data should be presented in a different way to illustrate this more clearly.

In association with the revision of the method using 10-year trajectory, the figure and descriptions for it have been changed.

P.3, L.6-9 & Table 1: The criteria for the classification of the air mass origins (k=1-5) seems to me somehow uncomplete or ambiguous:

1.) How trajectories are classified, if they satisfy the criteria < 350 K, < 4 km and 20° N < lat. < 30° N? Are they counted as tropical troposphere (k=2) or as mid-latitude LT (k=3)?

The trajectories had been categorized either compartment which was satisfied first. The authors confess that it was not appropriate, so the criteria have been revised to avoid such overlap as pointed out by this comment. The revised criteria have been shown in Table 1 in the revised manuscript.

2.) How trajectories are classified, if they satisfy the criteria > 380 K, lat. < 45° N and pot. vorticity > 6 PVU? Are the counted as UTLS (k=5)? This would mean that backward trajectories initialised e.g. at 15-16 km geopotential height and lat. = 45° N which has travelled upand equatorward would be counted as UTLS (see Fig. 2 right panel all points south of 45° N and above 15 km). To my feeling, some trajectories that should be assigned to the shallow branch (k=1b) of the BDC are classified here as UTLS (k=5).

Yes, they had been counted as k=5. Following this comment, the criteria for stratosphere (k=1) has been also revised as shown in Table 1 in the revised manuscript.

Another problem of this UTLS criteria (k=5) in combination with the 90-days backward trajectory limitation is that the mixing ratios of the tracers assigned to this category or region spanning from > 350 K in the tropics or > 4 km in the extratropics up to 25 km for lat. > 45° N are very different. For example CO, mixing ratios ranging from > 100 ppb (extratropical UT) to < 20 ppb (stratospheric values for lat. < 45° N) are condensed into the original UTLS time series needed to reconstruct the observations. The consequence is that

this original time series is mainly defined by the fact where in the UTLS the trajectory stemmed from.

As described in the General Reply, 10-year trajectory has been employed in addition to the revision of the classification of original regions. The authors believe that the problem pointed out here has been fixed.

P.3, L.22: The AoA definition in this paper is different to that of Hall&Plumb (1994). Here also for purely tropospheric transport AoA values are calculated, i.e. from the UT to the lower extratropical troposphere ($< 4\,\mathrm{km}$) or the tropical troposphere $< 350\,\mathrm{K}$ potential temperature. Hall and Plumb defined an only stratospheric AoA using the tropopause as the reference surface. However, the AoA defined here is closer to the AoA derived from tracer measurements, e.g. SF6, for which the reference surface is in most cases and for practical reasons the tropical lower troposphere. This should be mentioned and clarified somehow, because tropospheric AoA are not really common.

To mention this point, the sentence "Thus, the AoA definition used here differs from that of Hall and Plumb (1994), who defined AoA as the elapsed time an air parcel spends in the stratosphere after across the tropopause" has been added in P3, L24ff in the revised manuscript.

P.3, L.26-30: It is not evident, if the underestimation of AoA found by Inai (2018) in the midlatitude stratosphere holds for the UTLS. This could be evaluated with AoA derived from the SF6 CONTRAIL observation in the ExUTLS. This issue is briefly discussed in section 4.3 and it is implicitly shown in Fig. 12f, but it would be much clearer, if the authors would show a figure with SF6-derived AoA vs. 10-years backward trajectories derived AoA. This issue is of high interest (too short transport timescales into the stratosphere for ERA-Interim driven trajectories) and it also would have implications for the interpretation of chemical active tracers, for which the exposure time to stratospheric photochemistry is of interest.

As described in the General Reply, the 10-year trajectory has been employed and the SF6-derived AoA is compared with traj-derived AoA to correct them.

P.4, L.10: No chemical decay during transport from the origin to the initial position during the 90 days of transport is included – this is definitely not true for CO (see also the general comments above).

Chemical decay of CO as well as the other species have been included as described in the General Reply.

P.4, L.19-27: Would it not be more consistent to use higher temporal resolved reference data from the NOAA/ESRL atmospheric baseline observatories for the definitions of the tropospheric time series? You already use the Barrow site (BRW) together with the Summit site (SUM, downgraded to a sampling site) to define high-latitude (lat. > 45° N) lower troposphere (k=4) time series. The airborne measurements at 11 km between 10° N and 30° N could be used to evaluate the differences between the remote tropical LT and the flight level.

The authors consider that there are two attitudes to incorporate such data into this analysis. The one is use of higher temporal resolved data as you pointed out, and another is use of larger special representative data. The authors have conducted such aircraft observations by ourselves and accumulate such data which have larger special representativeness. We choose own larger special representative data to use.

P.5, L.3: It is hard to believe that this equation system is not under-determined. At least there should be some auxiliary constraints, e.g. mixing ratio X for a tracer with stratospheric sink should be lower for high-latitude stratosphere (k=1) than for the UTLS (k=5), i.e. X(k=1) < X(k=5).
How the minimisation of the equation 4 has been technically performed? With a simple but robust parameter sweep or with a more sophisticated (but maybe numerical more instable) algorithm? This is to my opinion quite essential for the outcome of this paper. Therefore, this (inverse) procedure and the sensitivity of the results to the choice of parameters should be explained and shown in more details (see also general comments).

The inversion method has been eliminated in association with revision of the reconstruction method.

P.5, L.13-15: This means that you exclude most or at least a large part of the upper tropospheric CONTRAIL data, because CO > 80 ppb is not a spurious event in the extratropical UT of the northern hemisphere (e.g. Engel et al., 2006 and references within), especially during winter and spring. Sometimes, it would be better to use tropopause related coordinates (or filter) instead of exclusively using equivalent latitude-potential temperature

coordinates.

The number of such measurements that CO > 80 ppb and Theta > 340 K and Lat_eq > 60deg N is not large. Such measurements have been identified by cross-marks in Figs. 4, 5, and 8 of the revised manuscript.

P.6, L.1-2: This finding is different from the results of Hoor et al. (2005) and Boenisch et al. (2009). Are there any explanation for these differences? I would expect that there is a certain time lag between the time of maximal downwelling (winter) and the maximal stratospheric characteristic of the LMS (spring).

The result has been changed in association with the use of 10-year trajectory, and it has become consistent with the results of Hoor et al. (2005) and Boenisch et al. (2009). The statement has been changed (P7, L19-22).

P.7, L.1-2: The seasonality and the mixing ratios for CO in the high-latitude stratosphere (k=1) and UTLS (k=5) are unrealistic, especially for spring (see e.g. Tilmes et al., 2010 and references within). This is to my view the consequence that errors, e.g. missing chemistry, in the reconstruction of ExUTLS observations will be compensated by the reconstructed original time series of CO for the regions k=1 and 5.

The estimation method of compositions in the stratospheric air mass (k=1) has been revised as described in the first part of Sect. 3.2 (P8, L11ff), in association with revision of the reconstruction method.

P.7, L.5-10 & Figure 7: Why does the seasonality of AoA and SF6 differ, especially in the UTLS (see Fig. 7d vs. 7f)? During August, the phase of the oldest AoA in the UTLS, one would expect the lowest (detrended) SF6 mixing ratios. This seems here not to be the case, August corresponds to the season with the highest (detrended) SF6 mixing ratios, equivalent to the youngest AoA. What is the explanation for this contradiction?

The estimation method of traj-derived AoA has been revised. The revised results show in-phase seasonality of AoA and SF6, the description has been changed (P8, L25-29 in the revised manuscript).

P.7, L.19-21, & Figure 8-11: For me, it looks like April is simply the month with the most

stratospheric characteristic of the LMS – highest AoA and lowest mixing ratios of the chemical active tracers above the 4pvu-contour.

That is right. The following description has been made here (P9, L6-10): "The reconstructions and AoA for April (Fig. 16) show spatial distributions of all species that generally increase with decreasing potential temperature, equivalent latitude, or potential vorticity, as is the case for January. However, the gradients are larger, particularly for $CH_4$ and $N_2O$ mixing ratios, such that in regions where the potential vorticity is >6 PVU the mixing ratios are much smaller than those in January, but in regions where the potential vorticity is <4 PVU the mixing ratios are almost the same as in January."

P.7, L.29 & Figure 10: Why does SF6, as a proxy for AoA, not show, in contrast to N2O and CH4, the minima at 370 K in the ExUTLS region (see Fig 10 a+b+d)?

Such "sandwich" structures have been commonly shown in Fig. 17 of the revised manuscript.

P.8, L.6-7 & Figure 10-11: "*The distribution of AoA during this season (autumn, comment by the reviewer) is similar to that during summer, with the AoA of nearly the entire region with potential vorticity of < 8 PVU being less than 1 year.*"
The seasonality of AoA here is different to that found by Boenisch et al. (2009). They found the minimum in AoA in October with AoA below 0.5 years for most of the LMS (< 8pvu). What is the explanation for the difference in the seasonality found in the study here?

The result has been changed in association with the use of 10-year trajectory, and it has become consistent with the results of Boenisch et al. (2009).

P.8, L.16-17: A direct comparison of AoA derived from SF6 and backward trajectories would be better (see comment: P.3, L.26-30). How does the contradiction of different seasonality of SF6 and AoA (derived from trajectories) in the UTLS fit to this result (see comment: P.7, L.5-10 & Figure 7)?

The comparison of traj-derived AoA and SF6-derived AoA has been made and the UTLS category has been removed.

P.8, L.29-30: How you confirm the impact of the Asian Monsoon (ASM) with your study? Do you use an algorithm marking and detecting ASM air, e.g. like Vogel et al. (2016)?

No, we simply confirm that the trajectories originating in Asian region increased in summer season. The statement has been corrected to "trajectories originating in the tropical troposphere over around Asia are strengthened." (P10, L10)

P.9, L.6 & Figure 13c: "*During winter, however, tropical tropospheric air masses dominate.*" This is true not only during winter, but also during spring (until the beginning of May).

Though the figure has been changed, but it remains true, so "and spring" has been added. (P10, L18)

P.9, L.10:"*In the high-latitude lower ExUTLS, mixing fractions of the mid- and high-latitude LT are enhanced but their fractions are lower than those of the mid-latitude lower ExUTLS.*" What is the reason? More of the trajectories started in the stratosphere in high- compared to mid-equivalent latitude lower ExUTLS (PV>2pvu at the initial starting point)? A tropopause related analysis would help here to understand how much of the effect here is related to the starting location (UT or LMS) and how much is related to weaker uplift into the UT in mid- compared to high-equivalent latitudes. Equivalent latitude might be not the optimal coordinate in the troposphere. In contrast to the stratosphere, where PV is dominated by the strong stratification (dTheta/dz), in the troposphere PV is dominated by relative vorticity. A consequence is that e.g. WCBs in the UT assigned with high tropospheric PV values (e.g. Madonna et al., 2014) would be classified as high-equivalent latitude air mass.

What the authors meant to here was "In the high-equivalent latitude lower (HL; such acronym has been used in the revised manuscript) ExUTLS, origin fractions of the mid- and high-latitude LT are enhanced during summer. Origin fractions of the high-latitude LT are comparable to those in the ML ExUTLS, but smaller than those of the mid-latitude LT in the HL ExUTLS."
Therefore, the literature has been revised (P10, L18-20), in addition, the following discussion has been added (P10, L20ff): "This can be explained by enhanced exchange at the bottom edge of the subtropical jet (i.e., along the 320–330 K surface for summer, e.g., Gettelman et al., 2011). As shown in Fig. 12d, enhanced origin fractions of the mid-latitude LT are distributed along such isentropes."

Relating this comment, the authors find that the phase "mid- latitude lower ExUTLS" should

be changed "mid-**equivalent** latitude lower ExUTLS," this revision has been also done.

P.9, L.11-13 & Fig. 13+14: The seasonal pattern in Fig 13 and 14 has not the same pattern for species with strongly varying original time series in the different compartments (k=1-5), because the reconstructed time series shown in Fig 14 are a superposition of the mixing fractions shown in Fig. 13 with the original time series for k=1-5 of the individual tracers. The difference in the seasonal pattern between Fig 13 and Fig 14 is most obvious for CO2 which has a strong tropospheric seasonal cycle that is superimposed on the tropospheric mixing fractions.

The statement in the original manuscript was not correct. The statement has been revised to "Figure 20 reveals that seasonal variations in the reconstructions for each species and the trajectory-estimated AoA in each of the four locations have patterns that differ because they are based on a superposition of the origin fractions shown in Fig. 19 with the original time series for $k = 1-4$ of the individual tracers shown in Fig. 14." (P10, L25-28)

P.9, L.33: Please add here the reference to Hoor et al. (2004)

It has been added (P11, L14).

P.10, L.20 & Fig. 15: The tracer-tracer relationships or "mixing lines" (AoA is a tracer like e.g. SF6) for sufficient long-lived tracers (chemical lifetime must be greater than at least the horizontal transport timescale) is in theory the consequence of sufficiently rapid mixing along isentropic surfaces. Please cite here the review by Plumb (2007) which includes many of the references to the pioneering works on this topic in the 80s and 90s.

Thank you very much for this informative comment. It has been cited and the following statement has been added (P12, L2-4): "Such linear "mixing lines" also suggest that the mixing took place rapidly (i.e., at a time-scale shorter than their chemical lifetimes) along an isentropic surface (Plumb, 2007 and references therein)."

P.10, L.21-23 & Fig. 15: CO2 mixing ratios and AoA does not correlate below a level of about 3 years AoA, because the propagated signal of the tropospheric seasonal cycle into the stratosphere is still detectable (not smeared out over a broad enough age spectra covering several seasonal cycles). This is the simple reason why CO2 mixing ratios in the LMS cannot be used to calculate AoA (see e.g. Engel et al., 2002; Boenisch et al., 2009).

Thank you for the instructive comment. CO2 is deleted in the sentence and the following statements has been added: "According to Engel et al. (2002) and Bönisch et al. (2009), the mixing ratios of $CO_2$ and AoA do not correlate below a level of ~3 years AoA because the propagated signal of the tropospheric seasonal cycle into the stratosphere is still detectable. In agreement with their results, the CONTRAIL $CO_2$ measurements also converge to the sign-reversed trend with increasing AoA." (P12, L5-8)

P.10, L.28-30 & Fig. 15f: "*Figure 15f shows seasonal variations in AoA and integrated PDF from 0 to 10 years for air masses originating in the high-latitude stratosphere*"
Fig. 15f only shows integrated PDFs for AoA from 0 to 6 years – please correct this.

The meaning of this sentence was that "The figure shows seasonal variations in AoA and the value that is calculated by integration of "age spectrum" (PDF) from 0 to *tf* for air masses originating in the stratosphere" The sentence has reworded (P12, L13-15).

P.12, L.3-4: The young-bias of AoA derived from backward trajectories in the LS should be verified (see my comment P.3, L.26-30 above)

It has been verified as described in the General Reply.

P.12, L.8-9: Please show this (see comment above).

It has been shown in Fig. 4 which has been newly made.

P.12, L.15-17: "*Moreover, these estimates are indirectly validated by the CONTRAIL observations, through the reconstruction of the chemical distributions (as evident in Figs 8–11).*"
This is only partially true, because you have a kind of free parameters, these are the original mixing ratio from the deep stratosphere (k=1) and the mixing ratios resided in the ExUTLS (k=5) during the 90 days of the backward trajectory simulation. Herewith, the interaction between CONTRAIL observed and trajectory-based mixing ratios can be adjusted. This is to my opinion most obvious for CO. The estimated CO for k=1 and 5 compensates other errors, e.g. chemical decay of CO during the transport from source region to the ExUTLS (see also other specific and general comments above).

In association with the change of the reconstruction method, the free parameters have been eliminated.

P.12, L.18-24: Both problems discussed here briefly, non-linear tropospheric trend and the lack of agreement in reconstruction of CONTRAIL observations during summer (Fig. 10e), concern mainly CO2, so please clarify and mention this here.

For the non-linear trend, the statement "In this study, linear trends for $CH_4$, $N_2O$, $SF_6$, and $CO_2$ are assumed for the reconstruction. Although this is a simplified treatment, given the length of the analysis period, these trends are roughly constant over this time period with the exception of $CH_4$, and the $CH_4$ reconstructions are more strongly affected by chemical loss, as is evident in a comparison of Figs 5a and 8a" has been added in P14, L33ff.
For the disagreement during summer, the statement "particularly for CO2 (Fig. 5e)" has been added in P14, L26.

P.12, L.24-26: It is true that the equivalent latitude-potential temperature (EqLat-Theta) coordinate system accounts for dynamical features in the stratosphere, because adiabatic motion is dominant in this strongly stably stratified region of the atmosphere. This is not true for the troposphere which is much more unstable (low static stability). Potential temperature (and PV) are not conserved or only for a much shorter timescale, because diabatic motion is much more prominent there. Therefore, potential temperature and equivalent latitude are not the coordinate system of choice in the troposphere. Also the problem of tracer uplift from the PBL into the UT during summer (most prominent for CO2, see above) is not minimised in an EqLat-Theta coordinate system.

Thank you for this instructive comment. The description "which are dynamically conserved quantities in the stratosphere. In the troposphere, which is more unstable, potential temperature and potential vorticity are not conserved, or are conserved only on much short timescales, because of diabatic motion. It should be noted that tracer uplift from the LT into the UT during summer (particularly for $CO_2$, as discussed above) cannot be reduced with the coordinate system employed here" has been added in P14, L27-30.

P.13, L.7: The mentioned role of Asian monsoon (ASM) is very likely, but it is speculation here, because it is not shown in this study, how much of the trajectories originated from the ASM (see also comment above).

The authors simply confirm that the trajectories originating in Asian region increased in summer season, so the statement "in association with the Asian summer monsoon" has been deleted.

P.13, L.15: "*The reconstructions agree well with CONTRAIL measurements in the ExUTLS.*" If this is one key messages of the summary then the limitations of the 90-days backward trajectories and the sensitivity of the results due to this limitation have to be discussed in much more detail (see general and specific comments above).

The 10-years trajectory has been employed as described in the General Reply.

P.13, L.23-24:"*This method provides a means to understand both dynamical transport and chemical distribution from a new perspective.*" There has been done a lot to understand dynamical, tracer transport and chemical processes in the UTLS. Some of these studies has been mentioned in this review and should be discussed in relation to the results in this manuscript. As outlined in my general comment, I am not convinced that the actual manuscript could contribute to the actual state of knowledge, but the results should be at least discussed in this framework. The uniqueness of this approach here, combination of different tracers and backward trajectories, could to my opinion be exploited much better, if one would use 10-years instead of 90-days backward trajectories.

The 10-years trajectory has been used, and the authors believe that this study has been significantly improved.

In addition to above revision following the reviewers' comments, the authors have added new Appendix (Appendix B and relating two figures) in the revised manuscript to visualize large perspective of seasonal variation in ExUTLS.

References:

Appenzeller, C., J. R. Holton, and K. H. Rosenlof (1996), Seasonal variation of mass transport across the tropopause, J. Geophys. Res., 101(D10), 15071-15078.

Birner, T., and H. Bönisch (2011), Residual circulation trajectories and transit times into the extratropical lowermost stratosphere, Atmos. Chem. Phys., 11(2), 817-827, doi:10.5194/acp-11-817-2011.

Bönisch, H., A. Engel, J. Curtius, T. Birner, and P. Hoor (2009), Quantifying transport into the lowermost stratosphere using simultaneous in-situ measurements of SF6 and CO2, Atmos. Chem. Phys., 9(16), 5905-5919, doi:10.5194/acp-9-5905-2009.

Diallo, M., Legras, B., and Chédin, A. (2012): Age of stratospheric air in the ERA-Interim, Atmos. Chem. Phys., 12, 12133-12154, https://doi.org/10.5194/acp-12-12133-2012, 2012.

Diallo, M., B. Legras, E. Ray, A. Engel, and J. A. Añel (2017), Global distribution of CO2 in the upper troposphere and stratosphere, Atmos. Chem. Phys., 17(6), 3861-3878, doi:10.5194/acp-17-3861-2017.

Engel, A., M. Strunk, M. Muller, H. P. Haase, C. Poss, I. Levin, and U. Schmidt (2002), Temporal development of total chlorine in the high-latitude stratosphere based on reference distributions of mean age derived from CO2 and SF6, J. Geophys. Res., 107(D12), doi:Artn 4136 Doi 10.1029/2001jd000584.

Engel, A., et al. (2006), Highly resolved observations of trace gases in the lowermost stratosphere and upper troposphere from the Spurt project: an overview, Atmos. Chem. Phys., 6, 283-301, doi:DOI 10.5194/acp-6-283-2006.

Fueglistaler, S., H. Wernli, and T. Peter (2004), Tropical troposphere-to-stratosphere transport inferred from trajectory calculations, Journal of Geophysical Research: Atmospheres, 109(D3), doi:10.1029/2003JD004069.

Fueglistaler, S., M. Bonazzola, P. H. Haynes, and T. Peter (2005), Stratospheric water vapor predicted from the Lagrangian temperature history of air entering the stratosphere in the tropics, Journal of Geophysical Research: Atmospheres, 110(D8), doi:10.1029/2004JD005516.

Fueglistaler, S., and P. H. Haynes (2005), Control of interannual and longer-term variability of stratospheric water vapor, Journal of Geophysical Research: Atmospheres, 110(D24), doi:10.1029/2005JD006019.

Gettelman, A., P. Hoor, L. L. Pan, W. J. Randel, M. I. Hegglin, and T. Birner (2011), The Extratropical Upper Troposphere and Lower Stratosphere, Rev. Geophys., 49, doi:Artn Rg3003, Doi 10.1029/2011rg000355.

Hall, T. M., and R. A. Plumb (1994), Age as a Diagnostic of Stratospheric Transport, J. Geophys. Res., 99(D1), 1059-1070.

Hoor, P., C. Gurk, D. Brunner, M. I. Hegglin, H. Wernli, and H. Fischer (2004), Seasonality and extent of extratropical TST derived from in-situ CO measurements during SPURT, Atmos. Chem. Phys., 4, 1427-1442.

Hoor, P., H. Fischer, and J. Lelieveld (2005), Tropical and extratropical tropospheric air in the lowermost stratosphere over Europe: A CO-based budget, Geophys. Res. Lett., 32(7), L07802, doi:10.1029/2004gl022018.

Hoor, P., H. Wernli, M. I. Hegglin, and H. Boenisch (2010), Transport timescales and tracer properties in the extratropical UTLS, Atmos. Chem. Phys., 10(16), 7929-7944, doi:10.5194/acp-10-7929-2010.

Madonna, E., H. Wernli, H. Joos, and O. Martius (2014), Warm Conveyor Belts in the ERA-Interim Dataset (1979–2010). Part I: Climatology and Potential Vorticity Evolution, Journal of Climate, 27(1), 3-26, doi:10.1175/jcli-d-12-00720.1.

Ploeger, F., and T. Birner (2016), Seasonal and inter-annual variability of lower stratospheric age of air spectra, Atmos. Chem. Phys., 16(15), 10195-10213, doi:10.5194/acp-16-10195-2016.

Plumb, R. A. (2007), Tracer interrelationships in the stratosphere, Rev. Geophys., 45(4), Artn Rg4005, doi:10.1029/2005rg000179.

Ray, E. A., F. L. Moore, J. W. Elkins, G. S. Dutton, D. W. Fahey, H. Vomel, S. J. Oltmans, and K. H. Rosenlof (1999), Transport into the Northern Hemisphere lowermost stratosphere revealed by in situ tracer measurements, J. Geophys. Res., 104(D21), 26565-26580.

Riese, M., F. Ploeger, A. Rap, B. Vogel, P. Konopka, M. Dameris, and P. Forster (2012), Impact of uncertainties in atmospheric mixing on simulated UTLS composition and related radiative effects, J. Geophys. Res., 117(D16), D16305, doi:10.1029/2012jd017751.

Schoeberl, M. R., L. C. Sparling, C. H. Jackman, and E. L. Fleming (2000), A Lagrangian view of stratospheric trace gas distributions, Journal of Geophysical Research: Atmospheres, 105(D1), 1537-1552, doi:10.1029/1999JD900787.

Shepherd, T. G. (2002), Issues in Stratosphere-troposphere Coupling, Journal of the Meteorological Society of Japan. Ser. II, 80(4B), 769-792, doi:10.2151/jmsj.80.769.

Škerlak, B., M. Sprenger, and H. Wernli (2014), A global climatology of stratosphere–troposphere exchange using the ERA-Interim data set from 1979 to 2011, Atmos. Chem. Phys., 14(2), 913-937, doi:10.5194/acp-14-913-2014.

Tilmes, S., et al. (2010), An aircraft-based upper troposphere lower stratosphere O3, CO, and H2O climatology for the Northern Hemisphere, Journal of Geophysical Research: Atmospheres, 115(D14), doi:10.1029/2009JD012731.

Vogel, B., et al. (2016), Long-range transport pathways of tropospheric source gases originating in Asia into the northern lower stratosphere during the Asian monsoon season 2012, Atmos. Chem. Phys., 16(23), 15301-15325, doi:10.5194/acp-16-15301-2016.

---

## Author Comment (AC2) · 27 Feb 2019

GENERAL REPLY

The authors thank the two reviewers for their constructive comments and helpful suggestions that have improved the manuscript.

As mentioned by the both reviewers, the reconstruction method had some shortcomings, but the reviewers, especially reviewer #1, gave concrete and great suggestions to resolve them. The manuscript has been largely revised and improved along the suggestions. The authors describe first the revisions in the reconstruction method as a General Reply.

The suggestions are as follows:
(1) use of 10-year trajectory instead of 90-day, which can eliminate the remaining trajectories in the UTLS (k=5);
(2) reconstruction of chemical passive tracers with evaluation of transport timescale (as the first step); and
(3) reconstruction of chemical active tracers including chemical decay (as the second step).
The authors have made these procedures and some relating revisions as follows.

For point (1), as noted by the reviewers, the remaining trajectories in the UTLS (k=5) could be eliminated, i.e., it was confirmed that the all trajectories are categorized into any origins of k = 1, 2, 3, or 4 within 10 years (Note that the criteria have also been revised to avoid some shortcomings). In addition, the inversion method to estimate tracer mixing ratio for k=1 and 5 in the original manuscript could be also eliminated, i.e., the all tracers in the ExUTLS have been reconstructed only from their mixing ratios assumed in the high-latitude LT, mid-latitude LT, and tropical troposphere.

For point (2), using the 10-year trajectory, Age of Air (AoA) as well as SF6 and CO2 distributions have been reconstructed from the trajectories including the "Tail correction" (e.g., Diallo et al., 2012). The CH4, N2O, CO have been also reconstructed without any chemical decay in this step. AoA has been also estimated using observed SF6 mixing ratios obtained by CONTRAIL, and then the two AoAs have been compared to correct transport timescale expressed in the trajectories.

For point (3), the chemical active tracers, CH4, N2O, and CO, are finally reconstructed with simulating their chemical loss along an "average path" (Schoeberl et al., 2000). The use of the concept of average path was also suggested by the reviewer #1. Based on this concept, the authors believe that the active

tracers have been successfully reconstructed together with estimation of seasonally depending their chemical loss rate.

In relation to a suite of these revisions, the latter half of section 2.1 (Estimating the origin fraction and AoA), large part of section 2.2 (Air mass original composition and reconstruction) were significantly revised, especially section 2.2 was reorganized and a new subsections 2.2.1 and 2.2.2 were created for reconstructions of chemical passive and active tracers, respectively. The analyzing results, figures, and discussions were thus also changed in association with the revision of the reconstruction method, but the main thesis was essentially not changed.

References:

Diallo, M., Legras, B., and Chédin, A.: Age of stratospheric air in the ERA-Interim, Atmos. Chem. Phys., 12, 12133-12154, https://doi.org/10.5194/acp-12-12133-2012, 2012.

Schoeberl, M. R., Sparling, L. C., Jackman, C. H., and Fleming, E. L.: A Lagrangian view of stratospheric trace gas distributions, Journal of Geophysical Research: Atmospheres, 105(D1), 1537-1552, doi:10.1029/1999JD900787, 2000.

The authors believe that the revised manuscript has been improved by incorporating the more appropriate reconstruction method.

Point-by-point responses to the comments of individual reviewer are provided below.

REPLY TO COMMENTS BY REVIEWER #2

The author is grateful for the thorough review and constructive comments on the manuscript. All of the points raised by the reviewer have been addressed. The major revision was made following the #1 reviewer's comments, please see first the "General Reply" section. Point-by-point responses are provided below, in red text.
The paper by Inai et al. investigates the air mass composition of the extratropical upper troposphere and lower stratosphere (exUTLS), and relates to CONTRAIL in-situ observations of several trace gas species (e.g., CH4, N2O, SF6, CO, CO2). The focus of the study lies on seasonal variations in air mass fractions and mixing ratios. In particular, it is found that seasonality in CH4, N2O and SF6 mixing ratios is controlled by transport from the deep stratosphere, due to the locations of the main chemical sink regions, whereas CO and CO2 are mainly controlled by transport from the tropical troposphere.

The air mass and tracer composition of the exUTLS is of particular relevance for global climate due to the radiative characteristics of this region. Hence, the present study fits well into the scope of ACP. The paper is well written and presented, and the current literature is appropriately discussed. I recommend publication after taking into account the several comments below, which I regard somewhere between major and minor.

Detailed comments:
1. Initialization: The trajectory initialization is somewhat unclear to me. In the respective text part it is said, that back trajectories are initialized between 0-140 deg E, but the corresponding Fig. 1 shows initialization locations for 0-360 deg E (P2/L27). How is the initialization done exactly?

Fig. 1 in the original manuscript was not correct. The figure has been corrected.

2. Model-measurement comparison: The CONTRAIL measurements are mainly from Siberia. How is the model-measurement comparison done, exactly at the measurement locations/times, or just averaged over specific regions? I would suggest to explain this clearly directly after the description of the trajectory initialization (P2).

Indeed, it could be designed to release trajectories exactly at the measurement location/time and it may make directly comparison with the CONTRAIL measurements; this study, however, attempts to reconstruct spatial-extending and uniform spatiotemporal tracer distributions as well as their transport, therefore, we choose to employ the grating initialization. Following the suggestion, the statement "Although trajectories could be released at the exact CONTRAIL measurement locations and times, the grating initialization is employed because this study attempt to obtain uniform spatiotemporal tracer distributions as well as their transports by capitalizing on the CONTRIAL measurements" has been added in P2, L31ff in the revised manuscript.

3. Reconstruction method: It would be good to mention (around P4/L10) that Eq. (2) holds only for species which are chemically inert along the trajectories. Can you give some quantitative information how well this assumption holds for the species and regions considered here? Perhaps some of the difference between reconstruction and measurements (e.g., Figs. 7-10) is related to neglecting chemistry effects?

As described in the General Reply, the reconstruction method has been revised. The chemically active species are reconstructed taking the chemical decay into account.

4. Origin mixing ratios (P4/L28): Why not using higher altitude in-situ measurements (e.g., from balloons, Geophysica/Halo/ER2/... aircrafts) or global satellite observations for the reference mixing ratios? At least the "inversion method" outlined below could be validated with such data.

As described in the General Reply, the reconstruction method has been revised. In association with the revision, the "inversion method" has been eliminated.

5. Minima in tracer distributions around 370K in spring/summer (P7/L21ff): I do not think these minima are just artifacts of the reconstruction. The fact that spring/summer transport of young tropical air strengthens first around 380-400K, leading to a "sandwich" structure with older air masses below is consistent with recent findings by Krause et al. (2018) (see e.g.

their Fig. 14) and Ploeger and Birner (2016) (e.g., their Fig. 7).

In agreement with these papers, Fig. 9/10 show evidence for strongest polward transport above about 380K, causing the mixing ratio minima below. I would suggest to discuss these distributions more appropriately.

Thank you for this informative comment and suggestion. Though the sandwich structures have changed their aspects due to the revision of reconstruction method, they have appeared at around 350 K as shown in Fig. 17. Following above suggestion, the following statements have been added (P9, L13-17): "In particular, all five chemical species show minima at ~350 K north of 60° N equivalent latitude. These minima might be formed by remainder of the deep stratospheric air masses which were transported during spring. The tracer minima near ~350 K at high equivalent latitudes begin forming in June. This "sandwich" structure in the ExUTLS has been reported by Ploeger and Biner (2016) for summer and by Krause et al. (2018) for spring. In agreement with their studies, the sandwich structures can show evidence for strong poleward transport above ~400 K, leading to mixing ratio minima at lower altitudes."

6. Trajectory method: Kinematic trajectories show stronger dispersion compared to diabatic trajectories (e.g., Schoeberl et al., 2003). Are the results presented here robust also for diabatic transport? At least include appropriate discussion in Sect. 4.3 ("Limitations of the current study").

The authors have not used diabatic trajectory, so we do not explicitly know how much the results change. Instead, statements "Trajectory results also generally depend on the vertical condition, i.e., kinematic (employed by the current study) or diabatic (employed by, for example, Diallo et al., 2017). Previous studies suggest that using kinematic trajectories leads to a stronger dispersion and somewhat young bias in AoA estimates compared with using diabatic trajectories (e.g., Schoeberl et al., 2003; Diallo et al., 2012). Therefore, using diabatic trajectories in this analysis might result in a correction factor ($\gamma TT$) of <1.5." have been added in Sect. 4.3 (P14, L1-4).

Specific and technical comments:
P1/L29: maybe better "at/along the subtropical jet"?
Change made as suggested.

P3/L23: "...where IT satisfies..."?
Corrected.

P3/L28: What is the "actual value" what is referred to here? Observations? Which?

It is referred to observation. Statement "actual value" has been changed to "observed value."

P7/L29: ware –> were

Corrected.

P9/L10ff: The sentence "In addition ..." sounds unclear to me - I suggest rewording.

The sentence has been reworded to "In addition to seasonal variations in origin fractions, seasonal variations in the tracer mixing ratios in origin regions (Fig. 14) also affect chemical distributions in the ExUTLS." (P10, L24-25)

P9/L19: shown –> show

Corrected.

P10/L28ff: I don't understand the description of Fig. 15f. What PDF is integrated here (transit time pdf?). What is the unit of the y-axis? Please clarify and improve the description.

The integration is done for age spectrum, so statement "the value that is calculated by integration of "age spectrum" (PDF)" was added in P12, L14.

P12/L7: The Ploeger and Birner reference cited here is not in the reference list.

The reference has been added.

P12/L20: non-linear

Corrected.

In addition to above revision following the reviewers' comments, the authors have added new Appendix (Appendix B and relating two figures) in the revised manuscript to visualize large perspective of seasonal variation in ExUTLS.

References:
Krause et al. (2018), Atmos. Chem. Phys., 18, 6057-6073.
Schoeberl et al. (2003), J. Geophys. Res, 118, D3.

---

## Referee Report (RR1)

Reviewer (Comments):
**Re-Review of "Seasonal characteristics of chemical and dynamical transports into the extratropical upper troposphere/lower stratosphere" (ExUTLS) by Yoichi Inai et al.**

**Recommendation: Publication after minor revision**

The revised paper has improved significantly and the authors did an impressive job and addressed the questions raised in the review thoroughly. From my side, there are only a few open questions left open that should be addressed before publications.

The paper should be submitted after addressing the comments below.

**General comments:**

My only general comment is that I am still convinced that the additional use of tropopause related coordinates would improve the comparison with the CONTRAIL data set. This concerns the sections 3.3 and 4.1, where the authors compare the reconstructions of the different tracers with the CONTRAIL observations (Fig. 15-18 and Fig. 21). Especially in the region close to the tropopause, i.e. the 320 K potential temperature level in high- and mid-latitudes (or more precisely equivalent latitudes), the intercomparison suffers from the small number of observations. The reasons is, that the mixing ratio gradients across the tropopause are large and that the estimation of the mean values in these bins (320 +/-5 K pot. temp.  and 45°/75° +/-5° eq. lat.) needs sufficient statistics therefore. This is no major issue and it is nothing wrong about the way the intercomparison is done in the paper – it is just a suggestion.

**Specific comments:**

**P.6, L.14-17:** It is the right idea to use observation, i.e. Volk et al. (1997), to determine the chemical decay along the "average path" (AP) into the ExUTLS of the northern hemisphere – the region of interest in this work here. However, there are two caveats using the data set of Volk et al. (1997) here:

a) A large part of the ASHOE/MAESA campaign took place in the southern hemisphere. Therefore, this data set might not be the best representation for the chemical decay along the AP from the troposphere into the northern hemisphere ExUTLS. This should be at least mentioned in the context here.

b) The gradients dX/dAoA (X: Tracer mixing ratio and AoA: Age of Air) is not solely determined by the chemical decay along the AP in the stratosphere, but also by the tropospheric trend of the tracer that propagates into the stratosphere. That means, the observed gradients dX/dAoA by Volk et al. are partially influenced by the growth rates of the tropospheric time series of N2O and CH4 before 1994. Luckily, the growth rates of both tracers in the time interval 5 years before the individual observations took place are rather similar. For CH4, the growth rates are 7.9 ppb/year for ASHOA/MAESA and 5.3-7.5 ppb/year for CONTRAIL. For N2O, the trend since 1990 is very close to linear with a growth rate of 0.81 ppb/year. Therefore, this should not be a big issue for the analysis here, but it should be mentioned.

The numbers for CH4 and N2O growth rates shown above are derived from the reference data sets of NOAA ESRL available on their websites:
https://www.esrl.noaa.gov/gmd/hats/combined/N2O.html
https://www.esrl.noaa.gov/gmd/ccgg/trends_ch4/

**P.6, L.21-22:** "*..., so here it is assumed that the gradient of CO mixing ratio with respect to AoA is 20-times larger than that of CH4.*"
Could you please explain this in a bit more detail in the paper, why you assume that the gradient $dX_{CO}/dAoA$ is 20-time larger than $dX_{CH4}/dAoA$. The tropospheric trends of CO and CH4 and their chemistry in the stratosphere are slightly different, so I would expect a slightly different gradient of chemical loss along the AP. This will most probably not really be an issue for the result of this study, but the assumption should be motivated here.

**Fig 6:** The remaining fraction of CO in the stratosphere is not going down to zero as shown in the Figure 6, but reaches an equilibrium value due to production processes balancing the photochemical loss. The stratospheric equilibrium value is about 10-15 ppbv (e.g. Krause et al., 2018) which corresponds to about 10% of the tropospheric value.

**P.8, L.15-17:** "*..., CH4 and N2O in stratospheric air masses show distinct seasonal variations but somewhat different phase,...*"
Is there any explanation, why the seasonality in the stratosphere is different? Both tracers are rather long-lived in the lower stratosphere (shown in Fig. 14) and should therefore be dominated by transport processes. It is not really clear to me, how this should lead to such a difference in the seasonality. I would definitely expect for CH4, in the same way as for N2O and SF6, the lowest mixing ratios in the ExUTLS in spring and not in summer.

**Fig. 14 Panel (c):** "*Tropospheric CO*"
It looks rather unlikely to me that tropospheric background values for high-latitude CO are much lower than mid-latitude CO and comparable to tropical CO. This is also in contradiction to the meridional distribution shown in GLOBALVIEW-CO provided by NOAA ESRL. I would expect at least CO mixing ratios in the high-latitudes that are as high as in the mid-latitudes. The difference here is most likely caused by the fact that NOAA ESRL background values have been used for the high-latitudes, but own aircraft based measurements have been used for mid-latitudes and tropics that are maybe not filtered in order to retrieve background values. This means that the mid-latitude and tropical tropospheric CO time series, in contrast to the high-latitudes, containing a mixture of polluted and unpolluted air masses. Please clarify this issue.

See: https://www.esrl.noaa.gov/gmd/ccgg/globalview/co/co_intro.html

**Fig. 14 Panel (d):** "*Tropospheric SF6*"
The same seems also to be the case for tropospheric SF6. High-latitude SF6 should show higher values than tropical SF6, see e.g. Rigby et al. (2010) or Waugh et al. (2013). Please clarify also.

See: https://www.esrl.noaa.gov/gmd/hats/combined/SF6.html

**P.11, L.13-14:** "*This seasonal variation in the upper ExUTLS is consistent with observational estimates by Hoor et al. (2004) and Strahan et al. (2007).*"

This is true, but the comparison with the results from Hoor et al. (2004) have to be taken with some caution, because this study analyse the $CO_2$ seasonal cycle in tropopause relative coordinates and therefore the direct comparison is not straightforward.

**P.13, L.10-12:** *"These overestimations of $N_2O$ and $CH_4$ mixing ratios for the original stratospheric air masses might be due to overestimation of the AoA and/or overestimation of the origin fraction of air masses originating in the deep branch of the BDC."*
Maybe I am wrong, but I think that the diagnosed overestimation of N2O and CH4 in May/November cannot be simply related to an overestimation of the origin fractions of deep stratospheric air. Assuming that the origin fraction from deep BDC would be smaller than the estimated N2O values from Andrews et al. (2001) would become higher, but also the N2O values reconstructed from the trajectories in this study would become higher.

**P.13, L.33:** *"…, the relationship approaches those of Andrews et al. (2001) and Röckmann et al. (2011)."*
It could be helpful to extend this sentence a bit (*"…, the **N2O-AoA and CH4-AoA** relationship approaches those of Andrews et al. (2001) and Röckmann et al. (2011) respectively."*) for a better readability and clarification.

**References:**

Krause, J., Hoor, P., Engel, A., Plöger, F., Grooß, J.-U., Bönisch, H., Keber, T., Sinnhuber, B.-M., Woiwode, W., and Oelhaf, H.: Mixing and ageing in the polar lower stratosphere in winter 2015–2016, Atmos. Chem. Phys., 18, 6057-6073, https://doi.org/10.5194/acp-18-6057-2018, 2018.

Rigby, M., Mühle, J., Miller, B. R., Prinn, R. G., Krummel, P. B., Steele, L. P., Fraser, P. J., Salameh, P. K., Harth, C. M., Weiss, R. F., Greally, B. R., O'Doherty, S., Simmonds, P. G., Vollmer, M. K., Reimann, S., Kim, J., Kim, K.-R., Wang, H. J., Olivier, J. G. J., Dlugokencky, E. J., Dutton, G. S., Hall, B. D., and Elkins, J. W.: History of atmospheric $SF_6$ from 1973 to 2008, Atmos. Chem. Phys., 10, 10305-10320, https://doi.org/10.5194/acp-10-10305-2010, 2010.

Waugh, D. W., et al. (2013), Tropospheric SF6: Age of air from the Northern Hemisphere midlatitude surface, J. Geophys. Res. Atmos., 118, 11,429–11,441, doi:10.1002/jgrd.50848.

---

## Author Response (AR2)

REPLY TO COMMENTS BY REVIEWER #1

The authors are grateful too much again for the additional review and constructive comments on the manuscript. Point-by-point responses are detailed below, in red text.

Reviewer (Comments):
Re-Review of "Seasonal characteristics of chemical and dynamical transports into the extratropical upper troposphere/lower stratosphere" (ExUTLS) by Yoichi Inai et al.

Recommendation: Publication after minor revision

The revised paper has improved significantly and the authors did an impressive job and addressed the questions raised in the review thoroughly. From my side, there are only a few open questions left open that should be addressed before publications.

The paper should be submitted after addressing the comments below.

General comments:
My only general comment is that I am still convinced that the additional use of tropopause related coordinates would improve the comparison with the CONTRAIL data set. This concerns the sections 3.3 and 4.1, where the authors compare the reconstructions of the different tracers with the CONTRAIL observations (Fig. 15-18 and Fig. 21). Especially in the region close to the tropopause, i.e. the 320 K potential temperature level in high- and mid-latitudes (or more precisely equivalent latitudes), the intercomparison suffers from the small number of observations. The reasons is, that the mixing ratio gradients across the tropopause are large and that the estimation of the mean values in these bins (320 +/-5 K pot. temp. and 45°/75° +/-5° eq. lat.) needs sufficient statistics therefore. This is no major issue and it is nothing wrong about the way the intercomparison is done in the paper – it is just a suggestion.

As pointed out, there is steep tracer gradient at around the tropopause, therefore it might be better to employ the tropopause related coordinates. It might be able to provide finer structures of the origin fractions and reconstructions in the region close to the tropopause. It is also true that a number of studies employ relative height with reference to the tropopause from such standpoint, however, the authors found that employing such

tropopause related coordinates makes an artificial bias in seasonal variation of observational data. Now, we are researching this issue and evaluating the impact, and it seems to be up to 10% of seasonal amplitude for CH4, N2O, SF6, and CO2 according to our preliminary results (we are also preparing the manuscript about this issue as to submit AMT journal).

In the current paper, the authors choose to use the current coordinate to avoid this issue even if it might lead bleary structures of the origin fractions and reconstructions in the region close to the tropopause.

Specific comments:

P.6, L.14-17: It is the right idea to use observation, i.e. Volk et al. (1997), to determine the chemical decay along the "average path" (AP) into the ExUTLS of the northern hemisphere – the region of interest in this work here. However, there are two caveats using the data set of Volk et al. (1997) here:

a) A large part of the ASHOE/MAESA campaign took place in the southern hemisphere. Therefore, this data set might not be the best representation for the chemical decay along the AP from the troposphere into the northern hemisphere ExUTLS. This should be at least mentioned in the context here.

To mention the arising point, the statement "Note that there are two caveats for this assumption. The first is that a large part of Volk's data was obtained in the Southern Hemisphere. Therefore, they may not be the best representation for chemical decay along the AP from the troposphere into the Northern Hemisphere ExUTLS." has been added in P6, L22--24 in the revised manuscript.

b) The gradients dX/dAoA (X: Tracer mixing ratio and AoA: Age of Air) is not solely determined by the chemical decay along the AP in the stratosphere, but also by the tropospheric trend of the tracer that propagates into the stratosphere. That means, the observed gradients dX/dAoA by Volk et al. are partially influenced by the growth rates of the tropospheric time series of N2O and CH4 before 1994. Luckily, the growth rates of both tracers in the time interval 5 years before the individual observations took place are rather similar. For CH4, the growth rates are 7.9 ppb/year for ASHOA/MAESA and

5.3-7.5 ppb/year for CONTRAIL. For N2O, the trend since 1990 is very close to linear with a growth rate of 0.81 ppb/year. Therefore, this should not be a big issue for the analysis here, but it should be mentioned.

The numbers for CH4 and N2O growth rates shown above are derived from the reference data sets of NOAA ESRL available on their websites:
https://www.esrl.noaa.gov/gmd/hats/combined/N2O.html
https://www.esrl.noaa.gov/gmd/ccgg/trends_ch4/

To mention the arising point, the statement "The second is that the relationship between AoA and the chemical loss rate is not only determined by the chemical decay along the AP in the stratosphere, but also by the tropospheric trend of tracers that propagate into the stratosphere. However, the trends of CH4 and N2O over the five years before the individual observations in Volk et al. (1997) and in the current study are similar. Therefore, this should not significantly affect the analysis presented here." has been added in P6, L24--28 in the revised manuscript.

P.6, L.21-22: "..., so here it is assumed that the gradient of CO mixing ratio with respect to AoA is 20-times larger than that of CH4."
Could you please explain this in a bit more detail in the paper, why you assume that the gradient dXCO/dAoA is 20-time larger than dXCH4/dAoA. The tropospheric trends of CO and CH4 and their chemistry in the stratosphere are slightly different, so I would expect a slightly different gradient of chemical loss along the AP. This will most probably not really be an issue for the result of this study, but the assumption should be motivated here.

The reason why we assumed the 20-times larger gradient of dXCO/dAoA than that of dXCH4/dAoA is based on some observational results (e.g., Herman et al., 1999). According this comment together with the next comment, the dXCO/dAoA has changed to more appropriate value. Please see the next.

Fig 6: The remaining fraction of CO in the stratosphere is not going down to zero as shown in the Figure 6, but reaches an equilibrium value due to production processes balancing the photochemical loss. The stratospheric equilibrium value is about 10-15

ppbv (e.g. Krause et al., 2018) which corresponds to about 10% of the tropospheric value.

According to Herman et al. (1999), the chemical loss rate of CO is 1) almost 20-times larger than that of CH4 at ~100 hPa level, and 2) it becomes small exponentially with increasing height. Further, as pointed out by above comment, 3) Krause et al. (2018) shows that the remaining amount of CO reaches an equilibrium value which is about 10-15 ppb.

Based on the three points, the assumed dXCO/dAoA and chemical loss rate of CO have been changed as shown in the renewed Fig. 6c and f. Relating this revision, the statement "According to Herman et al. (1999), the chemical loss rate of CO is estimated to be 20-times larger than that of CH4 in the tropical UTLS and it exponentially attenuates with increasing height. Furthermore, the remaining fraction of CO in the stratosphere reaches an equilibrium value because of production processes balancing the chemical loss, which corresponds to ~10 % of the tropospheric value (e.g., Krause et al., 2018). Thus, the chemical decay for CO is assumed to be an e-folding time with respect to AoA (Tau_AoA^CO) that Tau_AoA^CO = 0.7*2.0^Gannma, where Gamma is AoA in years. The corresponding relative abundance of CO and the gradient with respect to AoA are evaluated as shown in Fig. 6c." has been added in P6. L33--P7, L5.

Following this change, relating figures (Fig. 6 and 8) have been renewed.

P.8, L.15-17: "..., CH4 and N2O in stratospheric air masses show distinct seasonal variations but somewhat different phase,..."

Is there any explanation, why the seasonality in the stratosphere is different? Both tracers are rather long-lived in the lower stratosphere (shown in Fig. 14) and should therefore be dominated by transport processes. It is not really clear to me, how this should lead to such a difference in the seasonality. I would definitely expect for CH4, in the same way as for N2O and SF6, the lowest mixing ratios in the ExUTLS in spring and not in summer.

For the first sentence, the reason why the seasonality in the stratosphere is different between CH4 and N2O is affected by the difference chemical sinks for the two species. About this matter, the statement "The slight phase difference between CH4 and N2O in stratospheric air masses might reflect differences in their chemical loss mechanisms. The chemical loss of CH4 is controlled by reactions with OH, O(1D), and Cl, whereas

that of N2O is controlled primary by photolysis and secondarily by reactions with O(1D). Therefore, the seasonality of CH4 is affected not only by seasonal variations of solar radiation that is primary and direct factor for N2O loss, but also by OH abundance along a MP." has been made in P14, L28--32.
The acronym MP means "modal path," please see the following comment about it.

For the question in the latter part, the authors have examined those seasonality which the seasonal variations in SF6 and AoA are on-phase whereas they differ from those in CH4 and N2O, furthermore the seasonality in CH4 and N2O are also slightly different. As a result, they could be interpreted as a combination of seasonally varying chemical loss rates on a transport time-scale near the modal time and a path close to that of the modal path. This argument has been described in newly made Sect. 4.3 with a new figure (Fig. 21). Please see the section, and the authors believe that it provides the answer to the question in the referee's comment.

Relating above, the statement "Further discussion of this topic is included in the next section, together with the mechanism driving the Pi/2 phase-lagged, i.e., rolling relationship between CH4 and N2O mixing ratios and AoA in stratospheric air masses." has been added in P13, L15--16.

Fig. 14 Panel (c): "Tropospheric CO"
It looks rather unlikely to me that tropospheric background values for high-latitude CO are much lower than mid-latitude CO and comparable to tropical CO. This is also in contradiction to the meridional distribution shown in GLOBALVIEW-CO provided by NOAA ESRL. I would expect at least CO mixing ratios in the high-latitudes that are as high as in the mid- latitudes. The difference here is most likely caused by the fact that NOAA ESRL background values have been used for the high-latitudes, but own aircraft based measurements have been used for mid-latitudes and tropics that are maybe not filtered in order to retrieve background values. This means that the mid-latitude and tropical tropospheric CO time series, in contrast to the high-latitudes, containing a mixture of polluted and unpolluted air masses. Please clarify this issue.

See: https://www.esrl.noaa.gov/gmd/ccgg/globalview/co/co_intro.html

Fig. 14 Panel (d): "Tropospheric SF6"

The same seems also to be the case for tropospheric SF6. High-latitude SF6 should show higher values than tropical SF6, see e.g. Rigby et al. (2010) or Waugh et al. (2013). Please clarify also.

See: https://www.esrl.noaa.gov/gmd/hats/combined/SF6.html

As pointed by these comments, the background CO and SF6 mixing ratios in high-latitude are comparable to those in mid-latitude and larger than those in tropical troposphere. Further as also pointed above, the own aircraft measurements used for mid-latitudes and tropics may contain a mixture of polluted and unpolluted air masses. In real atmosphere, the tracer distribution in the ExUTLS is determined not only by influx of background air masses, but also by that of polluted air masses. Therefore, it must take such polluted air masses into account to reconstruct plausible distribution of such trace gases in the ExUTLS. Since, in particular, for CO and SF6, their artificial sources are mostly distributed in mid-latitude LT, therefore we might have been able to reconstruct the tracer distribution which agrees well with the CONTRAIL measurements. A more proper approach might be to assign background values with the addition of incremental values due to pollution assumed in each latitude region. Such an approach will be the focus of future work, instead, the following statements have made in Sect. 2.2.1 and 4.4 to mention this issue.

- The statement in P5, L20--21 in the previous manuscript has been changed to "This averaging procedure is discussed in more detail in Sect. 4.4 together with a caveat for the use of those aircraft measurement data which has somewhat different implication from the following ground-based data." (P5, L23--25 in the revised manuscript).

- The following description has been added in the last paragraph in Sect. 4.4.
"For the aircraft measurements data used as original mixing ratios for air masses originating in the tropical troposphere and mid-latitude LT, particularly, those collected by TU over sea close to Japan may contain a mixture of polluted and unpolluted air masses in some degree. On this point, they have different implication from measurement data obtained by background monitoring sites which is employed as that for the high-latitude LT. For example, the background CO and SF6 mixing ratios in mid-latitude are comparable to those in high-latitude troposphere (as confirmed, e.g., in NOAA/ESRL                                 web                                 sites; https://www.esrl.noaa.gov/gmd/ccgg/globalview/co/co_intro.html,

https://www.esrl.noaa.gov/gmd/hats/combined/SF6.html), whereas those used for the mid-latitude LT are significantly larger than those for high-latitude LT, except during winter. In real atmosphere, the tracer distribution in the ExUTLS is determined not only by influx of background air masses, but also by that of polluted air masses. Therefore, it must take such polluted air masses into account to reconstruct plausible distribution of trace gases, i.e., CO and SF6, in the ExUTLS. Since their artificial sources are mostly distributed in mid-latitude LT, therefore we might have been able to reconstruct the tracer distribution which agrees well with the CONTRAIL measurements. A more proper approach would be to assign background values with the addition of incremental values due to pollution assumed in each latitude region, such an approach will be the focus of our future work".

In addition to above revision, to mention the averaging procedure taken for air masses originating in the region k = 2 more explicitly, Fig. 13 (Fig. 14 in the previous manuscript) has been modified to indicate pre-averaged values, and the caption has been changed to "Seasonal variations in (a) CH4, (b) N2O, (c) CO, (d) SF6, and (e) CO2 mixing ratios assumed to (solid green) tropical tropospheric, (orange) mid-latitude LT, and (red) high-latitude LT air masses. Note that green dashed lines in (a–e) show the average mixing ratios of the tropical tropospheric and mid-latitude LT, and they are practically assigned to tropical tropospheric air masses to account for underestimations of vertical transport from the LT in the trajectory analysis…".

P.11, L.13-14: "This seasonal variation in the upper ExUTLS is consistent with observational estimates by Hoor et al. (2004) and Strahan et al. (2007)."
This is true, but the comparison with the results from Hoor et al. (2004) have to be taken with some caution, because this study analyse the CO2 seasonal cycle in tropopause relative coordinates and therefore the direct comparison is not straightforward.

Indeed, the main results of Hoor et al. (2004) are presented and discussed in the tropopause relative coordinate, however, they also present the CO2 seasonal variation in potential temperature coordinate (Fig. 9b in their paper). The result is essentially same as that in their main results. Therefore, the authors think that this statement is not wrong as is.

P.13, L.10-12: "These overestimations of N2O and CH4 mixing ratios for the original stratospheric air masses might be due to overestimation of the AoA and/or overestimation of the origin fraction of air masses originating in the deep branch of the BDC."

Maybe I am wrong, but I think that the diagnosed overestimation of N2O and CH4 in May/November cannot be simply related to an overestimation of the origin fractions of deep stratospheric air. Assuming that the origin fraction from deep BDC would be smaller than the estimated N2O values from Andrews et al. (2001) would become higher, but also the N2O values reconstructed from the trajectories in this study would become higher.

For safety description, the statement "and/or overestimation of the origin fraction of air masses originating in the deep branch of the BDC." has been deleted.

P.13, L.33: "..., the relationship approaches those of Andrews et al. (2001) and Röckmann et al. (2011)."

It could be helpful to extend this sentence a bit ("..., the N2O-AoA and CH4-AoA relation- ship approaches those of Andrews et al. (2001) and Röckmann et al. (2011) respectively.") for a better readability and clarification.

Thank you for your consideration for making readers easily understand. Change made as suggested.

Additional revision:

- To smoothly read, the statement "Note that these values for stratospheric air masses are estimated based on their final state, unlike the case for regions k = 2, 3, and 4, for which the values correspond to their original state". has been moved from P8, L18--20 in the previous to P8, L30--31 in the revised manuscripts.

- Relating to the revision of methodology and added discussions, the last part of Abstract and the 3rd paragraph in Summary have been modified.

- Fig. 6 and 7 in the previous manuscript has been merged.

[revised manuscript text omitted]